# *CodeBrain*: Bridging Decoupled Tokenizer and Multi-Scale Architecture for EEG Foundation Model

**Jingying Ma**[1][*], **Feng Wu**[1][*], **Qika Lin**[1][†], **Yucheng Xing**[1,3], **Chenyu Liu**[4],
**Ziyu Jia**[5,6][†], **Mengling Feng**[1,2]

[1]Saw Swee Hock School of Public Health, National University of Singapore, Singapore
[2]Institute of Data Science, National University of Singapore, Singapore
[3]Guangzhou Research Translation and Innovation Institute,
National University of Singapore, Guangzhou, China
[4]College of Computing and Data Science, Nanyang Technological University, Singapore
[5]Beijing Key Laboratory of Brainnetome and Brain-Computer Interface,
Institute of Automation, Chinese Academy of Sciences, Beijing, China
[6]Brainnetome Center, Institute of Automation, Chinese Academy of Sciences, Beijing, China
```
{jingyingma, wufeng, xingyucheng}@u.nus.edu;
qikalin@foxmail.com; chenyu003@e.ntu.edu.sg
jia.ziyu@outlook.com; mornin@nus.edu.sg
```

## Abstract

Electroencephalography (EEG) provides real-time insights into brain activity and supports diverse applications in neuroscience. While EEG foundation models (EFMs) have emerged to address the scalability issues of task-specific models, current approaches still yield clinically uninterpretable and weakly discriminative representations, inefficiently capturing global dependencies and neglecting important local neural events. We present *CodeBrain*, a two-stage EFM designed to fill this gap. In the first stage, we introduce the **TFDual-Tokenizer**, which decouples heterogeneous temporal and frequency EEG signals into discrete tokens, quadratically expanding the representation space to enhance discriminative power and offering domain-specific representation-level interpretability by suggesting potential links to neural events and spectral rhythms. In the second stage, we propose the multi-scale **EEGSSM** architecture, which combines structured global convolution with sliding window attention to efficiently capture both sparse long-range and local dependencies, reflecting the brain's small-world topology. Pretrained on the largest public EEG corpus, *CodeBrain* achieves strong generalization across eight downstream tasks and ten datasets under distribution shifts, supported by comprehensive ablations, scaling-law analyzes, and interpretability evaluations. The code and the pretrained weights are available at https://github.com/jingyingma01/CodeBrain.

## 1 Introduction

Electroencephalography (EEG) captures brain activity via scalp electrodes (Niedermeyer & da Silva, 2005) and provides high temporal-resolution signals for neuroscience and cognitive research (da Silva, 2013). To enable automated analysis, researchers have developed various task-specific models for applications such as sleep staging (Lee et al., 2025; Ma et al., 2025a), emotion recognition (Liu et al., 2024b; Jia et al., 2025), motor imagery (Li et al., 2020; Jia et al., 2020), and other applications (Guerra et al., 2024; Hu et al., 2024). However, building separate models from scratch for each task is resource-intensive and limits scalability, as shared knowledge across tasks cannot be effectively

---

[*]Co-first authors.
[†]Corresponding authors.

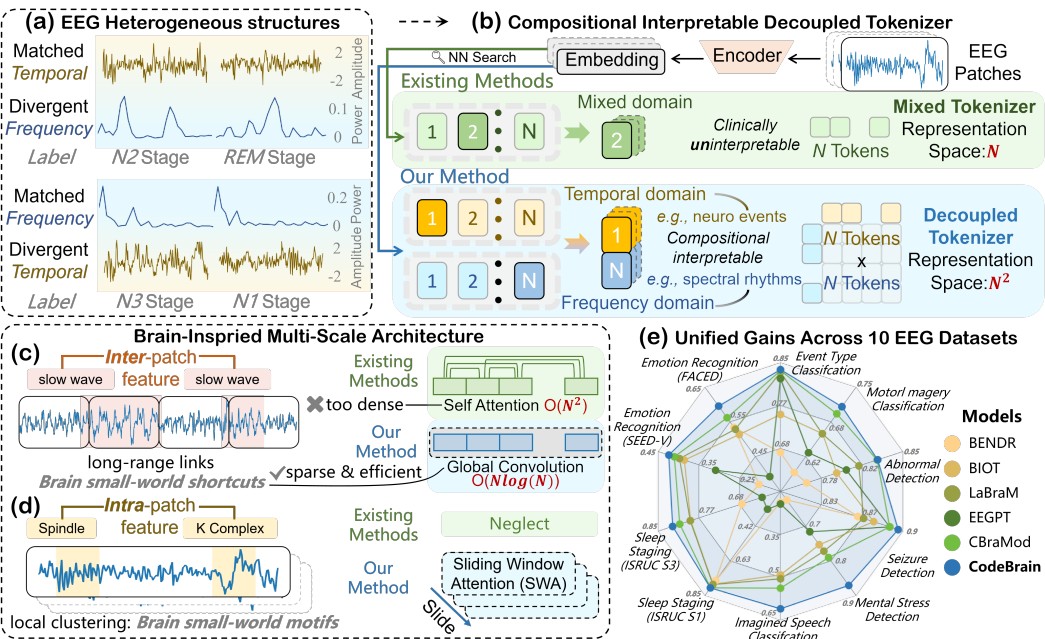

Figure 1: Rationale and overview of *CodeBrain* beyond existing EFMs. **(a)** EEG signals are heterogeneous, as patches matched in one domain may diverge in the other. **(b)** We then propose a decoupled tokenizer for domain-specific interpretable representations while expanding the representation space. **(c-d)** Inspired by the brain's small-world topology, a multi-scale architecture further captures inter-patch dependencies efficiently, while modeling overlooked intra-patch neural events. **(e)** These designs deliver performance gains across ten EEG datasets.

leveraged. Moreover, variations in channel configurations and input lengths across EEG tasks further hinder knowledge transfer. To tackle these issues, EEG foundation models (EFMs) are developed to learn universal representations for diverse downstream tasks (Zhou et al., 2025a).

Inspired by masked self-supervised pretraining in natural language processing (Van Den Oord et al., 2017; Devlin et al., 2019), current EFMs commonly adopt patch-wise representation learning: EEG signals are divided into patches, encoded into latent representations, and trained to reconstruct the masked portions. While this approach offers flexibility across varying channel configurations and input lengths by adjusting the patch number and arrangement, direct raw-signal reconstruction (Wang et al., 2024a; 2025) remains challenging due to the inherent noise and variability of EEG. To mitigate this, recent studies have introduced codebook-based tokenization (Jiang et al., 2024; Pradeepkumar et al., 2025), which abstracts away low-level fluctuations and provides a more robust latent space. Despite these advances, existing EFMs still face critical limitations, calling for new architectures.

**Failing to Decouple Heterogeneous EEG for Domain-Specific Interpretability.** Recent EFMs adopt vector quantization for noise-robust representation (Jiang et al., 2024; Pradeepkumar et al., 2025), following the VQ-VAE framework originally designed for images, where homogeneous visual features make a single tokenizer sufficient (Van Den Oord et al., 2017; Mentzer et al., 2024; Jin et al., 2025; Lin et al., 2025a). In contrast, EEG exhibits heterogeneous structures: temporal and frequency components reflect distinct aspects of brain activity (Miwakeichi et al., 2004). As illustrated in Fig. 1(a), signals matched in one domain may diverge in the other. Therefore, a mixed tokenizer can conflate domain-specific patterns (Liu et al., 2024c), weakening representation capacity and producing tokens difficult to align with clinically interpretable neural events or spectral rhythms.

**Struggling with Efficiently Modeling Global Brain Dependencies.** EEG signals exhibit sparse global dependencies and strong local correlations, reflecting the brain's small-world topology (Bullmore & Sporns, 2009; Bassett & Bullmore, 2006; He et al., 2009). Efficient modeling of such structure requires capturing relationships in a scalable way. However, most EFMs (Yang et al., 2023; Jiang et al., 2024; Wang et al., 2024a; 2025) adopt Transformer architectures with fully connected

self-attention (Vaswani et al., 2017). This over-connected design is misaligned with the brain's sparse structure and struggles to efficiently capture global dependencies due to its quadratic complexity with sequence length (Tay et al., 2021; Hong et al., 2025; Tegon et al., 2025).

**Neglecting Local Dependencies within EEG Patches.** EEG signals exhibit rich local waveform structures over short temporal windows, reflecting crucial transient neural events (e.g., sleep waveforms in Fig. 1(d)) (Tatum IV, 2021). However, most existing EFMs represent each EEG patch as a single token and apply attention mechanisms only at the patch level (Wang et al., 2025; Jiang et al., 2024), thereby ignoring important local dependencies within patches.

To address the above challenges, we propose *CodeBrain*, a novel EEG foundation model that integrates a decoupled tokenizer for domain-specific representation-level interpretability with a brain-inspired multi-scale architecture. *CodeBrain* is trained in two stages. In the first stage, we introduce the **TFDual-Tokenizer** (Fig. 1(b)), which decouples temporal and frequency EEG components into discrete tokens. In the second stage, we develop **EEGSSM**, a masked self-supervised framework inspired by the brain's small-world topology. EEGSSM adopts a structured global convolution backbone, conceptually related to recent state-space sequence models (Smith et al., 2023; Li et al., 2022; Gu et al., 2022a), for sparse and efficient global modeling with a sliding window attention (SWA) mechanism for capturing local neural events overlooked by prior studies(Fig. 1(c–d)). Our detailed contributions are summarized as follows:

- **Decoupled Tokenizer for Domain-Specific Representation-Level Interpretability.** We propose the TFDUAL-TOKENIZER, which decouples temporal and frequency EEG components into discrete tokens. This design quadratically expands the representation space, and qualitative analyzes suggest that some tokens correspond to neural events and spectral rhythms. A contrastive objective is further applied to the temporal branch to stabilize training. To the best of our knowledge, this is the first tokenizer in EFMs to provide domain-specific representation-level interpretability.

- **Brain Small-World Topology Inspired Multi-Scale Architecture.** We design EEGSSM, a patch-wise self-supervised framework for EEG. Guided by the brain's small-world topology, it employs structured global convolution to capture sparse long-range temporal dependencies and sliding window attention to model local neural events. In addition, dynamic positional embeddings are used to flexibly learn spatial channel correlations.

- **Strong Generalization and Comprehensive Validation.** Pretrained on the largest publicly available EEG corpus, TUEG (Obeid & Picone, 2016), *CodeBrain* achieves strong performance on eight downstream tasks across ten datasets (Fig. 1(e)) with distribution shifts in cohorts and channel configurations. This suggests the model design plays a central role in generalization. Comprehensive ablations, scaling-law analyzes, together with visualization and quantitative analyzes, further confirm its robustness, scalability, and provide domain-specific representation-level interpretability.

## 2 METHODOLOGY

### 2.1 MODEL ARCHITECTURE

We introduce *CodeBrain*, a two-stage pretraining framework designed to learn interpretable and universal EEG representations (Fig. 2). The model is motivated by complementary goals: 1) **domain-specific interpretability** via decoupled tokenization of heterogeneous temporal and frequency information, achieved by the proposed **TFDual-Tokenizer**, and 2) **multi-scale modeling** of EEG sequences inspired by the brain's small-world topology, addressed by the **EEGSSM** framework. This design lets Stage 1 learn a tokenizer of patch-level codes, while Stage 2 leverages it for EEG representations. We next provide a formal definition of the two stages to clarify their respective roles.

**Stage 1: Decoupled Tokenization.** Given a normalized EEG patch $\mathbf{x} \in \mathbb{R}^L$, where $L$ is the patch length, our goal is to discretize $\mathbf{x}$ into temporal and frequency tokens, enabling domain-specific representation learning. Specifically, let $Vt \in \mathbb{R}^{K \times D}$ and $Vf \in \mathbb{R}^{K \times D}$ denote the temporal and frequency codebooks, where $K$ is the vocabulary size and $D$ is the embedding dimension of each token. The tokenizer function is defined as: $vt, vf = f_{\text{tokenizer}}(\mathbf{x}), \quad vt, vf \in \mathbb{R}^D$.

**Stage 2: EEG Representation Learning.** Given unlabeled EEG sequences $\mathcal{X} = \{X_m\}_{m=1}^N$, where each $X_m \in \mathbb{R}^{C \times f \times T}$ consists of $C$ channels, sampling rate $f$, and $T$ seconds. We divide each

sequence into $n$ non-overlapping patches of $t$ seconds, so each patch length is $L = f \cdot t$. The goal is to train an encoder $f_{\text{enc}} : \mathbb{R}^{C \times n \times L} \to \mathbb{R}^{C \times n \times D}$ that produces latent representations $Z_m = f_{\text{enc}}(X_m)$.

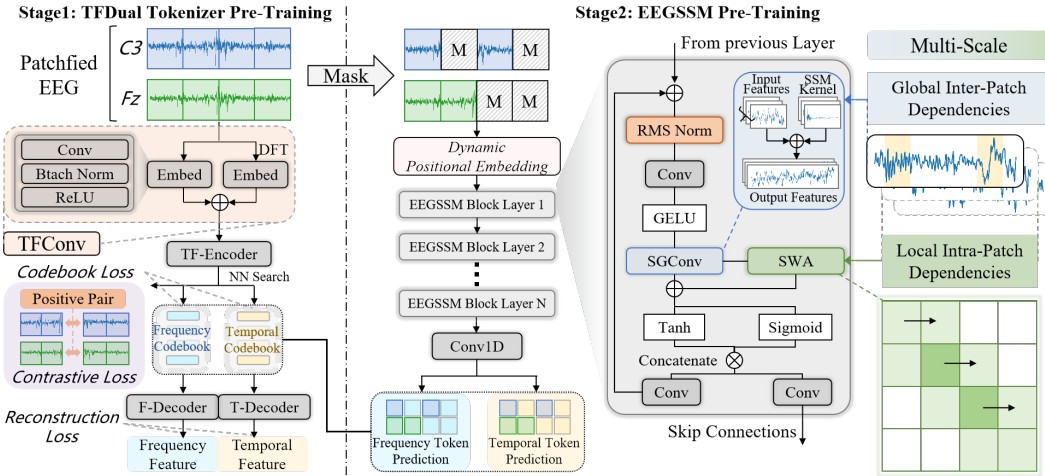

Figure 2: Overview of the *CodeBrain* framework. **Left:** TFDUAL-TOKENIZER learns to discretize EEG signals into temporal and frequency tokens using two separate codebooks, by reconstructing both the temporal waveforms and the frequency-domain magnitude and phase. **Right:** EEGSSM learns representations by predicting the discrete tokens of masked patches generated by TFDUAL-TOKENIZER.

## 2.2 TFDUAL-TOKENIZER PRETRAINING

Our TFDUAL-TOKENIZER includes a shared *neural encoder*, a *dual tokenizer* with separate code-books, and two *decoders*. The *neural encoder* extracts joint time-frequency embeddings from EEG patches, which are then discretized into temporal and frequency tokens by the *dual tokenizer*. Each token stream is reconstructed by a decoder to supervise codebook learning in its respective domain.

**Neural Encoder** For each patch $\mathbf{x}_i \in \mathbb{R}^L$, we apply the Discrete Fourier Transform (DFT) (Cooley & Tukey, 1965) to obtain its frequency representation:

$$\mathbf{x}_i[k] = DFT(\mathbf{x}_i), \tag{1}$$

where $\mathbf{x}_i[k]$ denotes the $k$-th frequency component. $\mathbf{x}_i[k]$ and the $\mathbf{x}_i$ are fed into the **TFConv module**, where they are processed in parallel through stacks of convolutional, batch normalization, and ReLU layers. The temporal representation $e_i^t = TFConv(\mathbf{x}_i)$ and frequency representation $e_i^f = TFConv(\mathbf{x}_i[k])$ are concatenated to form a time-frequency embedding $e_i^p = Concat\{e_i^t, e_i^f\}$.

To get patch representation $\tilde{e}_i$, we then add a positional embedding $e_{\text{pos}}$ and feed it into **TF-Encoder**:

$$\tilde{e}_i = Encoder(e_i^p + e_{pos}). \tag{2}$$

We choose a Transformer encoder here since this stage is *patch-to-token*, where its ability to model local contextual relations makes it well-suited for capturing patch-level patterns. To keep the tokenizer channel-agnostic, $e_{\text{pos}}$ is shared temporal embeddings without channel-specific identities.

**Dual Tokenizer** We use two separate tokenizers with distinct codebooks for the temporal and frequency domains, denoted as $vt_j, vf_j \in \mathbb{R}^D$, where $j = 1, \ldots, K$. Given the patch representation $\tilde{e}_i$ from the neural encoder, each tokenizer independently selects the nearest code from its codebook:

$$pt_i = \arg\min_j \|\tilde{e}_i - vt_j\|^2, \quad pf_i = \arg\min_j \|\tilde{e}_i - vf_j\|^2, \tag{3}$$

where $pt_i$ and $pf$ denote the closest positions for the embeddings in the temporal and frequency domain codebook. The effectiveness of the Dual Tokenizer is based on the following proposition:

**Proposition 2.1** *Decoupling temporal and frequency codebooks yields representations that are no less effective than those from a joint codebook.*

For this Proposition, we provide the proof in Appendix D, empirical validation in Sections 3.4 and 3.5, and interpretability analysis of the Dual Tokenizer in Appendix B.

**Frequency Codebook Training** To train the frequency codebook, we reconstruct amplitude and phase from the code embeddings. For each EEG patch, we apply the DFT to obtain the frequency representation: $\mathbf{x}_i[k] = \mathrm{Re}\{\mathbf{x}_i[k]\} + j \cdot \mathrm{Im}\{\mathbf{x}_i[k]\}$ where $\mathrm{Re}\{\mathbf{x}_i[k]\}$ and $\mathrm{Im}\{\mathbf{x}_i[k]\}$ are the real part and imaginary part respectively, then the amplitude and phase can be calculated as:

$$A_i = \sqrt{\mathrm{Re}(\mathbf{x}_i[k])^2 + \mathrm{Im}(\mathbf{x}_i[k])^2}, \quad \phi_i = \arctan 2\left(\mathrm{Im}(\mathbf{x}_i[k]), \mathrm{Re}(\mathbf{x}_i[k])\right). \quad (4)$$

We use z-score normalization to ensure stable training. The code embedding $vf_i$, retrieved from the frequency codebook, is passed through the **F-Decoder**, which consists of a Transformer encoder followed by two linear layers:

$$y_i^A = Encoder(MLP(\tilde{e}_i)), \quad y_i^P = Encoder(MLP(\tilde{e}_i)), \quad (5)$$

where $y_i^A$ and $y_i^P$ are the predicted amplitude and phase, respectively. The frequency codebook's training objective is the mean squared error (MSE) loss:

$$\mathcal{L}_i^f = ||y_i^A - A_i||_2^2 + ||y_i^P - \phi_i||_2^2. \quad (6)$$

**Temporal Codebook Training** Direct reconstruction of temporal features might lead to non-convergence (Jiang et al., 2024). To address this, we combine contrastive loss with reconstruction loss to train a temporal codebook. Inspired by studies on physiological signals (Kiyasseh et al., 2021), we assume temporal dependencies exist between EEG segments, especially within the same channel. For an EEG segment $X_m \in \mathbb{R}^{C \times n \times L}$, we split it into two halves of length $n/2$. We use **TF-Encoder** in Eq. (2) to obtain the representations of these two parts $X_{m1}, X_{m2} \in \mathbb{R}^{C \times \frac{2}{n} \times L}$ separately:

$$e_{mi}^h = Encoder(X_{mi}), i \in \{1, 2\}. \quad (7)$$

We encourage the latent representations of different parts within a single segment $e_{m1}^h$ and $e_{m2}^h$ to be similar, while making those of the same part across different segments $e_{mi}^h$ and $e_{sk}^h$ as distinct as possible, where $X_s \in \mathbb{R}^{C \times n \times L}, s \neq m, k \neq i$. SimCLR loss (Chen et al., 2020) is used for training:

$$\mathcal{L}_m^{CL} = -\log \frac{\exp\left(\mathrm{sim}(e_{m1}^h, e_{m2}^h)/\tau\right)}{\sum\limits_{k=1}^{2N} \mathbb{1}_{[k \neq i]} \exp\left(\mathrm{sim}(e_{mi}^h, e_{sk}^h)/\tau\right)}, \quad (8)$$

where $\mathrm{sim}(e_{m1}^h, e_{m2}^h)$ is the cosine similarity, $\tau > 0$ is the temperature parameter, and $\mathbb{1}_{[k \neq i]}$ is an indicator function used to exclude itself. We introduce a **T-Decoder** to reconstruct raw signals from the temporal code embedding $\tilde{e}_i$, which consists of a Transformer encoder followed by a linear projection. Let $y_t$ be the Transformer output from Eq. (2); the overall training objective is:

$$\mathcal{L}_i^t = \mathcal{L}_{CL} + ||y_i^t - \mathbf{x}_i||_2^2, \quad y_i^t = Encoder(MLP(\tilde{e}_i)). \quad (9)$$

Finally, the training objective for the TFDual-Tokenizer can be defined as:

$$\mathcal{L}_{\text{tokenizer}} = \sum_{X_m \in \mathcal{X}} \mathcal{L}_m^{CL} + \sum_{X_m \in \mathcal{X}} \sum_{i=0}^{n} \mathcal{L}_i^f + \mathcal{L}_i^t + \underbrace{||\mathrm{sg}(\tilde{e}_i) - vt_{pt_i}||^2 + ||\mathrm{sg}(\tilde{e}_i) - vf_{pf_i}||^2}_{\text{codebook loss}}$$
$$+ \underbrace{||\tilde{e}_i - \mathrm{sg}(vt_{pt_i})||^2 + ||\tilde{e}_i - \mathrm{sg}(vf_{pf_i})||^2}_{\text{commitment loss}}, \quad (10)$$

where $\mathrm{sg}(\cdot)$ denotes stop-gradient to avoid updating encoder parameters.

## 2.3 EEGSSM Pretraining

In this stage, we introduce a novel convolutional structured state space model framework, called **EEGSSM**, composed of multiple EEGSSM blocks. To adapt to unseen channels, we first learn dynamic positional embeddings using a single depthwise 2D convolution with an asymmetric kernel, following the ACPE design (Wang et al., 2025), enabling the model to learn relative inter-channel structures and generalize across heterogeneous EEG channel layouts. The resulting features are processed by EEGSSM blocks. Another 1D convolutional layer then maps the output back to the token space for reconstructing the indices of masked tokens produced by the TFDual-Tokenizer.

**EEGSSM Block**  Our EEGSSM block is composed of several blocks, which are integrated together through a residual connection mechanism. An EEGSSM block consists of a Layer Normalization, SGConv layer, SWA layer, and a gating component. Afterward, we feed the intermediate variables into the SGConv layer to obtain a global receptive field through convolution SSM.

*SGConv Layer.* SGConv is a structured SSM model (see Appendix C) using convolution architecture, and its convolution structure can be represented as a DFT formula:

$$y = F_N^{-1} D_k F_N u, D_k = \text{diag}(\overline{K} F_N), \tag{11}$$

where $F_N$ denotes the DFT matrix of size $N$, and the convolution can be computed in $O(N \log N)$ via FFT. As a type of convolutional SSM, SGConv improves the convolution kernel $\overline{K}$ in Eq. (11) by introducing two features: sparse parameterization and kernel decay, making SGConv easier and more efficient to compute compared to the traditional S4 kernel. Let $L$ be the length of the input sequence. The convolution kernel $\overline{K}$ of SGConv is composed of several sub-kernels. Assuming the size of the first sub-kernel is $d$, with parameters $w_i \in \mathbb{R}^{d \times d}$, then the number of sub-kernels can be expressed as $N = \log_2(\frac{L}{d}) + 1$. The convolution kernel $\overline{K}$ in Eq. (11) can thus be initialized as:

$$\overline{K} = \frac{1}{Z}[k_0, k_1, ..., k_{N-1}], k_i = \alpha^i \text{Upsample}_{2^{max[i-1,0]}d}(w_i), \tag{12}$$

where $\alpha$ denotes the decay coefficient, usually chosen to be 0.5, which induces the decaying structure, and Upsample$(x)$ denotes upsampling $x$ to length $l$. We also introduce the normalization constant $Z$ to ensure that the convolution operation does not change the scale of the input.

*Sliding Window Attention Layer.* We included a sliding window attention (SWA) layer (Liu et al., 2021b) to capture fine-grained local temporal dependencies. Specifically, a small fixed-length window is applied along the temporal dimension, allowing the model to directly attend to local contextual information within each window. By restricting attention to local regions, SWA effectively captures intra-patch temporal dynamics that are overlooked by previous models. Furthermore, under a fixed window size, SWA reduces the quadratic complexity of global self-attention to linear complexity, enabling efficient training.

*Gate Mechanism.* we use a gating mechanism to control the output of the block. We employ a gated unit similar to Wavenet (Obeid & Picone, 2016), which can suppress useless or irrelevant features and help stabilize training in deep networks. We concatenate the global features output $y_{sg}$ by the SGConv layer with the local features output $y_{swa}$ by SWA and feed them into a gated unit:

$$z = tanh(W_f \times Concat(y_{sg}, y_{swa})) \odot \sigma(W_g \times Concat(y_{sg}, y_{swa})), \tag{13}$$

$$y_1 = \text{Conv}(z), y_2 = \text{Conv}(z), \tag{14}$$

where $tanh(\cdot)$ and $\sigma(\cdot)$ are tanh function and sigmoid function, $\odot$ denotes an element-wise multiplication operator, $W_f$ and $W_g$ are learnable convolution filters. $y_1$ becomes the input to the next block, while $y_2$ will be aggregated to the output of SSM blocks through a skip connection.

**Pre-Training Objective**  To help the EEGSSM model learn general EEG representations, we use a Masked Autoencoder (MAE) for self-supervised pre-training. For a patched sample $X = \{x_i \mid i \in [1, 2, \ldots, C]\}$, we randomly generate a mask $\mathcal{M} = \{m_i \mid i \in [1, 2, \ldots, C]\}$ from a Bernoulli distribution of $r$ proportion, where $m_i \in \{0, 1\}$. We reconstruct the token indices of masked EEG patches from the TFDUAL-TOKENIZER by cross-entropy loss. Let $y_i$ denotes the output of the EEGSSM block, the probability that the EEG signal matches the corresponding token $v_i$ in the codebooks:

$$p(v_i|x_i) = softmax(Conv1D(y_i)). \tag{15}$$

Suppose the size of the pre-training set is $N$, the final cross-entropy loss is:

$$\mathcal{L}_p = -\sum_{j=0}^{N} \sum_{n \in \{m_i=1\}}^{C} p(v_{nj}|x_{nj}). \tag{16}$$

## 3 EXPERIMENTS

### 3.1 DATASETS

**Pre-Training**. We pretrain *CodeBrain* on the TUH EEG Corpus (Obeid & Picone, 2016), the largest publicly available EEG dataset to date. Data processing follows a standardized pipeline based on prior works (Wang et al., 2025; Zhou et al., 2025b): recordings shorter than 5 minutes are excluded, and the first and last minutes of each segment are removed. We retain 19 commonly used EEG channels (C3, C4, Cz, F3, F4, Fp1, Fp2, F7, F8, Fz, O1, O2, P3, P4, Pz, T3, T4, T5, T6), selected based on the international 10–20 system for electrode placement (HH, 1958). We apply band-pass filtering (0.3-75 Hz), and notch filtering at 60 Hz to remove noise. The data is resampled to 200 Hz and divided into 30-second non-overlapping segments. Segments with absolute amplitudes over 100 $\mu$V are filtered out. To normalize the signals, each value is divided by 100. Each segment is split into 1-second windows, resulting in 570 EEG patches per sample. After preprocessing, 1,109,545 samples (about 9,246 hours) are retained for pretraining.

**Downstream Tasks.** We evaluate *CodeBrain* on eight downstream tasks across ten public EEG datasets, which span diverse applications and exhibit distribution shifts from the pretraining dataset to assess generalizability. Detailed dataset configurations are in Table 3.1. We perform cross-subject or cross-session splits with strict separation between training, validation, and test sets. For **FACED**, we use 80 subjects for training, 20 for validation, and the remaining 23 for testing. In **SEED-V**, each session consists of 15 trials, which are evenly divided into training, validation, and test sets. **ISRUC_S3** consists of 10 subjects, for which we apply an 8:1:1 cross-subject split. **MentalArithmetic** consists of 36 subjects, and we use a 7:1:1 cross-subject split. **BCIC2020-T3** follows the official competition protocol. In **CHB-MIT**, we use recordings from 19 subjects for training, and from 2 subjects each for validation and testing. Additional dataset details are provided in the Appendix G.

Table 1: Summary of downstream tasks and associated EEG datasets.

| Downstream Tasks | Datasets | #Channels | Length | #Samples | Class |
|---|---|---|---|---|---|
| Emotion Recognition | FACED (Chen et al., 2023) | 32 | 10s | 10,332 | 9-class |
| | SEED-V (Liu et al., 2021a) | 62 | 1s | 117,744 | 5-class |
| Sleep Staging | ISRUC_S1 (Khalighi et al., 2016) | 6 | 30s | 86,320 | 5-class |
| | ISRUC_S3 (Khalighi et al., 2016) | 6 | 30s | 8,500 | 5-class |
| Imagined Speech Classification | BCIC 2020-T3 (Jeong et al., 2022) | 64 | 3s | 6,000 | 5-class |
| Mental Stress Detection | Mental Arithmetic (Mumtaz, 2016) | 20 | 5s | 1,707 | 2-class |
| Seizure Detection | CHB-MIT (Shoeb, 2009) | 16 | 10s | 326,993 | 2-class |
| Motor Imagery Classification | SHU-MI (Goldberger et al., 2000) | 32 | 4s | 11,988 | 2-class |
| Event Type Classification | TUEV (Obeid & Picone, 2016) | 16 | 5s | 112,491 | 6-class |
| Abnormal Detection | TUAB (Obeid & Picone, 2016) | 16 | 10s | 409,455 | 2-class |

### 3.2 EXPERIMENT SETTINGS

**Experiment Setup.** (1) Pretraining Setup. All experiments are conducted on NVIDIA 40GB A100 GPUs. The TFDUAL-TOKENIZER is trained with temporal and frequency codebooks of 4096 codes (32 dimensions) for 20 epochs, using a batch size of 256 and a learning rate of 1e-4, across six A100 GPUs for approximately ten hours. An 8-layer EEGSSM backbone (15.17M) with a masking ratio of 0.5 is trained for 10 epochs, using a batch size of 256 on two A100 GPUs for about 24 hours.

(2) Finetuning Strategy. We evaluate the quality of the pretrained representations under full finetuning. All downstream task datasets are resampled to 200 Hz to match the pretraining configuration. A three-layer MLP is applied to aggregate channel information, compress the $x$-second sequence, and map the representation to the target class, with activation and dropout between layers.

**Baselines.** We compare our model with a comprehensive set of baseline models. Among the non-foundation baselines, **EEGNet** (Lawhern et al., 2018) and **EEGConformer** (Song et al., 2022) represent compact architectures designed for efficient EEG decoding. **ContraWR** (Yang et al., 2021) is a contrastive-learning–based small model, while **ST-Transformer** (Song et al., 2021) provides a transformer backbone. These models serve as representative lightweight baselines commonly adopted across EEG classification tasks.

Table 2: Comparison results of different methods on representative downstream tasks.

| Methods | FACED (9-Class) | | | SEED-V (5-Class) | | |
|---|---|---|---|---|---|---|
| | Cohen's Kappa | Weighted F1 | Balanced Acc | Cohen's Kappa | Weighted F1 | Balanced Acc |
| EEGNet | $0.3342 \pm 0.0251$ | $0.4124 \pm 0.0141$ | $0.4090 \pm 0.0122$ | $0.1006 \pm 0.0143$ | $0.2749 \pm 0.0098$ | $0.2961 \pm 0.0102$ |
| EEGConformer | $0.3858 \pm 0.0186$ | $0.4514 \pm 0.0107$ | $0.4559 \pm 0.0125$ | $0.1772 \pm 0.0174$ | $0.3487 \pm 0.0136$ | $0.3537 \pm 0.0112$ |
| ContraWR | $0.4231 \pm 0.0151$ | $0.4887 \pm 0.0078$ | $0.4887 \pm 0.0078$ | $0.1905 \pm 0.0188$ | $0.3544 \pm 0.0121$ | $0.3546 \pm 0.0105$ |
| ST-Transformer | $0.4137 \pm 0.0133$ | $0.4795 \pm 0.0096$ | $0.4810 \pm 0.0079$ | $0.1083 \pm 0.0121$ | $0.2833 \pm 0.0105$ | $0.3052 \pm 0.0072$ |
| BENDR | $0.4716 \pm 0.0095$ | $0.5340 \pm 0.0086$ | $0.5320 \pm 0.0083$ | $0.0335 \pm 0.0062$ | $0.2026 \pm 0.0330$ | $0.2231 \pm 0.0059$ |
| BIOT | $0.4476 \pm 0.0254$ | $0.5136 \pm 0.0112$ | $0.5118 \pm 0.0118$ | $0.2261 \pm 0.0262$ | $0.3856 \pm 0.0203$ | $0.3837 \pm 0.0187$ |
| LaBraM | $0.4698 \pm 0.0102$ | $0.5288 \pm 0.0188$ | $0.5273 \pm 0.0107$ | $0.2386 \pm 0.0209$ | $0.3974 \pm 0.0111$ | $0.3976 \pm 0.0138$ |
| EEGPT | $0.4639 \pm 0.0023$ | $0.3924 \pm 0.0017$ | $0.4607 \pm 0.0014$ | $0.1323 \pm 0.0062$ | $0.3090 \pm 0.0052$ | $0.3061 \pm 0.0044$ |
| CBraMod | $0.5041 \pm 0.0122$ | $0.5618 \pm 0.0093$ | $0.5509 \pm 0.0089$ | $0.2569 \pm 0.0143$ | $0.4101 \pm 0.0108$ | $0.4091 \pm 0.0097$ |
| CodeBrain | $\mathbf{0.5406} \pm 0.0084$ | $\mathbf{0.5953} \pm 0.0113$ | $\mathbf{0.5941} \pm 0.0098$ | $\mathbf{0.2735} \pm 0.0032$ | $\mathbf{0.4235} \pm 0.0022$ | $\mathbf{0.4137} \pm 0.0023$ |

| Methods | ISRUC_S3 (5-Class) | | | BCIC 2020-T3 (5-Class) | | |
|---|---|---|---|---|---|---|
| | Cohen's Kappa | Weighted F1 | Balanced Acc | Cohen's Kappa | Weighted F1 | Balanced Acc |
| EEGNet | $0.7396 \pm 0.0155$ | $0.7407 \pm 0.0184$ | $0.7121 \pm 0.0134$ | $0.4413 \pm 0.0102$ | $0.3016 \pm 0.0123$ | $0.4413 \pm 0.0096$ |
| EEGConformer | $0.7482 \pm 0.0164$ | $0.7501 \pm 0.0211$ | $0.7212 \pm 0.0181$ | $0.4488 \pm 0.0154$ | $0.3133 \pm 0.0183$ | $0.4506 \pm 0.0133$ |
| ContraWR | $0.7493 \pm 0.0150$ | $0.7513 \pm 0.0185$ | $0.7226 \pm 0.0164$ | $0.4407 \pm 0.0182$ | $0.3078 \pm 0.0218$ | $0.4257 \pm 0.0162$ |
| ST-Transformer | $0.7388 \pm 0.0195$ | $0.7399 \pm 0.0223$ | $0.7116 \pm 0.0197$ | $0.4247 \pm 0.0138$ | $0.2941 \pm 0.0159$ | $0.4126 \pm 0.0122$ |
| BENDR | $0.5995 \pm 0.0151$ | $0.6789 \pm 0.0142$ | $0.6352 \pm 0.0095$ | $0.0607 \pm 0.0093$ | $0.2379 \pm 0.0165$ | $0.2485 \pm 0.0075$ |
| BIOT | $0.7168 \pm 0.0119$ | $0.7834 \pm 0.0096$ | $0.7598 \pm 0.0109$ | $0.3650 \pm 0.0176$ | $0.4917 \pm 0.0079$ | $0.4920 \pm 0.0086$ |
| LaBraM | $0.7194 \pm 0.0162$ | $0.7843 \pm 0.0189$ | $0.7617 \pm 0.0122$ | $0.3800 \pm 0.0242$ | $0.5054 \pm 0.0205$ | $0.5060 \pm 0.0155$ |
| EEGPT | $0.6160 \pm 0.0856$ | $0.6375 \pm 0.0632$ | $0.6650 \pm 0.0311$ | $0.0567 \pm 0.0164$ | $0.2441 \pm 0.0105$ | $0.2453 \pm 0.0131$ |
| CBraMod | $0.7407 \pm 0.0251$ | $0.8056 \pm 0.0219$ | $0.7844 \pm 0.0126$ | $0.4216 \pm 0.0163$ | $0.5383 \pm 0.0096$ | $0.5373 \pm 0.0108$ |
| CodeBrain | $\mathbf{0.7671} \pm 0.0091$ | $\mathbf{0.8202} \pm 0.0071$ | $\mathbf{0.7856} \pm 0.0031$ | $\mathbf{0.5127} \pm 0.0065$ | $\mathbf{0.6101} \pm 0.0053$ | $\mathbf{0.6101} \pm 0.0052$ |

| Methods | Mental Arithmetic (2-Class) | | | CHB_MIT (2-Class) | | |
|---|---|---|---|---|---|---|
| | AUROC | PRAUC | Balanced Acc | AUROC | PRAUC | Balanced Acc |
| EEGNet | $0.7321 \pm 0.0108$ | $0.5763 \pm 0.0102$ | $0.6770 \pm 0.0116$ | $0.8048 \pm 0.0136$ | $0.1914 \pm 0.0182$ | $0.5658 \pm 0.0106$ |
| EEGConformer | $0.7424 \pm 0.0128$ | $0.5829 \pm 0.0134$ | $0.6805 \pm 0.0123$ | $0.8226 \pm 0.0170$ | $0.2209 \pm 0.0215$ | $0.5976 \pm 0.0141$ |
| ContraWR | $0.7332 \pm 0.0082$ | $0.5787 \pm 0.0164$ | $0.6631 \pm 0.0097$ | $0.8103 \pm 0.0144$ | $0.2279 \pm 0.0183$ | $0.6351 \pm 0.0122$ |
| ST-Transformer | $0.7132 \pm 0.0174$ | $0.5672 \pm 0.0259$ | $0.6631 \pm 0.0173$ | $0.8237 \pm 0.0491$ | $0.1422 \pm 0.0094$ | $0.5915 \pm 0.0195$ |
| BENDR | $0.6248 \pm 0.0765$ | $0.3661 \pm 0.0672$ | $0.5681 \pm 0.0448$ | $0.8632 \pm 0.0526$ | $0.3071 \pm 0.1240$ | $0.5609 \pm 0.0432$ |
| BIOT | $0.7536 \pm 0.0144$ | $0.6004 \pm 0.0195$ | $0.6875 \pm 0.0186$ | $0.8761 \pm 0.0284$ | $0.3277 \pm 0.0460$ | $0.7068 \pm 0.0457$ |
| LaBraM | $0.7721 \pm 0.0093$ | $0.5999 \pm 0.0155$ | $0.6909 \pm 0.0125$ | $0.8679 \pm 0.0199$ | $0.3287 \pm 0.0402$ | $0.7075 \pm 0.0358$ |
| EEGPT | $0.7162 \pm 0.0171$ | $0.5081 \pm 0.0275$ | $0.5597 \pm 0.0171$ | $0.8438 \pm 0.0066$ | $0.3073 \pm 0.0641$ | $0.5481 \pm 0.0151$ |
| CBraMod | $0.7905 \pm 0.0073$ | $0.6267 \pm 0.0099$ | $0.7256 \pm 0.0132$ | $0.8892 \pm 0.0154$ | $0.3689 \pm 0.0382$ | $\mathbf{0.7398} \pm 0.0284$ |
| CodeBrain | $\mathbf{0.8707} \pm 0.0209$ | $\mathbf{0.7177} \pm 0.0421$ | $\mathbf{0.7514} \pm 0.0203$ | $\mathbf{0.8961} \pm 0.0174$ | $\mathbf{0.4377} \pm 0.0288$ | $0.7273 \pm 0.0240$ |

For EEG foundation models with publicly available pretrained weights, we include five representative methods that cover different pretraining paradigms. **BENDR** (Kostas et al., 2021) adopts a contrastive learning framework. **BIOT** (Yang et al., 2023) uses patch-based continuous tokenization. **LaBraM** introduces discrete neural tokens through vector quantization (Jiang et al., 2024). **EEGPT** (Wang et al., 2024a) and **CBraMod** (Wang et al., 2025) rely on the masked reconstruction.

**Evaluation Metric.** For multi-class classification, we report **Cohen's Kappa**, **Weighted F1 score**, and **Balanced Accuracy**, with Kappa based on validation performance for testing. For binary classification, we use Area Under the ROC Curve (**AUROC**), Area Under the Precision-Recall Curve (**PRAUC**), and **Balanced Accuracy**, with AUROC based on validation performance for testing. All experiments are repeated with five random seeds, and we report the mean and standard deviation.

More details about hyperparameters, baselines, and evaluation metrics are provided in Appendix J, H.

## 3.3 COMPARISON WITH BASELINES

We ensure consistent data splits across all baselines. Results are reported on six representative downstream datasets, with additional results provided in Appendix I. As shown in Table 2, *CodeBrain* achieves consistent performance gains compared to baselines. For multi-class classification, it

Table 3: The results of ablation studies for tokenizer configurations and module components.

| Dataset | Codebook | CL | SWA | SGConv | Gate | Cohen's Kappa | Weighted F1 | Balanced Accuracy |
|---|---|---|---|---|---|---|---|---|
| **FACED** 9-Class | Dual | ✓ | ✓ | ✓ | ✓ | **0.5406** $\pm$ 0.0084 | **0.5953** $\pm$ 0.0113 | **0.5941** $\pm$ 0.0098 |
| | Temporal | ✓ | ✓ | ✓ | ✓ | 0.4618 $\pm$ 0.0072 | 0.5277 $\pm$ 0.0067 | 0.5217 $\pm$ 0.0056 |
| | Frequency | ✓ | ✓ | ✓ | ✓ | 0.5006 $\pm$ 0.0224 | 0.5607 $\pm$ 0.0201 | 0.5580 $\pm$ 0.0187 |
| | Mixed | ✓ | ✓ | ✓ | ✓ | 0.4676 $\pm$ 0.0061 | 0.5319 $\pm$ 0.0052 | 0.5281 $\pm$ 0.0049 |
| | Dual | ✗ | ✓ | ✓ | ✓ | 0.5222 $\pm$ 0.0082 | 0.5811 $\pm$ 0.0084 | 0.5765 $\pm$ 0.0074 |
| | Dual | ✓ | ✗ | ✓ | ✓ | 0.5192 $\pm$ 0.0092 | 0.5792 $\pm$ 0.0093 | 0.5736 $\pm$ 0.0075 |
| | Dual | ✓ | ✓ | ✗ | ✓ | 0.1936 $\pm$ 0.1637 | 0.2627 $\pm$ 0.1824 | 0.2858 $\pm$ 0.1467 |
| | Dual | ✓ | ✓ | ✓ | ✗ | 0.2578 $\pm$ 0.0340 | 0.3363 $\pm$ 0.0270 | 0.3431 $\pm$ 0.0316 |
| **SEED-V** 5-Class | Dual | ✓ | ✓ | ✓ | ✓ | **0.2735** $\pm$ 0.0032 | **0.4235** $\pm$ 0.0022 | **0.4137** $\pm$ 0.0023 |
| | Temporal | ✓ | ✓ | ✓ | ✓ | 0.2633 $\pm$ 0.0116 | 0.4152 $\pm$ 0.0092 | 0.4068 $\pm$ 0.0074 |
| | Frequency | ✓ | ✓ | ✓ | ✓ | 0.2665 $\pm$ 0.0208 | 0.4186 $\pm$ 0.0177 | 0.4098 $\pm$ 0.0147 |
| | Mixed | ✓ | ✓ | ✓ | ✓ | 0.2708 $\pm$ 0.0047 | 0.4214 $\pm$ 0.0044 | 0.4124 $\pm$ 0.0032 |
| | Dual | ✗ | ✓ | ✓ | ✓ | 0.2589 $\pm$ 0.0065 | 0.4129 $\pm$ 0.0056 | 0.4029 $\pm$ 0.0042 |
| | Dual | ✓ | ✗ | ✓ | ✓ | 0.2561 $\pm$ 0.0051 | 0.4106 $\pm$ 0.0042 | 0.4019 $\pm$ 0.0034 |
| | Dual | ✓ | ✓ | ✗ | ✓ | 0.2062 $\pm$ 0.0099 | 0.3707 $\pm$ 0.0075 | 0.3620 $\pm$ 0.0083 |
| | Dual | ✓ | ✓ | ✓ | ✗ | 0.2212 $\pm$ 0.0076 | 0.3826 $\pm$ 0.0057 | 0.3757 $\pm$ 0.0052 |
| **ISRUC_S3** 5-Class | Dual | ✓ | ✓ | ✓ | ✓ | **0.7671** $\pm$ 0.0091 | **0.8202** $\pm$ 0.0071 | **0.7856** $\pm$ 0.0031 |
| | Temporal | ✓ | ✓ | ✓ | ✓ | 0.7314 $\pm$ 0.0210 | 0.7916 $\pm$ 0.0181 | 0.7565 $\pm$ 0.0244 |
| | Frequency | ✓ | ✓ | ✓ | ✓ | 0.7390 $\pm$ 0.0601 | 0.7986 $\pm$ 0.0514 | 0.7728 $\pm$ 0.0361 |
| | Mixed | ✓ | ✓ | ✓ | ✓ | 0.7400 $\pm$ 0.0217 | 0.7999 $\pm$ 0.0171 | 0.7673 $\pm$ 0.0157 |
| | Dual | ✗ | ✓ | ✓ | ✓ | 0.7558 $\pm$ 0.0333 | 0.8130 $\pm$ 0.0264 | 0.7801 $\pm$ 0.0132 |
| | Dual | ✓ | ✗ | ✓ | ✓ | 0.7359 $\pm$ 0.0324 | 0.7950 $\pm$ 0.0259 | 0.7621 $\pm$ 0.0311 |
| | Dual | ✓ | ✓ | ✗ | ✓ | 0.6218 $\pm$ 0.0427 | 0.6956 $\pm$ 0.0279 | 0.6664 $\pm$ 0.0316 |
| | Dual | ✓ | ✓ | ✓ | ✗ | 0.5478 $\pm$ 0.0302 | 0.6429 $\pm$ 0.0307 | 0.6258 $\pm$ 0.0448 |

achieves the largest gain of +0.0911 in Cohen's Kappa (21.6%), +0.0718 in Weighted F1 score (13.3%), +0.0728 in Balanced Acc (13.5%) on *BCIC 2020-T3* over the strongest baseline (Wang et al., 2025). For binary classification, it achieves the largest gain of +0.0802 in AUROC (10.1%), +0.0910 in PRAUC (14.5%) and +0.0258 in Balanced Acc (3.6%) on *Mental Arithmetic* over the strongest baseline. These results demonstrate the superior generalizability of *CodeBrain*.

## 3.4 ABLATION STUDY

We conduct ablation studies on three datasets with the same five seeds as in the main experiments to evaluate the key components of *CodeBrain* (Table 3.3). Below is the detailed analysis:

(1) *Tokenizer configuration*: we compare the proposed TFDUAL-TOKENIZER (**Dual**) with variants using a single domain codebook (**Temporal** or **Frequency**) or a shared codebook that jointly reconstructs both domains (**Mixed**). Across all datasets, the Dual codebook consistently yields superior performance, suggesting that maintaining separate temporal and frequency codebooks captures complementary information more effectively than merging them.

(2) *Contrastive learning* (CL): We evaluate the impact of contrastive learning in TFDUAL-TOKENIZER pretraining. It leads to consistent gains, indicating a better capture of temporal patterns. In addition, CL stabilizes the training dynamics and promotes faster and more stable convergence of the temporal codebook. Further analysis and evidence are provided in Appendix F.

(3) *Components of* EEGSSM: Evaluating the **SWA**, **SGConv**, and **Gate** modules, which demonstrate improvements in EEG representation learning. Including the SWA module consistently improves performance, confirming its regularization effect in capturing local dependencies. The gating mechanism also shows large impact, as it effectively stabilizes fine-tuning and prevents overfitting.

(4) *Scaling Laws*: Prior works (Wang et al., 2025; Jiang et al., 2024) explored scaling with 1-1000 hours of EEG data for pretraining. We extend to 1k-9k hours and 3M-150M models, ranging from a 3.86M 3-layer model with a hidden size of 128 to a 146.75M 12-layer model with hidden size of 384, enabling a systematic exploration of scaling laws across depth and width. As shown in Figure 3, Kappa consistently improves with more data and parameters. These results confirm that larger models yield consistent but diminishing returns. More results and efficiency analysis are provided in Appendix N. These findings indicate that the 8-layer (15.17M) model is a balanced choice between performance and computational efficiency.

We further investigate several key design choices, including mask ratio, SWA window size, codebook size, patch size, SGConv kernel parameters, and subband contributions in Appendix K. Additional results on robustness (Appendix M), computational efficiency (Appendix L), low-resource settings (Appendix O), and pretraining dynamics (Appendix E) further demonstrate the stability and efficiency.

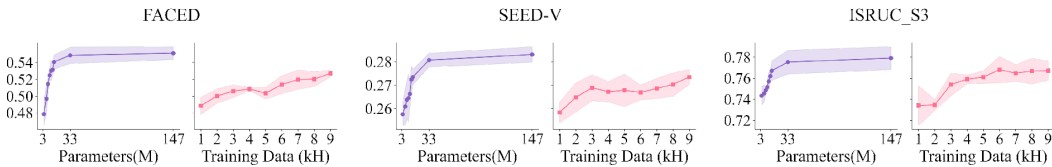

Figure 3: Model and training data scaling laws of *CodeBrain* across three datasets on Cohen's Kappa.

### 3.5 VECTOR VISUALIZATION

To demonstrate how the **TFDual-Tokenizer** models heterogeneous EEG, we visualize its learned temporal and frequency codes on the ISRUC_S3 dataset by mapping individual code indices back to their corresponding raw signals. As shown in Figure 4(a)(b), each domain-specific codebook captures meaningful representation-level structures: temporal codes align with neural events (e.g., slow waves), while frequency codes highlight spectral rhythms such as dominant delta activity, both of which are informative for sleep staging. However, in many cases, neither domain alone is sufficient, and richer structure emerges only from their composition, as in Figure 4(c)(d), where the same temporal code can pair with different frequency codes, and vise versa, to yield complementary representations. This decoupled design expands the representation space and enhances interpretability, with additional meaningful tokens to neurophysiological features and quantitative analyzes provided in Appendix B.

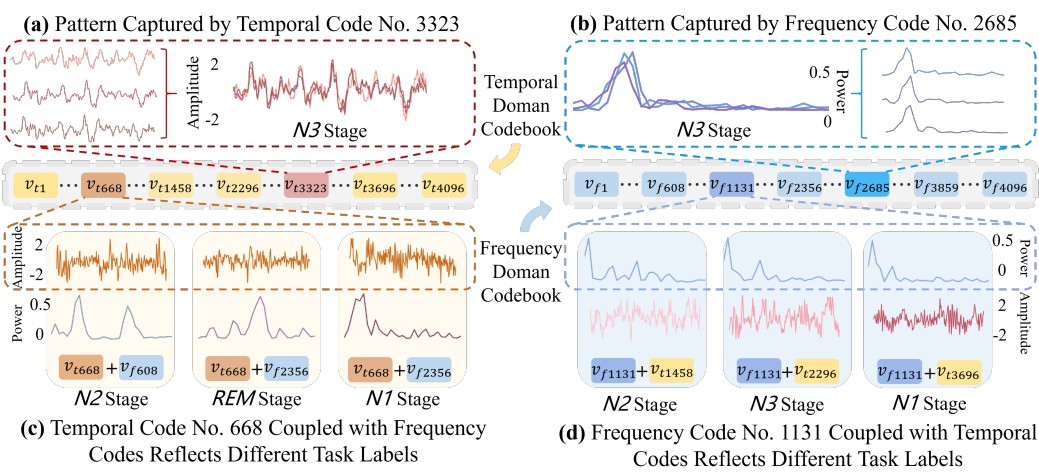

Figure 4: Decoupled time-frequency codes visualization on sleep staging dataset (ISRUC_S3).

### 4 CONCLUSION

In this paper, we present *CodeBrain*, an EEG foundation model that unifies interpretable tokenization with a brain-inspired multi-scale architecture. The TFDUAL-TOKENIZER decouples heterogeneous EEG signals, expanding the representation space while suggesting domain-specific representation-level interpretability, and the EEGSSM architecture integrates structured global convolution with sliding-window attention to efficiently capture both long-range and local dependencies. Pretrained on the large-scale TUEG corpus, *CodeBrain* demonstrates strong generalization across eight tasks and ten datasets under distribution shifts, with comprehensive ablations, scaling-law analyses confirming robustness and scalability. These results establish *CodeBrain* as a strong foundation for neural time-series representation learning. Limitations and future work are discussed in Appendix P.

## ACKNOWLEDGMENT

This research is partially supported by the Singapore National Medical Research Council (NMRC) research grant (MOH-001838); the AI for Public Health Program (25-1447-A0001) in the Saw Swee Hock School of Public Health, National University of Singapore; the Cisco-NUS Accelerated Digital Economy Corporate Laboratory (Award I21001E0002), funded by A*STAR, CISCO Systems (USA) Pte. Ltd., and the National University of Singapore; the National University of Singapore President's Graduate Fellowship; the NUS-Guangzhou Research Translation and Innovation Institute Scholarship; the Youth Science Fund Project of National Natural Science Foundation of China (No.62306317); the Beijing Nova Program (Grant No. 20250484804); and the Young Elite Scientists Sponsorship Program of the Beijing High Innovation Plan (No20250912).

## ETHICS STATEMENT

This research adheres to the ICLR Code of Ethics and Responsible Research Guidelines. We confirm that all aspects of the study conform to these guidelines. The work preserves anonymity requirements and does not deviate from the ethical principles set by ICLR. This paper does not involve human subjects, crowdsourcing, or sensitive user data. No Institutional Review Board (IRB) approval was required. The work does not release models or datasets with a high risk of misuse, and it poses no privacy, security, or legal compliance concerns. Furthermore, it does not contain potentially harmful insights, discrimination or bias issues, or conflicts of interest. All experiments are conducted on publicly available EEG datasets, following research integrity and responsible research practices as required by the ICLR Code of Ethics. In addition, disclosures regarding the use of LLM are in the Appendix Q.

## REPRODUCIBILITY STATEMENT

We have made extensive efforts to ensure the reproducibility of our work. The complete implementation and the pretrained weights are provided in `https://github.com/jingyingma01/CodeBrain`. For theoretical results, all assumptions and complete proofs of the claims are included in Appendix D. For experimental reproducibility, we provide a detailed description of the datasets, preprocessing procedures, and evaluation protocols in Section G, as well as comprehensive hyperparameter settings in Appendix J. These materials together ensure that both the theoretical and empirical results can be independently verified.

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

# Appendix Contents

## A  RELATED WORK

**EEG Foundation Models.**   Inspired by foundation models in vision and language domains (Wang et al., 2023; Achiam et al., 2023), EEG research is shifting from task-specific models (Ding et al., 2024; Wang et al., 2024b; Chen et al., 2025b; Liu et al., 2024a) to EFMs for learning expressive representations (Chen et al., 2025c; Liu et al., 2025). Current EFMs can be divided into two categories. 1) Contrastive learning-based (CL) (Chen et al., 2025a): BENDER (Kostas et al., 2021) first showed the ability of CL for EEG representations. Then, the Brant series (Zhang et al., 2023; Yuan et al., 2024; Zhang et al., 2024) enables joint representation learning across physiological signals using CL. 2) Reconstruction-based: BIOT (Yang et al., 2023) pioneers cross-modal pretraining for biosignals, including EEG. Subsequent models focus specifically on EEG, learning representations by predicting masked discrete tokens (Jiang et al., 2024; 2025) or reconstructing raw signals (Wang et al., 2024a; Mohammadi Foumani et al., 2024; Wang et al., 2025). Most existing EFMs adopt a Transformer architecture, which is suboptimal for EEG due to poor handling of sparse dependencies, quadratic complexity, and the ignorance of local dependencies by treating each patch as a token.

**EEG Tokenization.**   Tokenization has been key in Natural Language Processing (NLP) for generating generalizable and interpretable input representations (Sennrich et al., 2016; Kudo & Richardson, 2018). Inspired by this, early EFMs used patch-based continuous tokenization (Yang et al., 2023; Yuan et al., 2024) to handle EEG noise and variability, but without quantization, leading to unbounded and less interpretable representations. LaBraM (Jiang et al., 2024) introduced vector quantization to learn discrete EEG tokens, following the VQ-VAE design from vision tasks (Van Den Oord et al., 2017). However, this direct transfer overlooks EEG's heterogeneous structure, limiting representation capacity. Moreover, the tokenizer of LaBraM is trained only on frequency-domain reconstruction due to convergence issues with raw signals reconstruction. Later efforts (Pradeepkumar et al., 2025; Jiang et al., 2025; Guo et al., 2025) adopt a frequency-dominant paradigm, where some methods employ a single codebook for frequency modeling, limiting the representation space and interpretability.

**State Space Model.**   Recent works have focused on enhancing classic State-Space Models (SSM) to more efficiently model sequential data using deep learning. For example, Rangapuram et al. (2018) used recurrent neural networks to learn parameters in SSM. However, the most significant progress came from Gu et al. (2022b) with the introduction of the structured state space model (S4), which reduces the computational complexity of SSM in modeling long sequences using a special state transition matrix (Gu et al., 2020). These low-rank and normal matrices enable SSM to compute global convolution kernels efficiently through fast Fourier transform across the entire sequence. Subsequently, some works have further improved the shortcomings of S4 in areas such as model architecture (Smith et al., 2023) and convolution (Raghu et al., 2023), and have started applying it to tasks such as natural language processing (Dao et al., 2022) and time series analysis (Zhou et al., 2023). Recently, some works have also been applied to the EEG data, Tran et al. (Tran et al., 2024) leveraging SSMs to detect dementia. They extract temporal information from EEG signals through the Mamba architecture and combine it with frequency domain features to better manage the complexity of multivariate EEG. In the work of Gui et al. (2024), SSM has also become the backbone network of the EEG foundation model. This further highlights the fast reasoning speed and efficient memory usage of the SSM model when processing EEG signals.

## B  INTERPRETABILITY ANALYSES OF TFDUAL-TOKENIZER REPRESENTATIONS

We conduct representation-level interpretability analyses of the decoupled TFDUAL-TOKENIZER. As case studies, we visualize selected tokens alongside well-established physiological patterns, illustrating that frequency codes reflect spectral rhythms and temporal codes align with characteristic waveform events. These qualitative examples show that the tokenizer does not discretize signals arbitrarily but organizes them into domain-relevant structures. We further complement these visualizations with a quantitative analysis of class-specific token usage across four datasets, demonstrating that the learned codebooks induce structured representation patterns. Such interpretability may help facilitate clinical understanding and trust (Ma et al., 2025b; Xing et al., 2025).

## B.1 EXPLORING FREQUENCY TOKEN PATTERNS IN RELATION TO SPECTRAL RHYTHMS

To illustrate the interpretable structure of the learned frequency tokens, we take sleep staging as an example downstream dataset. We focus on N2 and N3 stages, since they are characterized by the most distinctive spectral rhythms in clinical sleep scoring: *spindle* in the sigma band (11-16 Hz) for N2, and *slow wave* in the delta band (0.5-4 Hz) for N3. Using token-activation statistics from the ISRUC_S3 test set, we select the most frequently activated class-specific frequency tokens.

As shown in Figure 5, frequency code No. 2298 tends to capture a sigma bump, with prominent peaks localized in the spindle range. This is similar to N2 spindles (Berry et al., 2012). Quantitatively, this code exhibited a clear sigma peak in approximately $15.4\%$ of its assigned patches. Similarly, Figure 6 shows that frequency code No. 32 predominantly encodes delta dominance, a typical frequency feature of N3 sleep. This code displayed clear delta dominance in about $18.3\%$ of its assigned segments. Taken together, these findings suggest that our frequency-branch tokenizer helps establish a frequency vocabulary for EEG.

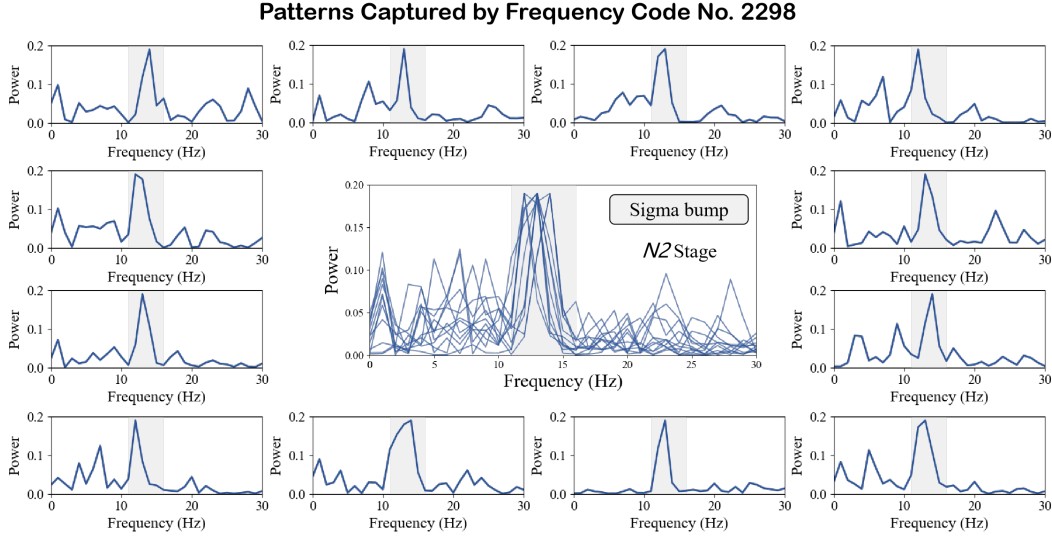

Figure 5: Frequency code capturing sigma bump, a typical spectral rhythm of N2 stage.

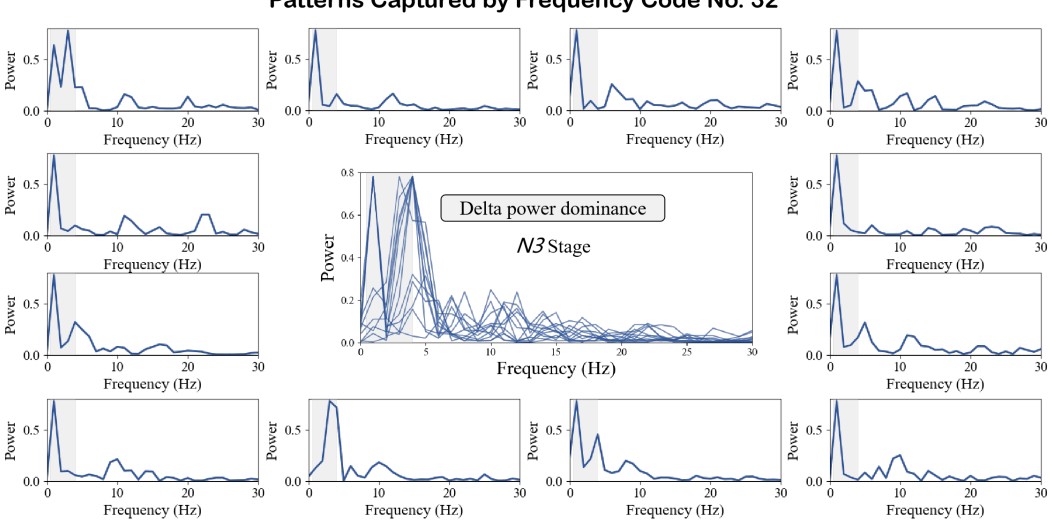

Figure 6: Frequency code capturing delta dominance, a typical spectral rhythm of N3 stage.

## B.2 EXPLORING TEMPORAL TOKEN PATTERNS IN RELATION TO NEURAL EVENTS

In addition to spectral rhythms, we applied the same token-activation-based selection procedure to the temporal codes on ISRUC_S3 to examine whether the tokenizer captures coherent and clinically recognizable structures in the raw EEG waveforms. Figure 8 and Figure 7 show that certain codes align with K-complexes and sleep spindles, the hallmark waveforms of N2 sleep. Quantitatively, $\approx 5.34\%$ of the assigned patches contained K-complexes and $\approx 7.01\%$ contained spindles. These rates are consistent with clinical expectations (1-2 K-complexes and 2-5 spindles per minute in N2 Berry et al. (2012)) and, when considering the overall prevalence of N2 in five-class sleep staging, still substantially exceed what would be expected from random token usage.

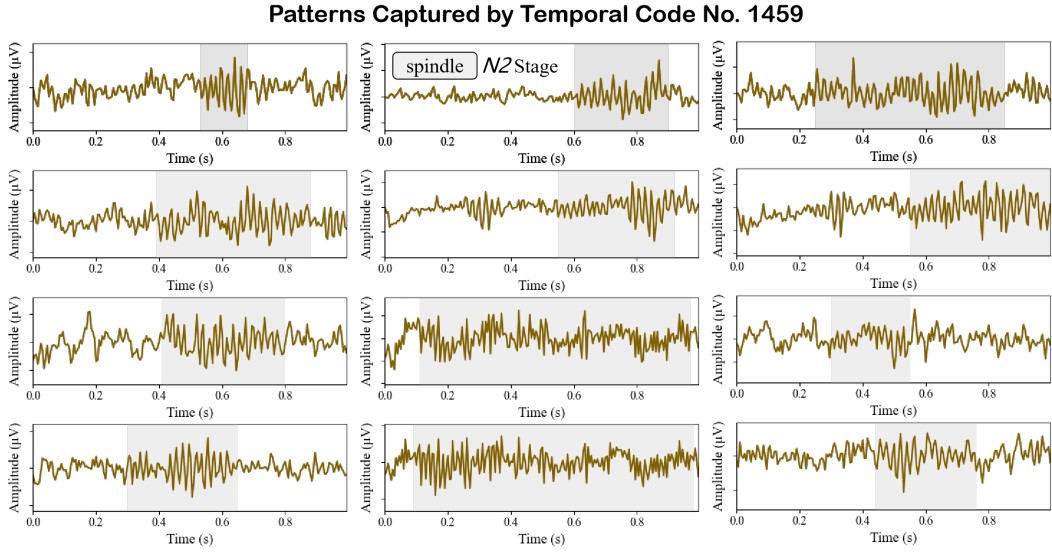

Figure 7: Temporal code capturing sleep spindle, a typical neural events of N2 stage.

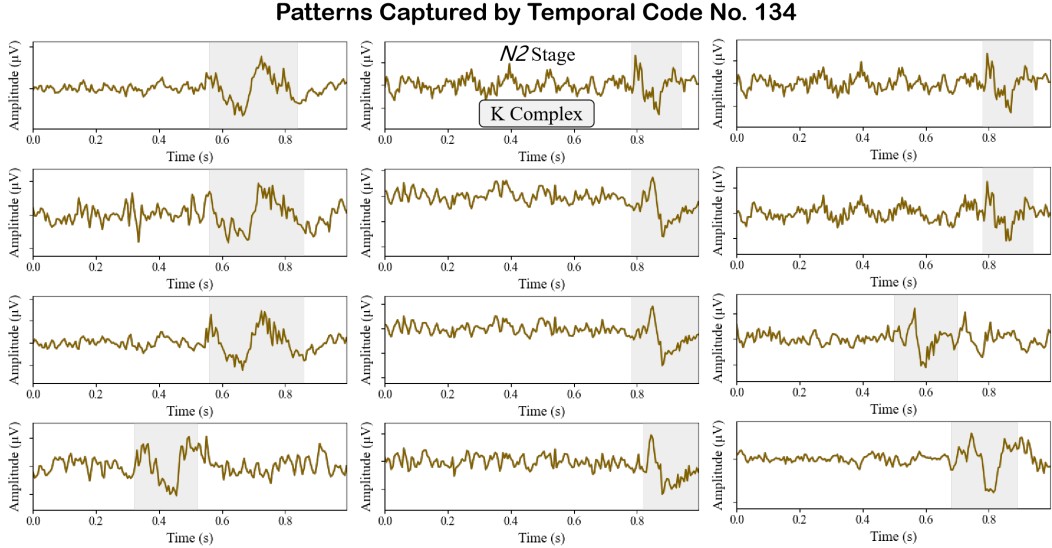

Figure 8: Temporal code capturing K complex, a typical neural events of N2 stage.

Figure 9 shows that temporal code No. 1537 corresponds to slow waves, the defining feature of N3 sleep, occurring in $\approx 40.4\%$ of its assigned patches. This exceeds the $20\%$ per-epoch criterion for N3 scoring Berry et al. (2012) and, given the overall prevalence of N3, indicates that this token provides a meaningful link to neural events.

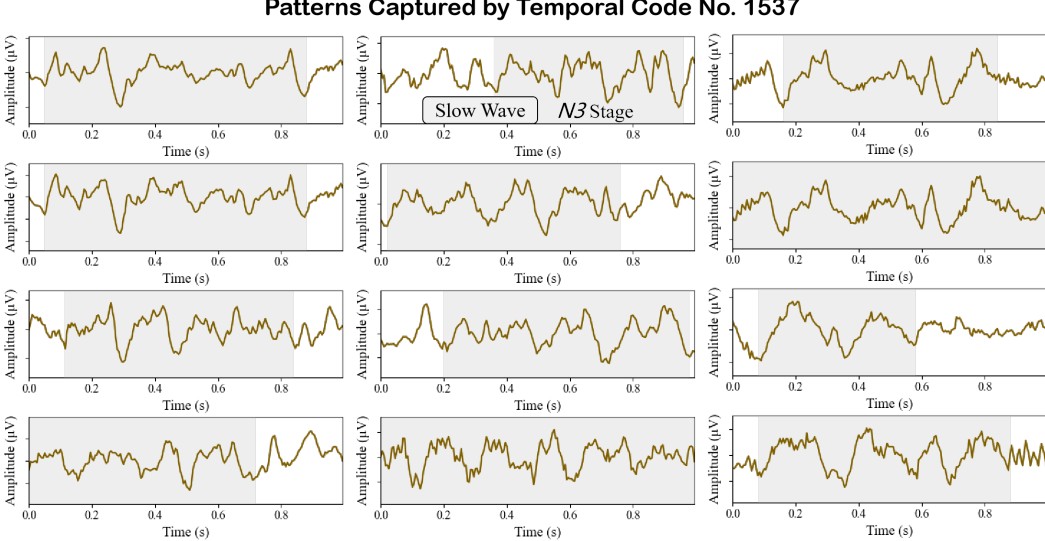

Figure 9: Temporal code capturing slow wave, a typical neural events of N3 stage.

Taken together, these observations suggest that the temporal branch of our tokenizer contributes to establishing a vocabulary of neurophysiological events, complementing the frequency-domain findings. While temporal waveforms are often noisier and harder to model than spectral rhythms (Jiang et al., 2024), our decoupled design allows temporal tokens to emerge with meaningful associations to critical neural events. This indicates that the temporal branch offers useful insights to clinically relevant waveforms and demonstrates the effectiveness of our approach in capturing complementary structure.

### B.3 CLASS-SPECIFIC TOKEN RATIO ANALYSIS FOR TFDUAL-TOKENIZER

To further characterize the representation structure learned by the TFDUAL-TOKENIZER, we analyze the distribution of token usage across classes on unseen downstream datasets. This analysis aims to examine whether the decoupled tokenizer induces more structured and class-consistent token activation patterns, which would support our motivation for separating temporal and frequency domains at the representation level.

A token (code) is considered class-specific if it predominantly appears in samples of a single class. Formally, for a given token $c$, let $N_c^{(y)}$ denote the number of times $c$ appears in class $y$. The dominance ratio is defined as:

$$\text{Dominance}(c) = \frac{\max_y N_c^{(y)}}{\sum_y N_c^{(y)}}. \tag{17}$$

If $\text{Dominance}(c) \geq \tau$ (we use $\tau = 1$), the token is deemed class-specific. The class-specific ratio for a codebook is then computed as:

$$\text{Class-Specific Token Ratio} = \frac{\text{\# class-specific tokens}}{\text{Total tokens in codebook}}. \tag{18}$$

Figure 10 illustrates the proportion of class-specific tokens derived from three configurations: using only the temporal codebook, only the frequency codebook, and a combination of both (TF-Decoupled). We employ two independent codebooks to capture complementary information via the proposed TFDUAL-TOKENIZER module.

Across all four datasets, including two for emotion recognition (FACED and SEED-V) and two for sleep staging (ISRUC_S3 and ISRUC_S1), the decoupled codebook consistently achieves the highest class-specific token ratios, reaching 54.7% on ISRUC_S3 and 46.4% on FACED. These

results confirm that decoupling temporal and frequency domain information significantly enhances the model's ability to capture structured representation.

Figure 10: Class-specific code ratio across different codebooks.

### B.4 Additional Ablations on Codebook Contribution

To further examine the role of the tokenizer in pretraining, we provide additional exploratory evidence on how the the codebooks influence representation learning. These analyses extend the comparisons in Table 3.3, where the decoupled design outperforms temporal-only, frequency-only, and mixed codebooks. Specifically, we compare the following two ablations on the ISRUC_S3 dataset:

- **Raw-signal reconstruction.** We remove the tokenizer entirely and train the EEGSSM backbone directly to reconstruct raw waveforms, thereby eliminating the discretization step.

- **Masked codebook.** We identify the top 50% most frequently activated tokens across both codebooks on ISRUC_S3. During pretraining, whenever a segment is assigned one of these tokens, we replace it with a placeholder, effectively masking half of the vocabulary and preventing the model from relying on these high-activation codes.

Table 4 reports downstream performance on ISRUC_S3 under these settings.

Table 4: Ablations of the codebook contribution on ISRUC_S3.

| Setting | Cohen's $\kappa$ | Weighted F1 | Balanced Acc |
|---|---|---|---|
| TFDual_tokenizer (ours) | **0.7671** $\pm$ 0.0091 | **0.8202** $\pm$ 0.0071 | **0.7856** $\pm$ 0.0031 |
| Raw-signal Reconstruction | 0.7503 $\pm$ 0.0087 | 0.8014 $\pm$ 0.0079 | 0.7763 $\pm$ 0.0048 |
| Masked Codebook | 0.7426 $\pm$ 0.0102 | 0.7931 $\pm$ 0.0084 | 0.7690 $\pm$ 0.0063 |

These experiments do not suggest that specific interpretable codes directly determine task outcomes. However, they provide supporting evidence that the decoupled tokenizer imparts useful structure during pretraining, and that entirely removing or partially disabling the codebooks leads to consistently degraded downstream performance. These observations reinforce the contribution of the tokenizer at the representation level.

## C  Structure State Space Model

The state-space model is a classic model in control theory, and it represents the operational state of a system using first-order differential equations (ODE). A continuous state-space model can be defined

in the following form:

$$x'(t) = Ax(t) + Bu(t), y(t) = Cx'(t) + Du(t), \quad (19)$$

where $u(t)$ is a vector that represents the input of the system, while $y(t)$ is a vector that represents the output of the system. $x(t)$ and its derivative $x'(t)$ represent the latent states of the system, typically in the form of an N-D vector. And $A, B, C, D$ here are the state, input, output, and feedforward matrices, defining the relationship between the input, output, and state vector. Following Gu et al. (Gu et al., 2022b), $\bar{D}$ is set equal to 0 since it can be replaced by the residual connection. Now, Equation 20 resembles an architecture similar to RNN, allowing us to recurrently compute $x_k$. Let the initial state be $x_{k-1} = 0$, and we can unroll Eq 20 as follows:

$$y_k = \overline{CA^k B} u_0 + \overline{CA^{k-1} B} u_1 + ... + \overline{CAB} u_{k-1} + \overline{CB} u_k, \quad (20)$$

$$y = \overline{K} u, \quad \overline{K} = (\overline{CB}, \overline{CA^1 B}, ..., \overline{CA^k B}). \quad (21)$$

Therefore, SSM can be transformed from the form of a recurrent neural network to a convolutional neural network. During training, the $\overline{K}$ can be considered as a 1-D globe convolution kernel, so the $y$ can be calculated via the "long" convolution, allowing us to use the fast Fourier transform to efficiently compute the SSM convolutional kernel $\overline{K}$. However, directly computing the convolution in Equation 21 can be very expensive for long sequences. We can use the Fast Fourier Transform to accelerate it. The form of this convolution can be written as:

$$y = F_N^{-1} D_k F_N u, D_k = \text{diag}(\overline{K} F_N), \quad (22)$$

where $F_N$ denotes the DFT matrix of size $N$. This FFT convolution has a computational complexity of $O(nlog(n))$. Following (Li et al., 2022; Fu et al., 2023), we hope to parameterize $K$ directly rather than through $\{A, B, C\}$ because we can eliminate complex parameterization and accelerate the entire convolution.

While Eq. 22 shows that the SSM can be computed using FFT-based convolution, it is important to formalize the complexity guarantee. We now state the following proposition.

**Proposition 1.** *Let $(A, B, C, D)$ denote the discretized state-space matrices of an S4 layer, with input sequence $u \in \mathbb{R}^N$. The output $y \in \mathbb{R}^N$ can be written as a linear convolution $y = k * u$ with kernel*

$$k_0 = D, \quad k_n = CA^{n-1} B, \quad n \geq 1.$$

*Using the convolution theorem and the FFT, this convolution can be computed in time $\Theta(N \log N)$.*

*Proof.* This proof follows the proof in *Lemma C.2* of Gu et al. (2022b):

Expanding the recurrence

$$x_{n+1} = Ax_n + Bu_n, \qquad y_n = Cx_n + Du_n, \quad (23)$$

yields

$$y_n = Du_n + \sum_{t=0}^{n-1} CA^{n-1-t} B\, u_t = \sum_{t=0}^{n} k_{n-t}\, u_t, \quad (24)$$

where $k_0 = D$ and $k_n = CA^{n-1} B$. Thus $y = k * u$.

Let $\mathcal{F}_N$ denote the $N$-point DFT. By the convolution theorem,

$$\mathcal{F}_N(y) = \mathcal{F}_N(k) \odot \mathcal{F}_N(u), \quad (25)$$

so that

$$y = \mathcal{F}_N^{-1}(\mathcal{F}_N(k) \odot \mathcal{F}_N(u)). \quad (26)$$

The cost of forward FFT, element-wise product, and inverse FFT is $\Theta(N \log N)$.

The S4 parameterization ensures $A$ admits a diagonal-plus-low-rank (DPLR) structure, so the kernel takes the form

$$k_n = \sum_{j=1}^{r} \alpha_j \lambda_j^{n-1}, \qquad r \ll N. \quad (27)$$

Each exponential sequence can be generated recursively in $\mathcal{O}(N)$, yielding all $k_0, \ldots, k_{N-1}$ in linear time.

Combining the above,

$$T(N) = \underbrace{\Theta(N \log N)}_{\text{FFT convolution}} + \underbrace{\mathcal{O}(N)}_{\text{kernel generation}} = \Theta(N \log N). \quad (28)$$

## D   THEORETICAL ANALYSIS OF DECOUPLED CODEBOOK TRAINING

**Proposition 2.1.** *Let $X = (X_t, X_f)$ denote the temporal and frequency representations of an EEG segment, assumed approximately independent. Consider an additive reconstruction distortion $d(x, \hat{x}) = d_t(x_t, \hat{x}_t) + d_f(x_f, \hat{x}_f)$. Under a fixed total codebook size $K = 2^R$, a product codebook $\mathcal{C}_t \times \mathcal{C}_f$ with $|\mathcal{C}_t| = 2^{R_t}$ and $|\mathcal{C}_f| = 2^{R_f}$, $R_t + R_f = R$, achieves a minimum expected distortion*

$$D^\star_{prod}(R) = \min_{R_t + R_f = R} \left( D^\star_t(R_t) + D^\star_f(R_f) \right), \tag{29}$$

*which satisfies*

$$D^\star_{mix}(R) \geq D^\star_{prod}(R), \tag{30}$$

*where $D^\star_{mix}(R)$ is the minimum distortion of a single mixed codebook $\mathcal{C}_{mix} \subset \mathbb{R}^{d_t + d_f}$ of size $2^R$.*

*Proof.* The argument combines the separability of the rate-distortion (R-D) function for independent sources under additive distortion and the high-rate quantization approximation to the Shannon R-D limit.

Let $X = (X_t, X_f)$ with $X_t \perp X_f$, and distortion

$$d(x, \hat{x}) = d_t(x_t, \hat{x}_t) + d_f(x_f, \hat{x}_f). \tag{31}$$

Then the Shannon R-D function satisfies

$$R(D) = \min_{D_t + D_f = D} \left( R_t(D_t) + R_f(D_f) \right), \tag{32}$$

where $R_t(\cdot)$ and $R_f(\cdot)$ are the marginal R-D functions for $X_t$ and $X_f$. By convex duality, the optimal test channel factorizes:

$$p(\hat{x}_t, \hat{x}_f \mid x_t, x_f) = p(\hat{x}_t \mid x_t)\, p(\hat{x}_f \mid x_f), \tag{33}$$

and the optimal distortion allocation solves

$$\min_{D_t, D_f} R_t(D_t) + R_f(D_f) \quad \text{s.t. } D_t + D_f = D, \tag{34}$$

with KKT condition $R'_t(D^\star_t) = R'_f(D^\star_f)$.

At rate $R_t$ bits for $X_t$ and $R_f$ bits for $X_f$ (with $R_t + R_f = R$), the minimal achievable distortions are $D^\star_t(R_t)$ and $D^\star_f(R_f)$. In the high-rate regime, practical vector quantizers approach these Shannon limits, yielding

$$D_{\text{prod}}(R_t, R_f) \approx D^\star_t(R_t) + D^\star_f(R_f). \tag{35}$$

Optimizing over all feasible splits gives

$$D^\star_{\text{prod}}(R) = \min_{R_t + R_f = R} \left( D^\star_t(R_t) + D^\star_f(R_f) \right). \tag{36}$$

Any mixed codebook $\mathcal{C}_{\text{mix}} \subset \mathbb{R}^{d_t + d_f}$ with $|\mathcal{C}_{\text{mix}}| = 2^R$ cannot beat the Shannon R-D limit, so

$$D^\star_{\text{mix}}(R) \geq D^\star(R). \tag{37}$$

But from (34)- (35),

$$D^\star(R) = \min_{R_t + R_f = R} \left( D^\star_t(R_t) + D^\star_f(R_f) \right) = D^\star_{\text{prod}}(R). \tag{38}$$

Therefore,

$$D^\star_{\text{mix}}(R) \geq D^\star_{\text{prod}}(R). \tag{39}$$

*Remark.* We acknowledge that theoretically, $X_t$ and $X_f$ are coupled via the Fourier Transform. However, within the context of Neural Vector Quantization, they behave as heterogeneous sources of information. For example, topological features in waveforms (e.g., K-complexes) and frequency densities in frequency (e.g., Alpha rhythms) impose orthogonal constraints on the codebook optimization landscape. Therefore, the "approximate independence" in Proposition 2.1 should be interpreted as the functional independence of semantic distortions in the latent representation space, rather than the statistical independence of the raw signals.

# E    PRETRAINING RESULTS

Our model follows a two-stage pretraining framework. In the first stage, we train the TFDUAL-TOKENIZER, which independently tokenizes EEG signals in both the temporal and frequency domains. This tokenizer is optimized to reconstruct the original raw EEG signals, amplitude, and phase components, thereby producing discrete code representations with structural interpretability. In the second stage, we pretrain the EEGSSM encoder using a masked modeling objective: given the original EEG signals as input, the model learns to predict the corresponding masked tokens generated from the TFDUAL-TOKENIZER.

This section reports the pretraining results of both stages, including loss convergence, reconstruction dynamics, and codebook utilization patterns.

## E.1    TFDUAL-TOKENIZER PRETRAINING RESULTS

**Total Training Loss**    The total pretraining loss curve of the TFDUAL-TOKENIZER is shown in Figure 11. The model demonstrates a rapid initial decrease in loss during the first few epochs, followed by a slower but consistent decline.

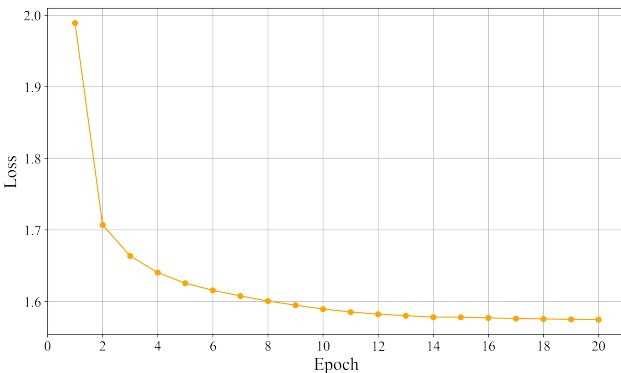

Figure 11: Pretraining loss curve of TFDUAL-TOKENIZER.

**Reconstruction Loss.**    We report the pretraining reconstruction loss of the TFDUAL-TOKENIZER in Figure 12. The temporal codebook is trained to reconstruct raw EEG signals in the time domain, while the frequency codebook is trained to reconstruct the corresponding amplitude and phase components in the frequency domain. All three loss curves exhibit a sharp initial decline, followed by a gradual convergence, indicating stable optimization.

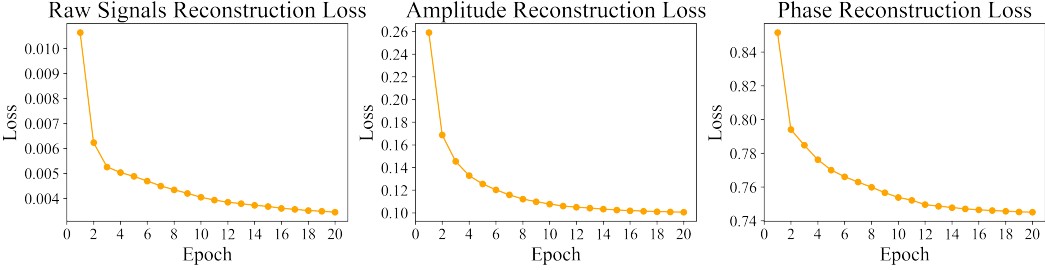

Figure 12: Pretraining loss curve of TFDUAL-TOKENIZER.

**Unused Codes Analysis.**    During pretraining, we track the number of unused codes in both the temporal and frequency codebooks of the TFDUAL-TOKENIZER, each with a size of 4096. As shown in Figure 13, the frequency codebook demonstrates a rapid decrease in unused codes, while the

temporal codebook shows a slower and more incremental reduction. A more detailed analysis of the temporal-frequency complementarity is provided in Section B.3.

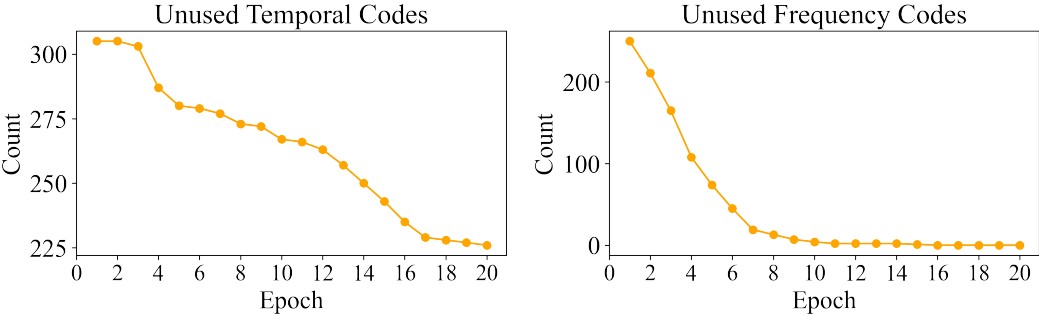

Figure 13: Unused code dynamics of the TFDUAL-TOKENIZER.

## E.2 EEGSSSM PRETRAINING RESULTS

We plot the pretraining loss curve of EEGSSM in Figure 14. We select epoch 10 as the checkpoint for downstream fine-tuning. We observe that the pretraining loss of EEGSSM decreases rapidly from epoch 1 to 6 (9.04 → 6.39), then flattens gradually after epoch 10 (6.01 → 5.66). Using epoch 10 for fine-tuning is a balance between representation strength and generalization. Overtraining on EEG, prone to noise and inter-subject variability, can reduce transferability. Epoch 10 serves as a conservative yet effective checkpoint. This practice is consistent with trends in foundation models from NLP (Devlin et al., 2019) and vision (Caron et al., 2021), where mid-training checkpoints often lead to better downstream performance than final ones due to reduced overfitting to the pretext task.

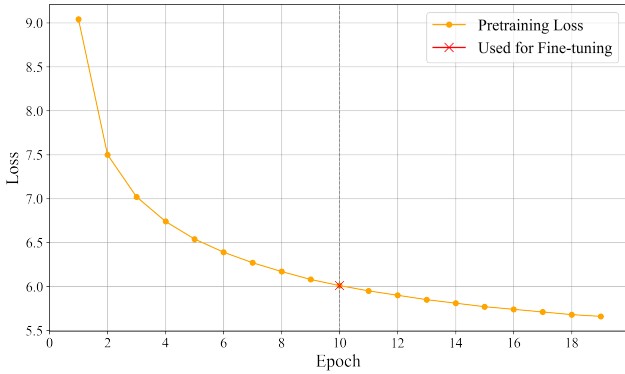

Figure 14: Pretraining loss curve of EEGSSM.

## F IMPROVING TEMPORAL CODEBOOK LEARNING VIA CONTRASTIVE LOSS

To improve the learning of the temporal codebook in our TFDUAL-TOKENIZER, we introduce a contrastive loss as one of the objectives during pretraining. This design is motivated by observations from prior work LaBraM (Jiang et al., 2024), where the authors report that reconstructing raw EEG signals leads to unconvergence, and thus omit the temporal reconstruction objective entirely.

To better understand this limitation, we first implemented a baseline reconstruction of raw signals within the LaBraM framework and observed that the training loss plateaued at a high value (between 0.128 and 0.131), showing no convergence over training epochs. To mitigate this, we introduce a *TF-Conv* module before the Transformer encoder, designed to extract temporal-frequency representations before tokenization. While this stabilizes training to some extent, we still observe significant issues with code utilization and loss convergence. Therefore, we incorporate a lightweight contrastive loss,

applied over temporal representations before quantization, to encourage the model to organize similar input patterns closer in the latent space. As shown in Figure 15, this improves the optimization of the reconstruction loss and reduces the number of unused temporal codes during training.

These results demonstrate that contrastive regularization acts as an effective prior for stabilizing discrete token learning, particularly when reconstructing raw signals. It both improves convergence and mitigates codebook collapse in the temporal branch of the tokenizer.

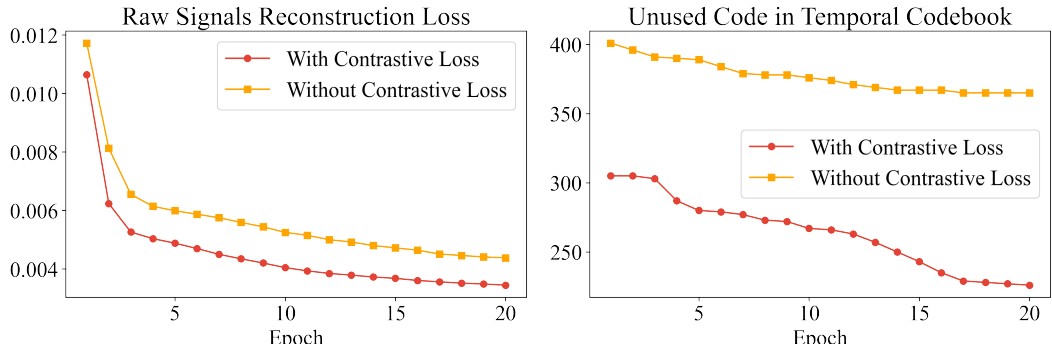

Figure 15: Effect of contrastive loss on temporal codebook learning.

# G  DATASET DESCRIPTION

We evaluate the *CodeBrain* across eight diverse downstream tasks covering ten publicly available EEG datasets. Notably, these datasets exhibit substantial variability in channel configurations (ranging from 6 to 64), sequence lengths (from 1s to 30s), and task complexities (2-class to 9-class), highlighting the versatility and robustness of *CodeBrain* across different EEG applications. The following sections describe each dataset in detail, including its task objective and data split strategy. A comprehensive analysis of each dataset is provided below.

**Emotion Recognition.**  We conduct emotion recognition experiments on two widely used EEG datasets: FACED (Chen et al., 2023) and SEED-V (Liu et al., 2021a).
The **FACED** dataset (Finer-Grained Affective Computing EEG Dataset) is a large-scale EEG dataset for emotion recognition tasks (Chen et al., 2023). It consists of 32-channel EEG recordings sampled at 250 Hz from 123 participants, each exposed to 28 video clips designed to elicit nine distinct emotional states: *amusement*, *inspiration*, *joy*, *tenderness*, *anger*, *fear*, *disgust*, *sadness*, and *neutral*. These cover both positive and negative affective categories. Each EEG trial is 10 seconds long and is subsequently resampled to 200 Hz, resulting in a total of 10,332 clean EEG segments. For fair comparison, we adopt the same subject-wise split as in (Wang et al., 2025): subjects 1-80 are used for training, 81-100 for validation, and 101-123 for testing, ensuring no subject overlap across splits and enabling evaluation of cross-subject generalization.
**SEED-V** (Liu et al., 2021a) is an EEG dataset designed for emotion recognition, covering five emotional categories: *happy*, *sad*, *neutral*, *disgust*, and *fear*. It consists of 62-channel EEG recordings collected at 1000 Hz from 16 subjects, each participating in three sessions. Each session includes 15 trials, which are evenly divided into training, validation, and test sets (5 trials each). The EEG signals are segmented into 1-second windows, yielding a total of 117,744 samples, and resampled to 200 Hz.

**Sleep Staging.**  We use two datasets, **ISRUC_S1** and **ISRUC_S3** (Khalighi et al., 2016), for sleep stage classification. Both datasets are annotated according to the American Academy of Sleep Medicine (AASM) standard (Berry et al., 2012), with five sleep stages: *Wake*, *NREM1 (N1)*, *NREM (N2)*, *NREM (N3)*, and *REM*. Each EEG segment corresponds to a 30-second epoch.
**ISRUC_S1** includes EEG recordings from 100 subjects using six channels at a sampling rate of 200 Hz. We adopt a subject-wise split, with 80 subjects for training, 10 for validation, and 10 for testing. As the transition rules between sleep stages carry important temporal patterns, we follow prior work (Wang et al., 2024a; 2025; Zhou et al., 2025b) and insert a Transformer layer on top of the prediction head during fine-tuning to better capture sequence-level dependencies. We set the input sequence

length to 20, and discard segments that cannot be evenly divided. In total, 86,320 labeled samples are retained.

**ISRUC_S3** is a smaller dataset comprising recordings from 10 subjects, also sampled at 200 Hz with six channels, totaling 8,500 labeled segments. We follow an 8:1:1 subject-wise split for training, validation, and testing.

**Imagined Speech Classification.** The **BCIC2020-T3** dataset (Jeong et al., 2022) was released as part of the 2020 International Brain-Computer Interface Competition and focuses on imagined speech decoding. It contains EEG recordings from 15 participants who were instructed to silently imagine speaking five specific words or phrases, *"hello"*, *"help me"*, *"stop"*, *"thank you"*, and *"yes"*. EEG signals were collected using 64 scalp channels at a sampling rate of 256 Hz and were subsequently resampled to 200 Hz for preprocessing consistency. Each subject completed 80 trials per class, resulting in a total of 6,000 trials. The dataset provides predefined training, validation, and test splits, with 60, 10, and 10 trials per class, respectively, facilitating fair model evaluation like existing baselines (Wang et al., 2025).

**Mental Stress Detection.** The **Mental Arithmetic** dataset (Mumtaz, 2016) supports the task of mental stress detection using EEG signals. It contains recordings from 36 subjects under two distinct cognitive conditions: *resting* and *active engagement* in mental arithmetic. EEG data labeled as "no stress" correspond to resting periods prior to the task, while "stress" labels are assigned to recordings during task performance. The signals were acquired using 20 electrodes placed according to the international 10-20 system, with an original sampling rate of 500 Hz. For consistency, the signals are resampled to 200 Hz and band-pass filtered between 0.5-45 Hz to suppress noise. Each recording is segmented into 5-second windows, yielding a total of 1,707 samples. We adopt a subject-wise split for fair evaluation with existing baselines (Wang et al., 2025): subjects 1-28 for training, 29-32 for validation, and 33-36 for testing.

**Seizure Detection.** The **CHB-MIT** dataset (Shoeb, 2009) is a widely used benchmark for seizure detection from EEG signals. It contains long-term EEG recordings from 24 patients diagnosed with intractable epilepsy, collected at the Children's Hospital Boston. The subjects underwent continuous monitoring over several days, during which seizures were recorded following the tapering of anti-epileptic medications. EEG signals were acquired using the international 10-20 system and originally sampled at 256 Hz. In our setting, we adopt 16 channels commonly used in prior work (Yang et al., 2023; Wang et al., 2025), resample all signals to 200 Hz, and segment them into 10-second non-overlapping windows, yielding 326,993 labeled samples across seizure and non-seizure classes. We follow a subject-wise split: subjects 1-20 for training (excluding subjects with insufficient recordings), 21-22 for validation, and 23-24 for testing. Notably, this dataset is highly imbalanced, with seizure events constituting only a small fraction of the total samples, posing significant challenges for model training and evaluation.

**Motor Imagery Classification.** The **SHU-MI** dataset (Goldberger et al., 2000) is designed for binary motor imagery classification, where participants are instructed to imagine movements of either the *left or right hand*. EEG signals were recorded from 25 subjects using a 32-channel setup at an original sampling rate of 250 Hz. To ensure consistency with the pre-training setting, all signals are resampled to 200 Hz and segmented into 4-second non-overlapping windows, resulting in 11,988 labeled samples. A subject-wise split is applied for fair model evaluation like existing baselines (Wang et al., 2025), with subjects 1-15 used for training, 16-20 for validation, and 21-25 for testing. This dataset supports the development of BCI systems that decode motor intentions from brain activity without actual movement.

**Event Type Classification.** The **TUEV** dataset (Obeid & Picone, 2016) is a clinically annotated EEG corpus used for multi-class event type classification. It includes six event categories: *spike and sharp wave (SPSW)*, *generalized periodic epileptiform discharges (GPED)*, *periodic lateralized epileptiform discharges (PLED)*, *eye movements (EYEM)*, *artifacts (ARTF)*, and *background activity (BCKG)*. EEG signals were originally recorded at 256 Hz using 23 channels. In line with prior work (Wang et al., 2025), we preprocess the data by selecting 16 bipolar montage channels based on the international 10-20 system. The signals are band-pass filtered between 0.3-75 Hz to suppress low- and high-frequency noise, and a 60 Hz notch filter is applied to eliminate power line interference. All

recordings are resampled to 200 Hz and segmented into 5-second windows, yielding 112,491 labeled samples. We follow the official train-test split and further divide the training subjects into training and validation sets in an 8:2 ratio, consistent with established benchmarks.

**Abnormal Detection.** The **TUAB** dataset (Obeid & Picone, 2016) is employed for binary abnormal EEG detection, where each EEG recording is labeled as either *normal* or *abnormal* based on clinical interpretation. Originally recorded at 256 Hz using 23 channels, the dataset provides large-scale EEG recordings suitable for evaluating diagnostic models. To ensure fair comparison with prior work (Wang et al., 2025), we follow a similar preprocessing protocol. Specifically, we select 16 bipolar montage channels following the international 10-20 system, apply band-pass filtering between 0.3-75 Hz to eliminate low- and high-frequency artifacts, and remove 60 Hz power line interference using a notch filter. The EEG signals are then resampled to 200 Hz and segmented into 10-second windows, resulting in 409,455 labeled samples. We follow the official train-test split and further divide the training set into training and validation subsets using an 8:2 subject-wise ratio, consistent with existing benchmarks.

## H  BASELINES AND METRICS DESCRIPTION

### H.1  METRICS

To comprehensively evaluate our model, we compare it with a set of strong baselines commonly used in EEG analysis. These baselines are evaluated using metrics tailored for class-imbalanced scenarios, which are prevalent in EEG datasets. The metrics include:

- **Balanced Accuracy**, which averages the recall across all classes and is particularly suitable for imbalanced multi-class classification tasks.
- **AUROC** and **PRAUC**, which assess the performance of binary classifiers under different thresholds. While AUROC measures the trade-off between sensitivity and specificity, PRAUC focuses on precision-recall trade-offs, especially informative under severe class imbalance.
- **Cohen's Kappa**, which quantifies inter-class agreement beyond chance and is employed as the primary metric for multi-class classification.
- **Weighted F1 Score**, which combines precision and recall while adjusting for class support, ensuring fair performance measurement across imbalanced datasets.

For model selection and comparison, AUROC is used as the main evaluation metric for binary classification tasks, and Cohen's Kappa is used for multi-class scenarios.

### H.2  BASELINES

We compare our *CodeBrain* model against a comprehensive set of baseline models that include widely used task-specific models, as well as publicly available EEG foundation models.

**EEGNet** (Lawhern et al., 2018). EEGNet is a compact convolutional neural network specifically designed for EEG-based BCI tasks. It adopts depthwise-separable convolutions to disentangle temporal filtering and spatial pattern learning, enabling efficient parameter usage while preserving discriminative EEG features.

**EEGConformer** (Song et al., 2022). EEGConformer integrates convolutional front-ends with Transformer blocks to jointly capture local temporal dynamics and longer-range dependencies. Its convolution modules extract short-term EEG patterns, while the attention mechanism models cross-channel and cross-time interactions.

**ContraWR** (Yang et al., 2021). ContraWR is a self-supervised representation learning framework that uses contrastive learning with a weakly-supervised relational task. By contrasting EEG segments from the same versus different contexts, the model learns invariant temporal representations without relying on explicit labels.

**ST-Transformer** (Song et al., 2021). ST-Transformer applies Transformer attention to EEG by factorizing spatial and temporal modeling. It processes EEG as a structured sequence across both

dimensions, where attention layers capture inter-channel relationships as well as time-varying dependencies.

We also compare *CodeBrain* against 5 publicly available EFMs that have released pre-trained weights, covering a diverse set of pretraining strategies to evaluate the effectiveness of different foundation model designs and pretraining paradigms under comparable settings.

**BENDR** (Kostas et al., 2021): We adopted **BENDR (Bert-inspired Neural Data Representations)** as our baseline model, as introduced by Kostas et al. BENDR is a pioneering deep learning architecture for Electroencephalography (EEG) data, leveraging transformers and a contrastive self-supervised learning task. This approach enables the model to learn meaningful representations from vast amounts of unlabeled EEG data.

**BIOT** (Yang et al., 2023): **BIOT (Biosignal Transformer for Cross-data Learning in the Wild)** is a transformer-based architecture designed to handle cross-dataset EEG signal classification under domain shifts. It leverages a domain-invariant attention mechanism and contrastive representation learning to enhance generalization across different recording conditions and subject populations.

**LaBraM**(Jiang et al., 2024): **LaBraM (Large Brain Model)** proposes a scalable transformer-based framework designed to learn generic EEG representations from large-scale brain signal datasets. By pretraining on a diverse corpus of EEG recordings, the model captures rich temporal and spatial features that transfer effectively to various downstream BCI tasks. The architecture incorporates efficient self-attention mechanisms and task-specific adapters to support flexible fine-tuning.

**EEGPT** (Wang et al., 2024a): **EEGPT** employs a dual self-supervised learning strategy that combines masked autoencoding with spatial-temporal representation alignment, enhancing feature quality by focusing on high signal-to-noise ratio (SNR) representations rather than raw signals. The model's hierarchical architecture decouples spatial and temporal processing, improving computational efficiency and adaptability to various brain-computer interface (BCI) applications.

**CBraMod** (Wang et al., 2025): **CBraMod (Criss-Cross Brain Foundation Model)** is a transformer-based EEG foundation model that addresses the heterogeneous spatial and temporal dependencies inherent in EEG signals. It introduces a criss-cross transformer architecture comprising parallel spatial and temporal attention mechanisms, enabling separate yet simultaneous modeling of spatial and temporal relationships.

# I ADDITIONAL EVALUATION ON OTHER BCI TASKS

We report the performance of *CodeBrain* on four additional EEG datasets not included in the main text, covering diverse domains of sleep staging, motor imagery, event detection, and abnormality classification in Tables 5 to 8. These allow us to assess the cross-domain generalization ability of our pretrained model beyond the main text.

We note that both TUAB and TUEV originate from the TUH EEG corpus (Obeid & Picone, 2016), which overlaps with our pretraining source (TUEG). To avoid overfitting to this distribution and promote generalization, we stop pretraining at epoch 10 as discussed in Section E.2. While this may limit gains on TUH datasets compared to previous EFM, such as CBraMod (trained for 40 epochs in the same pretraining dataset) (Wang et al., 2025), *CodeBrain* still achieves superior or competitive results.

**ISRUC_S1**    As shown in Table 5, *CodeBrain* achieves state-of-the-art performance on ISRUC_S1 in terms of Cohen's Kappa (0.7476) and Weighted F1 (0.8020), slightly surpassing CBraMod (Wang et al., 2025) by +0.34 and +0.09 points, respectively. Its Balanced Accuracy of 0.7835 is also competitive, trailing the best result by only -0.30. These results highlight the model's ability to capture temporal dependencies and learn discriminative representations for 5-class sleep staging under a cross-subject setting.

**SHU-MI**    As shown in Table 6, *CodeBrain* achieves the best overall performance on SHU-MI across all three metrics. It obtains an AUROC of 0.7124 and an PRAUC of 0.7166, slightly improving over the previous best by +1.36 and +0.27 points, respectively. For Balanced Accuracy, it reaches 0.6431

Table 5: Performance comparison on the ISURC_S1 (5-Class) dataset.

| Methods | Cohen's Kappa | Weighted F1 | Balanced Accuracy |
|---|---|---|---|
| EEGNet | $0.7040 \pm 0.0173$ | $0.7513 \pm 0.0124$ | $0.7154 \pm 0.0121$ |
| EEGConformer | $0.7143 \pm 0.0162$ | $0.7634 \pm 0.0151$ | $0.7400 \pm 0.0133$ |
| ContraWR | $0.7178 \pm 0.0156$ | $0.7610 \pm 0.0137$ | $0.7402 \pm 0.0126$ |
| ST-Transformer | $0.7013 \pm 0.0352$ | $0.7681 \pm 0.0175$ | $0.7381 \pm 0.0205$ |
| BENDRKostas et al. (2021) | $0.6956 \pm 0.0053$ | $0.7569 \pm 0.0049$ | $0.7401 \pm 0.0056$ |
| BIOTYang et al. (2023) | $0.7192 \pm 0.0231$ | $0.7790 \pm 0.0146$ | $0.7527 \pm 0.0121$ |
| LaBraMJiang et al. (2024) | $0.7231 \pm 0.0182$ | $0.7810 \pm 0.0133$ | $0.7633 \pm 0.0102$ |
| EEGPTWang et al. (2024a) | $0.2223 \pm 0.0227$ | $0.3111 \pm 0.0110$ | $0.4012 \pm 0.0177$ |
| CBraModWang et al. (2025) | $0.7442 \pm 0.0152$ | $0.8011 \pm 0.0099$ | $\mathbf{0.7865} \pm 0.0110$ |
| CodeBrain | $\mathbf{0.7476} \pm 0.0040$ | $\mathbf{0.8020} \pm 0.0018$ | $0.7835 \pm 0.0033$ |

(+0.61), with notably lower variance. These results underscore its strong generalization to motor imagery decoding under a cross-subject protocol.

Table 6: Performance comparison on the SHU-MI (2-Class) dataset.

| Methods | AUROC | PRAUC | Balanced Accuracy |
|---|---|---|---|
| EEGNet | $0.6283 \pm 0.0152$ | $0.6311 \pm 0.0142$ | $0.5889 \pm 0.0177$ |
| EEGConformer | $0.6351 \pm 0.0101$ | $0.6370 \pm 0.0093$ | $0.5900 \pm 0.0107$ |
| ContraWR | $0.6273 \pm 0.0113$ | $0.6315 \pm 0.0105$ | $0.5873 \pm 0.0128$ |
| ST-Transformer | $0.6431 \pm 0.0111$ | $0.6394 \pm 0.0122$ | $0.5992 \pm 0.0206$ |
| BENDRKostas et al. (2021) | $0.5863 \pm 0.0280$ | $0.5853 \pm 0.0268$ | $0.5573 \pm 0.0227$ |
| BIOTYang et al. (2023) | $0.6609 \pm 0.0127$ | $0.6770 \pm 0.0119$ | $0.6179 \pm 0.0183$ |
| LaBraMJiang et al. (2024) | $0.6604 \pm 0.0091$ | $0.6761 \pm 0.0083$ | $0.6166 \pm 0.0192$ |
| EEGPTWang et al. (2024a) | $0.6241 \pm 0.0071$ | $0.6266 \pm 0.0133$ | $0.5778 \pm 0.0162$ |
| CBraModWang et al. (2025) | $0.6988 \pm 0.0068$ | $0.7139 \pm 0.0088$ | $0.6370 \pm 0.0151$ |
| CodeBrain | $\mathbf{0.7124} \pm 0.0050$ | $\mathbf{0.7166} \pm 0.0106$ | $\mathbf{0.6431} \pm 0.0066$ |

**TUEV** As shown in Table 7, *CodeBrain* achieves the highest Cohen's Kappa (0.6912, an improvement of +0.0140 over the best baseline (Wang et al., 2025)) and Weighted F1 (0.8362) on TUEV. Although its Balanced Accuracy is lower than CBraMod (Wang et al., 2025), we attribute this to reduced sensitivity on the rare *SPSW* class. Since TUEV shares distributional overlap with our pretraining source (TUEG), we stop pretraining at epoch 10 to prevent overfitting, unlike CBraMod's 40-epoch training as discussed in subsection E.2. We also report LaBraM's results based on its original 23-channel setting (Jiang et al., 2024), while *CodeBrain* follows the 16-channel configuration used in CBraMod. Similarly, EEGPT (Wang et al., 2024a) does not adopt a linear fine-tuning protocol but applies two convolutional layers before entering the foundation model, followed by an MLP head. While such architectural choices may enhance performance, we follow their respective fine-tuning settings. In addition, following the experimental setup of CBraMod, we also report the results of experiments conducted by removing all TUEV and TUAB samples from the TUEG dataset. It can be seen that although our model's performance slightly declined on TUEV, it still surpassed other baselines. In such cases, *CodeBrain* still outperforming both LaBraM and EEGPT under their own fine-tuning settings clearly demonstrates the robustness of our approach.

**TUAB** As shown in Table 8, *CodeBrain* achieves the highest Balanced Accuracy (0.8294) on TUAB, slightly outperforming CBraMod (Wang et al., 2025). Similar to TUEV, TUAB is part of the TUH EEG corpus family (Obeid & Picone, 2016) and thus closely aligned with our pretraining

Table 7: Performance comparison on the TUEV (6-Class) dataset.

| Methods | Cohen's Kappa | Weighted F1 | Balanced Accuracy |
|---|---|---|---|
| EEGNet | $0.3577 \pm 0.0155$ | $0.6539 \pm 0.0120$ | $0.3876 \pm 0.0143$ |
| EEGConformer | $0.3967 \pm 0.0195$ | $0.6983 \pm 0.0152$ | $0.4074 \pm 0.0164$ |
| ContraWR | $0.3912 \pm 0.0237$ | $0.6893 \pm 0.0136$ | $0.4384 \pm 0.0349$ |
| ST-Transformer | $0.3765 \pm 0.0306$ | $0.6823 \pm 0.0190$ | $0.3984 \pm 0.0228$ |
| BENDRKostas et al. (2021) | $0.4271 \pm 0.0238$ | $0.6755 \pm 0.0216$ | $0.4363 \pm 0.0245$ |
| BIOTYang et al. (2023) | $0.5273 \pm 0.0249$ | $0.7492 \pm 0.0082$ | $0.5281 \pm 0.0225$ |
| LaBraMJiang et al. (2024) | $0.6637 \pm 0.0093$ | $0.8312 \pm 0.0052$ | $0.6409 \pm 0.0065$ |
| EEGPTWang et al. (2024a) | $0.6351 \pm 0.0134$ | $0.8187 \pm 0.0063$ | $0.6232 \pm 0.0114$ |
| CBraModWang et al. (2025) | $0.6772 \pm 0.0096$ | $0.8342 \pm 0.0064$ | $\mathbf{0.6671} \pm 0.0107$ |
| CodeBrain (Excluding) | $0.6838 \pm 0.0291$ | $0.8293 \pm 0.0163$ | $0.6375 \pm 0.0182$ |
| CodeBrain | $\mathbf{0.6912} \pm 0.0101$ | $\mathbf{0.8362} \pm 0.0048$ | $0.6428 \pm 0.0062$ |

source (TUEG). As discussed in subsection E.2, we adopt an early stopping strategy at epoch 10 to mitigate overfitting to this distribution, which may partly account for the slightly lower AUROC and PRAUC compared to CBraMod, trained for 40 epochs on the same dataset. While LaBraM leverages a 23-channel montage (Jiang et al., 2024) and EEGPT (Wang et al., 2024a) employs two convolutional layers before the foundation model, we retain their respective fine-tuning protocols for comparison. In addition, the impact of duplicate data in the pre-training set on our model is smaller on the TUAB dataset, and in some experiments it can even surpass situations without leaking. This may be due to the larger size of the TUAB dataset. Despite these potentially stronger configurations, *CodeBrain* still exceeds both models under their own settings, highlighting its strong and consistent generalization.

Table 8: Performance comparison on the TUAB (2-Class) dataset.

| Methods | Balanced Accuracy | PRAUC | AUROC |
|---|---|---|---|
| EEGNet | $0.7642 \pm 0.0036$ | $0.8299 \pm 0.0043$ | $0.8412 \pm 0.0031$ |
| EEGConformer | $0.7758 \pm 0.0049$ | $0.8427 \pm 0.0054$ | $0.8445 \pm 0.0038$ |
| ContraWR | $0.7746 \pm 0.0041$ | $0.8421 \pm 0.0104$ | $0.8456 \pm 0.0074$ |
| ST-Transformer | $0.7966 \pm 0.0023$ | $0.8521 \pm 0.0026$ | $0.8707 \pm 0.0019$ |
| BENDRKostas et al. (2021) | $0.7714 \pm 0.0248$ | $0.8412 \pm 0.0215$ | $0.8426 \pm 0.0237$ |
| BIOTYang et al. (2023) | $0.7959 \pm 0.0057$ | $0.8792 \pm 0.0023$ | $0.8815 \pm 0.0043$ |
| LaBraMJiang et al. (2024) | $0.8140 \pm 0.0019$ | $0.8965 \pm 0.0016$ | $0.9022 \pm 0.0009$ |
| EEGPTWang et al. (2024a) | $0.8038 \pm 0.0040$ | $0.8891 \pm 0.0018$ | $0.8811 \pm 0.0015$ |
| CBraModWang et al. (2025) | $0.8289 \pm 0.0022$ | $\mathbf{0.9258} \pm 0.0008$ | $\mathbf{0.9227} \pm 0.0011$ |
| CodeBrain (Excluding) | $0.8288 \pm 0.0064$ | $0.9061 \pm 0.0039$ | $0.9012 \pm 0.0020$ |
| CodeBrain | $\mathbf{0.8294} \pm 0.0013$ | $0.9100 \pm 0.0006$ | $0.9030 \pm 0.0009$ |

## J   HYPERPARAMETER SETTING

We provide detailed hyperparameter configurations for the two-stage pretraining of our *CodeBrain* model and the fine-tuning settings across ten downstream tasks.

### J.1 PRETRAINING SETTINGS

The pretraining process consists of two stages: (1) training the TFDUAL-TOKENIZER and (2) training the EEGSSM. The hyperparameters used in each stage are summarized in Table 9 and Table 10, respectively.

Table 9: Hyperparameters for TFDUAL-TOKENIZER.

| Hyperparameters | Values |
|---|---|
| **TFConv** | |
| Input channels | $\{1, 8, 4\}$ |
| Output channels | $\{8, 4, 4\}$ |
| Kernel size | $\{(1, 15), (1, 3), (1, 3)\}$ |
| Stride | $\{(1, 8), (1, 1), (1, 1)\}$ |
| Padding | $\{(0, 7), (0, 1), (0, 1)\}$ |
| Transformer encoder layers | 12 |
| Transformer decoder layers | 3 |
| Hidden size | 200 |
| MLP size | 800 |
| Attention head number | 8 |
| Temporal Codebook size | $4096 \times 32$ |
| Frequency Codebook size | $4096 \times 32$ |
| Codebook initialization | Random init + $L^2$ normalization |
| Batch size | 256 |
| Peak learning rate | 1e-4 |
| Minimal learning rate | 1e-5 |
| Learning rate scheduler | Cosine |
| Optimizer | AdamW |
| Adam $\beta$ | (0.9, 0.99) |
| Weight decay | 1e-4 |
| Warm-up steps | 5 |
| Total epochs | 20 |
| Data stride | 200 |
| Contrastive temperature ($\tau$) | 0.5 |

Table 10: Hyperparameters of pre-training.

| Hyperparameters | Values |
|---|---|
| Epochs | 10 |
| Batch size | 256 |
| Dropout | 0.1 |
| Optimizer | Adam |
| Learning rate | 1e-4 |
| Adam $\beta$ | (0.9, 0.999) |
| Adam $\epsilon$ | 1e-8 |
| Weight decay | 5e-3 |
| Scheduler | CosineAnnealingLR |
| Minimal learning rate | 1e-5 |
| Clipping gradient norm | 5 |

### J.2 PARAMETERS OF EEGSSM

The model architecture parameters used in EEGSSM during pre-training are shown in Table 11.

Table 11: Configuration of EEGSSM

| Parameters | Values |
|---|---|
| Input size | 200 |
| Hidden dimension | 200 |
| Output size | 200 |
| Number of layers | 8 |
| Max sequence length | 570 |
| SGConv state | 64 |
| SGConv bidirectional | True |
| Layer normalization | True |
| Sliding window attention length | 1s |

## J.3 FINE-TUNING SETTINGS ON DOWNSTREAM TASKS

The *CodeBrain* model is fine-tuned on ten downstream EEG classification tasks, each with task-specific hyperparameters. Following the general strategy adopted by prior EFMs, we adopt a lightweight three-layer MLP as the probe head for all downstream tasks and fine-tune the entire model end-to-end. Table 12 lists the fine-tuning configurations including learning rate, weight decay, dropout rate, and batch size for each task. For sleep staging tasks, due to their strong temporal structure, we follow prior work (Wang et al., 2024a; 2025) and insert an additional Transformer encoder on top of the projection head to jointly model the sequence of 20 consecutive EEG segments. This enables the model to capture inter-epoch transitions critical to sleep stage classification.

Table 12: Fine-tuning hyperparameters for downstream tasks.

| Dataset | Learning Rate | Weight Decay | Dropout | Batch Size |
|---|---|---|---|---|
| FACED | 5e-5 | 5e-4 | 0.1 | 16 |
| SEED-V | 5e-5 | 1e-2 | 0.1 | 64 |
| ISRUC_S1 | 1e-4 | 1e-1 | 0.2 | 48 |
| ISRUC_S3 | 1e-4 | 1e-1 | 0.2 | 48 |
| BCIC2020-T3 | 5e-5 | 5e-2 | 0.1 | 32 |
| Mental Arithmetic | 3e-5 | 1e-3 | 0.1 | 32 |
| CHB-MIT | 3e-5 | 1e-2 | 0.4 | 64 |
| SHU-MI | 5e-5 | 5e-3 | 0.3 | 64 |
| TUEV | 2e-5 | 5e-4 | 0.3 | 64 |
| TUAB | 1e-5 | 5e-5 | 0.4 | 512 |

## K ABLATION ON DESIGN CHOICES

### K.1 ABLATION ON MASK RATIO

We conduct an ablation study to investigate the effect of the mask ratio in the EEGSSM pretraining framework. As shown in Figure 16, downstream performance consistently exhibits a U-shaped trend with respect to the masking ratio across all three datasets: FACED, SEED-V, and ISRUC_S3. Moderate masking (e.g., ratios around 0.4-0.6) leads to optimal performance, whereas excessively low (e.g., 0.1) or high (e.g., 0.9) ratios degrade generalization.

To further illustrate this pattern, we visualize the training loss curves across different mask ratios in Figure 17. Interestingly, higher mask ratios result in slower convergence and higher final training loss, which is expected due to the increased difficulty of the reconstruction task. In contrast, lower mask ratios lead to faster and smoother loss reduction, but do not necessarily yield better downstream performance. This observation suggests a possible *optimization-vs-generalization trade-off*: easier pretext tasks (low mask ratio) are more optimizable but may encourage the model to learn shortcut solutions with limited generalizability, while overly difficult tasks (high mask ratio) may hinder effective representation learning due to insufficient learning signal. Moderate masking strikes a

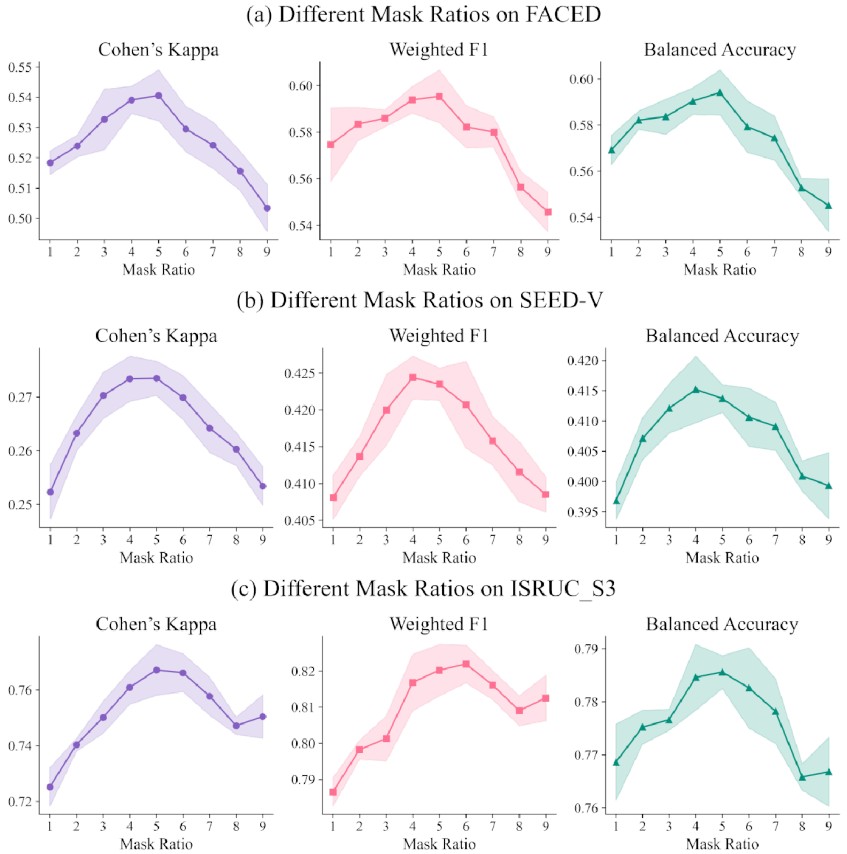

Figure 16: Performance across different mask ratios on FACED, SEED-V, and ISRUC_S3.

balance by being sufficiently challenging to promote abstraction, while still being learnable, thereby facilitating better generalization across downstream tasks.

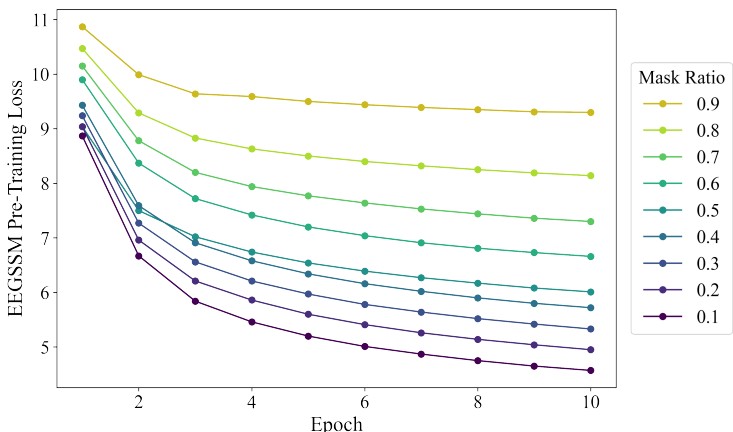

Figure 17: EEGSSM pre-Training loss curve for different mask ratios.

## K.2 ABLATION ON SWA WINDOW SIZE

We conduct an ablation study on the SEED-V and FACED datasets to investigate the effect of the SWA window size in the EEGSSM framework. The SWA window size determines the temporal context involved in attention for modeling intra-patch dependencies. In practice, the window size is set to an odd number so that each target patch attends to a symmetric temporal neighborhood. As shown in Figure 18 and Figure 19, a window size of 1 achieves the best performance on both datasets among the five evaluated settings. Increasing the window size enlarges the temporal context and makes SWA progressively closer to a broader self-attention mechanism. However, increasing the window size does not lead to consistent performance gains. Instead, the results fluctuate across different window sizes, and larger windows do not show clear advantages over the minimal setting. This observation suggests that the primary role of SWA is to capture local intra-patch temporal dynamics, which can be sufficiently modeled within a minimal context. Overall, the results indicate that SWA works best as a lightweight local module, providing effective intra-patch modeling without introducing unnecessary context.

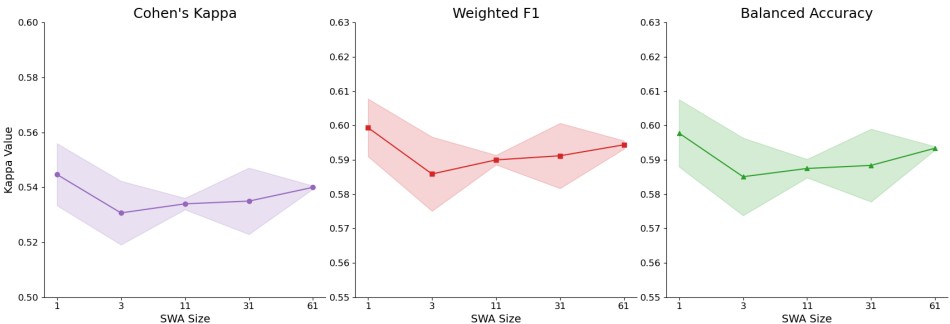

Figure 18: Performance of different SWA window sizes on the FACED dataset.

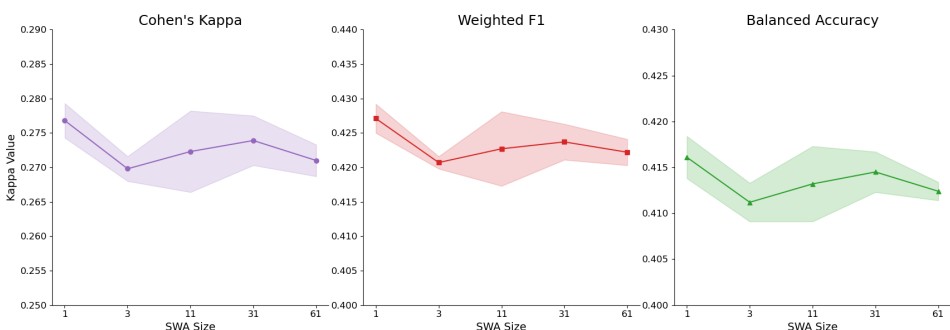

Figure 19: Performance of different SWA window sizes on the SEED-V dataset.

## K.3 ABLATION ON CODEBOOK SIZE

The size of the codebook is an important parameter; a codebook that is too large may lead to unstable training, while a codebook size that is too small may result in mixed information. Ideally, the codebook should maintain a small amount of unused codes but not be 0. We tested several combinations of different time-domain and frequency-domain codebook sizes to observe their unused codes during Tokenizer training.

Table 13 shows the unused temporal codes and unused frequency codes under different codebook size combinations. When both the Temporal codebook size and Frequency codebook size are set to 2048, the unused frequency code is 0, indicating that there are duplicate frequency codes in the current codebook. When choosing a codebook size of 8192, the unused temporal codes reached 3401 and the unused frequency codes reached 1260, indicating that a large number of codes in the completed

training codebook were not used. We ultimately selected 4096 for both. Although the frequency domain often yields richer representations, enlarging its codebook may increase reliance on it, so we kept temporal and frequency codebooks equal. Considering capacity and utilization, 4096-4096 offers the best trade-off.

Table 13: Temporal and frequency codebook statistics

| Temporal Codebook Size | Frequency Codebook Size | Unused Temporal Codes | Unused Frequency Codes |
|---|---|---|---|
| 2048 | 2048 | 12 | 0 |
| 2048 | 4096 | 116 | 0 |
| **4096** | **4096** | 225 | 165 |
| 4096 | 8192 | 321 | 931 |
| 8192 | 8192 | 3401 | 1260 |

### K.4 Ablation on Patch Size

The size of the patch window is also an important adjustable parameter, which affects temporal resolutions and masking strategies. To explore the impact of patch window size on the model, we used window sizes ranging from 0.5s to 5s for complete two-stage pre-training and full-parameter fine-tuning. Note that for some datasets where the patch size is larger than the channel length, such as SEED-V, we pad the portion exceeding the available data to match the patch size in this experiment.

Table 14 and Table 15 show the performance of our method with different patch sizes on the SEED-V dataset and ISRUC_3 dataset. Notably, in SEED-V, patch lengths longer than 1s require heavy padding, causing large performance drops; in ISRUC_S3, the shortest 0.5s patches achieve the worst performance, likely because they fragment key waveforms in sleep staging (e.g., Spindle $\geq$0.5s). From these results, the 1s setting is supported by two key considerations:

**Broad compatibility with downstream task**. 1s is a divisor of most downstream sequence lengths (1-30s), minimizing padding and ensuring transferability. For example, on the SEED-V dataset, if the patch size chosen by the model is greater than 1s, some methods (such as padding) need to be adopted to enable model training. These methods may usually impair the model's performance because they introduce additional noise or increase computational load. EEG datasets with durations less than 1 second are relatively rare, as most datasets have at least 1 second of data. If the data does not exactly match the whole seconds, the cost of processing such a dataset is also relatively small. Prior EEG foundation models (e.g., LaBram (Jiang et al., 2024), CBraMod (Wang et al., 2025)) also adopt 1s patches for this reason.

**Semantic integrity.** Choosing a 1s patch length preserves the natural structure of EEG waveforms and prevents semantic fragmentation. Many physiologically meaningful EEG events have characteristic durations: for example, K-complexes are typically around 1s, spindles last 0.5-2s, and event-related potentials such as P300 occur in the range of 0.3-0.6s. If patches are shorter than these characteristic scales, the temporal branch of the TFDual-Tokenizer may only capture partial fragments of these waveforms, leading to loss of semantic context.

Table 14: Performance on SEED-V with different patch sizes

| Patch Length | Cohen's Kappa | Weighted F1 | Balanced Accuracy |
|---|---|---|---|
| 0.5s | $0.2640 \pm 0.0035$ | $0.4035 \pm 0.0025$ | $0.3841 \pm 0.0025$ |
| **1s** | $\mathbf{0.2735} \pm \mathbf{0.0032}$ | $\mathbf{0.4235} \pm \mathbf{0.0022}$ | $\mathbf{0.4137} \pm \mathbf{0.0023}$ |
| 2s | $0.1545 \pm 0.0046$ | $0.3313 \pm 0.0036$ | $0.3279 \pm 0.0049$ |
| 5s | $0.1557 \pm 0.0043$ | $0.3340 \pm 0.0029$ | $0.3271 \pm 0.0045$ |

### K.5 Ablation on SGConv Kernel Parameters

To justify the choice of the SGConv decay coefficient, we conduct a sensitivity analysis on three representative downstream datasets (FACED, SEED-V, ISRUC_S3). The results are shown in Table 16 The decay parameter $\alpha$ controls how rapidly spatial kernel weights diminish with topological distance and, therefore, plays a central role in balancing locality preservation and kernel sparsification.

Table 15: Performance on ISRUC_3 with different patch sizes

| Patch Length | Cohen's Kappa | Weighted F1 | Balanced Accuracy |
|---|---|---|---|
| 0.5s | $0.7405 \pm 0.0102$ | $0.7950 \pm 0.0097$ | $0.7420 \pm 0.0081$ |
| **1s** | $\mathbf{0.7671} \pm \mathbf{0.0091}$ | $\mathbf{0.8202} \pm \mathbf{0.0071}$ | $\mathbf{0.7856} \pm \mathbf{0.0031}$ |
| 2s | $0.7592 \pm 0.0079$ | $0.8113 \pm 0.0081$ | $0.7753 \pm 0.0087$ |
| 5s | $0.7601 \pm 0.0075$ | $0.8096 \pm 0.0074$ | $0.7791 \pm 0.0096$ |

Table 16: Sensitivity analysis of the SGConv decay coefficient $\alpha$ across three downstream datasets.

| SEED-V | Cohen's Kappa | Weighted F1 | Balanced Accuracy |
|---|---|---|---|
| $\alpha = 0.5$ **(default)** | $\mathbf{0.2735} \pm \mathbf{0.0032}$ | $\mathbf{0.4235} \pm \mathbf{0.0022}$ | $\mathbf{0.4137} \pm \mathbf{0.0023}$ |
| $\alpha = 0.1$ | $0.2629 \pm 0.0017$ | $0.4159 \pm 0.0006$ | $0.4044 \pm 0.0017$ |
| $\alpha = 0.9$ | $0.2318 \pm 0.0074$ | $0.3851 \pm 0.0141$ | $0.3819 \pm 0.0046$ |
| $\alpha = 2.0$ | $0.2332 \pm 0.0077$ | $0.3851 \pm 0.0090$ | $0.3809 \pm 0.0069$ |
| **FACED** | Cohen's Kappa | Weighted F1 | Balanced Accuracy |
| $\alpha = 0.5$ **(default)** | $\mathbf{0.5406} \pm \mathbf{0.0084}$ | $\mathbf{0.5953} \pm \mathbf{0.0113}$ | $\mathbf{0.5941} \pm \mathbf{0.0090}$ |
| $\alpha = 0.1$ | $0.5295 \pm 0.0080$ | $0.5853 \pm 0.0074$ | $0.5839 \pm 0.0069$ |
| $\alpha = 0.9$ | $0.4681 \pm 0.0069$ | $0.5270 \pm 0.0065$ | $0.5314 \pm 0.0050$ |
| $\alpha = 2.0$ | $0.4782 \pm 0.0037$ | $0.5282 \pm 0.0068$ | $0.5360 \pm 0.0035$ |
| **ISRUC_S3** | Cohen's Kappa | Weighted F1 | Balanced Accuracy |
| $\alpha = 0.5$ **(default)** | $\mathbf{0.7671} \pm \mathbf{0.0091}$ | $\mathbf{0.8202} \pm \mathbf{0.0071}$ | $\mathbf{0.7856} \pm \mathbf{0.0031}$ |
| $\alpha = 0.1$ | $0.7132 \pm 0.0646$ | $0.7834 \pm 0.0523$ | $0.7575 \pm 0.0621$ |
| $\alpha = 0.9$ | $0.6073 \pm 0.0230$ | $0.6966 \pm 0.0176$ | $0.6564 \pm 0.0058$ |
| $\alpha = 2.0$ | $0.5600 \pm 0.0492$ | $0.6572 \pm 0.0430$ | $0.6227 \pm 0.0427$ |

Across all datasets, $\alpha = 0.5$ consistently yields the highest performance. Larger values (e.g., $\alpha = 0.9$) moderately weaken spatial locality, while very small values (e.g., $\alpha = 0.1$) oversparsify the kernel and markedly reduce accuracy. Increasing $\alpha$ towards and beyond 1 (e.g., $\alpha = 2.0$) weakens the decay and assigns relatively larger weights to distant sub-kernels, which harms spatial locality and leads to clear performance drops, especially on ISRUC_S3. In addition, excessively large decay coefficients may amplify gradients during backpropagation and introduce training instability. For these reasons, we recommend $\alpha$ to $\leq 1$ and adopt $\alpha = 0.5$ as a stable and well-performing default.

## K.6 ABLATION ON SUBBAND

To better understand the frequency dependencies encoded by *CodeBrain*, we conduct a systematic subband ablation study by masking each of the five canonical EEG frequency ranges: $\delta$ (0.5-4 Hz), $\theta$ (4-8 Hz), $\alpha$ (8-13 Hz), $\beta$ (13-30 Hz), and $\gamma$ (¿30 Hz). Unlike random dropout, this setting allows us to examine how the model allocates representational importance across physiologically meaningful frequency components. Experiments are performed on both the emotion recognition dataset (SEED-V) and the sleep-staging dataset (ISRUC_S3), covering two distinct neurophysiological tasks.

### K.6.1 SEED-V (62 CHANNELS): SUBBAND CONTRIBUTIONS IN EMOTION RECOGNITION

Several key findings could be observed in this experiments:

- **Dominance of low-frequency structure.** Removing $\delta$, $\theta$, or $\alpha$ bands produces near-collapse of performance, indicating that emotional states are primarily encoded in slow and mid-range oscillations. This matches prior affective neuroscience findings showing that emotional arousal and valence strongly modulate rhythms below 13 Hz.

- **Residual robustness at higher frequencies.** Ablating $\beta$ and especially $\gamma$ reduces performance but does not catastrophically impair decoding. This suggests that *CodeBrain*

Table 17: Subband ablation results on the SEED-V dataset.

| Removed Band | Cohen's Kappa | Weighted F1 | Balanced Accuracy |
|---|---|---|---|
| None | **0.2735** ± 0.0032 | **0.4235** ± 0.0022 | **0.4137** ± 0.0023 |
| $\delta$ | 0.0297 ± 0.0027 | 0.2075 ± 0.0105 | 0.2201 ± 0.0004 |
| $\theta$ | 0.0142 ± 0.0050 | 0.1780 ± 0.0176 | 0.2098 ± 0.0039 |
| $\alpha$ | 0.0082 ± 0.0070 | 0.1306 ± 0.0536 | 0.2058 ± 0.0049 |
| $\beta$ | 0.0442 ± 0.0120 | 0.2377 ± 0.0068 | 0.2360 ± 0.0095 |
| $\gamma$ | 0.1372 ± 0.0081 | 0.3141 ± 0.0086 | 0.3092 ± 0.0044 |

leverages high-frequency activity as complementary contextual cues rather than primary discriminative features.

- **Contrast with raw-signal models.** Compared with prior end-to-end CNN/RNN models, the degradation patterns reveal that our decoupled tokenizer and cross-scale encoder preserve structured frequency dependencies rather than relying disproportionately on one frequency range.

These observations highlight that the learned frequency representation is aligned with known emotional knowledge while still maintaining robustness across a broad range of frequencies.

### K.6.2 ISRUC_S3 (6 CHANNELS): SUBBAND CONTRIBUTIONS IN SLEEP STAGING

Table 18: Subband ablation results on the ISRUC_S3 dataset.

| Removed Band | Cohen's Kappa | Weighted F1 | Balanced Accuracy |
|---|---|---|---|
| None | **0.7671** ± 0.0091 | **0.8202** ± 0.0071 | **0.7856** ± 0.0031 |
| $\delta$ | 0.0225 ± 0.0391 | 0.1751 ± 0.0843 | 0.2043 ± 0.0237 |
| $\theta$ | 0.0728 ± 0.0557 | 0.1386 ± 0.0341 | 0.2812 ± 0.0749 |
| $\alpha$ | 0.1048 ± 0.0221 | 0.1762 ± 0.0186 | 0.3226 ± 0.0293 |
| $\beta$ | 0.2131 ± 0.0660 | 0.2867 ± 0.1185 | 0.3835 ± 0.0446 |
| $\gamma$ | 0.3618 ± 0.0012 | 0.4831 ± 0.0291 | 0.4764 ± 0.0082 |

Results on ISRUC_S3 sleep staging dataset reveal a qualitatively different frequency profile from SEED-V:

- **Critical dependence on $\delta$ and $\theta$.** Removing slow-wave components almost eliminates information, consistent with their central role in NREM transitions and slow oscillations during deep sleep.

- **Higher-frequency bands remain more robust.** Masking $\beta$ and $\gamma$ still reduces performance but to a lesser extent, reflecting the fact that spindle- or arousal-related faster bursts are less dominant in the 6-channel ISRUC montage.

- **Physiology-specific feature reliance.** The frequency sensitivity patterns differ from SEED-V, demonstrating that *CodeBrain* adapts its feature allocation depending on task demands rather than relying on a fixed frequency prior.

The subband ablation results across two very different datasets show that *CodeBrain* could capture frequency structure, leveraging low-frequency dynamics for both emotion and sleep tasks while maintaining robustness to higher-band perturbations.

## L  BACKBONE EFFICIENCY COMPARISON

To evaluate the computational efficiency of our SGConv layer in the EEGSSM block, we conduct experiments by replacing it with three common sequence modeling modules: CNN, LSTM, and Transformer. We compare their model sizes, floating-point operations (FLOPs), and iteration times, as shown in Figure 20. Specifically, the CNN variant uses a 3-layer depthwise separable convolution

block, while the LSTM and Transformer variants use a single layer of standard LSTM and Transformer Encoder (implemented by Pytorch), respectively. In terms of parameter count, SGConv contains 15.17M parameters, fewer than CNN (16.22M), LSTM (17.35M), and Transformer (21.2M). For FLOPs, SGConv also achieves the lowest computational cost at 8.74G, compared to Transformer's 27.79G. Regarding iteration time, SGConv is slightly slower than Transformer and CNN models in terms of training speed, but it outperforms the LSTM model. In summary, SGConv effectively reduces the number of parameters while maintaining computational complexity, which helps the model to be trained and inferred on smaller GPUs.

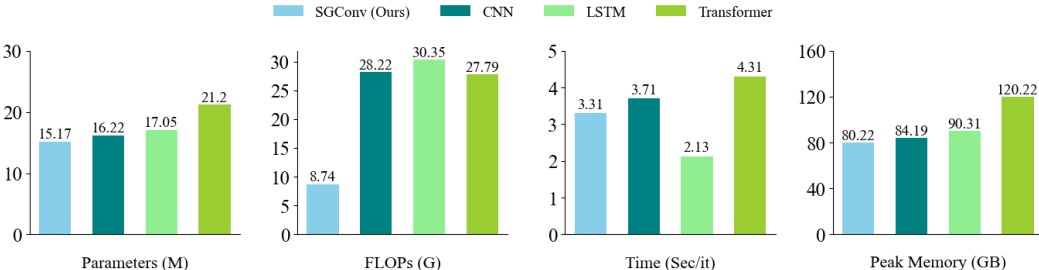

Figure 20: Computational overhead of using different backbones in the EEGSSM module.

To further contextualize the computational efficiency of *CodeBrain*, we provide a comparison against widely used EEG foundation-model baselines. Table 19 summarizes model parameters, multiply-accumulate operations (MACs), and floating-point operations (FLOPs), offering stable and hardware-agnostic metrics across architectures. Overall, *CodeBrain* achieves a favorable balance between computational cost and representational capacity. Its FLOPs and MACs remain substantially lower than large-scale models such as BENDR and EEGPT, while maintaining higher parameter efficiency than CBraMod and LaBraM. Notably, *CodeBrain* occupies a middle ground in model size: smaller than EEGPT while offering richer representational power than compact baselines like BIOT. This balanced compute-performance trade-off aligns with our design goal of building an efficient yet high-capacity EEG foundation model suitable for both research and deployment on modest hardware.

Table 19: Compute comparison between *CodeBrain* and representative EEG foundation-model baselines.

| Model | MACs | Params | FLOPs |
|---|---|---|---|
| BENDR | 12.51G | 959.84M | 25.02G |
| BIOT | 0.255G | 3.20M | 0.510G |
| LaBraM | 0.67G | 6.02M | 1.34G |
| CBraMod | 0.64G | 4.03M | 1.29G |
| EEGPT | 4.89G | 25.24M | 9.79G |
| **CodeBrain (Ours)** | 4.37G | 15.17M | 8.74G |

## M  MODEL ROBUSTNESS

### M.1  RANDOM CHANNEL DROPOUT

In real-world scenarios, the collection of EEG often encounters situations where channels are missing, especially when using machines from different manufacturers. To test the model's performance on datasets with missing channel data, we randomly mask some channels in the training data for full parameter fine-tuning. We selected the FACED and SEED-V datasets for experiments because they represent short-sequence and long-sequence cases respectively, and their numbers of channels are relatively complete.

We evaluate the performance of the *CodeBrain* model and CBraMod model in scenarios with missing channels. We conducted three different experiments, randomly masking 12.5%, 25%, and 50% of the channels in each experiment, respectively. Figure 21 shows results of the experiment. It can be seen

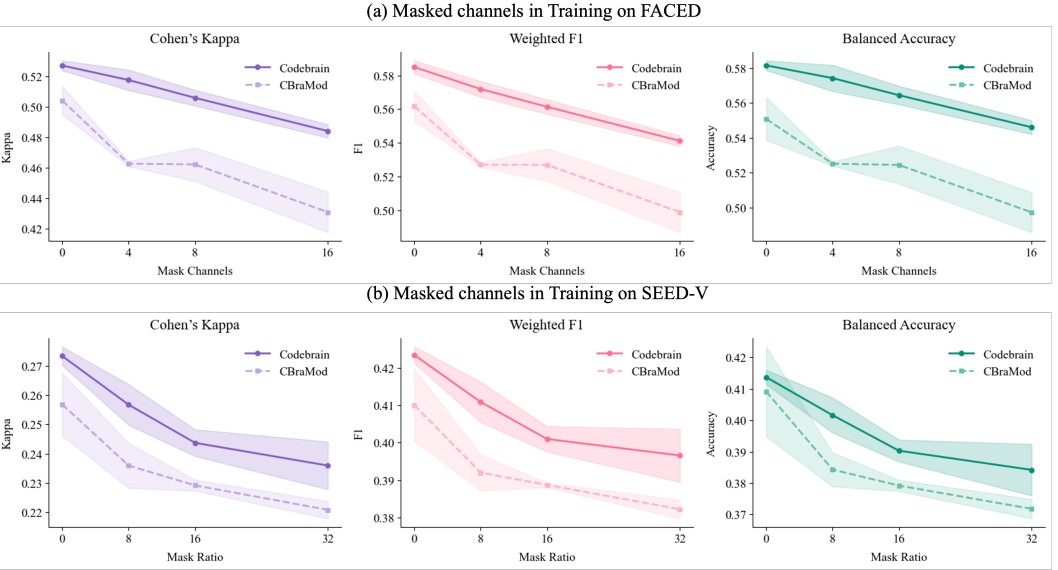

Figure 21: Performance after randomly masking different numbers of channels during the full parameter fine-tuning stage.

that our *CodeBrain* model outperforms the CBraMod model in all channel masking scenarios. On the FACED dataset, our model's performance after masking 25% of the channels is still close to that of CBraMod without masking. The performance decline of the CBraMod model is also faster than that of our model. This trend is even more pronounced on the SEED-V dataset. Through this experiment, we can demonstrate that the channel robustness of *CodeBrain* is stronger than that of the CBraMod model, retaining most of its performance even in cases of channel failure.

## M.2 Brain-Region Channel Dropout

While random channel masking simulates incidental electrode failures, it does not capture the structured spatial organization of the EEG montage. To provide a neuroscientifically meaningful robustness evaluation, we conduct region-based masking on both the high-density SEED-V dataset (62 channels) and the minimal-montage ISRUC_S3 dataset (6 channels). These experiments simulate clinically relevant scenarios such as reference-electrode failures, lobe-specific dropout, and hemisphere-level signal loss. We also compare *CodeBrain* with the strongest baseline, CBraMod.

### M.2.1 Region-Based Ablations on the High-Density SEED-V Dataset (62 Channels)

We design nine anatomically meaningful masking patterns, including (1) reference/hemisphere failures (midline-only, left-hemisphere masked, right-hemisphere masked), and (2) lobe-level dropout over occipital, frontal, temporal, central, frontocentral, and parietal regions. The results are summarized in Table 20.

Across nearly all masking conditions, *CodeBrain* maintains stronger performance than CBraMod, particularly under lobe-level dropout (occipital, temporal, central), suggesting that its spatial-temporal modeling is less reliant on any single anatomical region. The severe degradation under midline-only signals further highlights the importance of distributed multi-lobe information in emotion-related EEG.

Table 20: Region-masking results on the SEED-V dataset (62 channels).

| Mask Setting | Model | Cohen's Kappa | Weighted F1 | Balanced Accuracy |
|---|---|---|---|---|
| Baseline (no mask) | CBraMod | $0.2569 \pm 0.0143$ | $0.4101 \pm 0.0108$ | $0.4091 \pm 0.0097$ |
| | CodeBrain | $\mathbf{0.2735} \pm 0.0032$ | $\mathbf{0.4235} \pm 0.0022$ | $\mathbf{0.4137} \pm 0.0023$ |
| Only midline | CBraMod | $0.0114 \pm 0.0098$ | $\mathbf{0.1740} \pm \mathbf{0.0121}$ | $0.2077 \pm 0.0018$ |
| | CodeBrain | $\mathbf{0.0201} \pm 0.0098$ | $0.1610 \pm 0.0087$ | $\mathbf{0.2169} \pm 0.0067$ |
| Left hemisphere | CBraMod | $0.0672 \pm 0.0260$ | $0.2505 \pm 0.0227$ | $0.2541 \pm 0.0226$ |
| | CodeBrain | $\mathbf{0.0956} \pm 0.0030$ | $\mathbf{0.2779} \pm 0.0051$ | $\mathbf{0.2750} \pm 0.0030$ |
| Right hemisphere | CBraMod | $0.0586 \pm 0.0171$ | $0.2500 \pm 0.0214$ | $0.2466 \pm 0.0149$ |
| | CodeBrain | $\mathbf{0.0678} \pm 0.0195$ | $\mathbf{0.2540} \pm 0.0066$ | $\mathbf{0.2569} \pm 0.0166$ |
| Occipital | CBraMod | $0.1316 \pm 0.0217$ | $0.3066 \pm 0.0167$ | $0.3068 \pm 0.0174$ |
| | CodeBrain | $\mathbf{0.2258} \pm 0.0067$ | $\mathbf{0.3851} \pm 0.0040$ | $\mathbf{0.3818} \pm 0.0090$ |
| Frontal | CBraMod | $0.0632 \pm 0.0222$ | $0.2486 \pm 0.0177$ | $0.2491 \pm 0.0189$ |
| | CodeBrain | $\mathbf{0.1255} \pm 0.0090$ | $\mathbf{0.3025} \pm 0.0080$ | $\mathbf{0.2976} \pm 0.0046$ |
| Temporal | CBraMod | $0.1571 \pm 0.0284$ | $0.3309 \pm 0.0222$ | $0.3265 \pm 0.0227$ |
| | CodeBrain | $\mathbf{0.2311} \pm 0.0008$ | $\mathbf{0.3922} \pm 0.0010$ | $\mathbf{0.3824} \pm 0.0004$ |
| Central | CBraMod | $0.1562 \pm 0.0419$ | $0.3305 \pm 0.0336$ | $0.3247 \pm 0.0321$ |
| | CodeBrain | $\mathbf{0.2410} \pm 0.0078$ | $\mathbf{0.3997} \pm 0.0067$ | $\mathbf{0.3894} \pm 0.0055$ |
| Frontocentral | CBraMod | $\mathbf{0.1304} \pm 0.0304$ | $\mathbf{0.3077} \pm \mathbf{0.0263}$ | $\mathbf{0.3039} \pm 0.0244$ |
| | CodeBrain | $0.1254 \pm 0.0090$ | $0.3025 \pm 0.0080$ | $0.2976 \pm 0.0047$ |
| Parietal | CBraMod | $0.1571 \pm 0.0284$ | $0.3309 \pm 0.0222$ | $0.3265 \pm 0.0227$ |
| | CodeBrain | $\mathbf{0.2311} \pm 0.0008$ | $\mathbf{0.3922} \pm 0.0010$ | $\mathbf{0.3824} \pm 0.0004$ |

## M.2.2 REGION-BASED ABLATIONS ON THE MINIMAL-MONTAGE ISRUC_S3 DATASET (6 CHANNELS)

To test robustness under extreme spatial sparsity, we perform structured region masking on the 6-channel A1/A2-referenced ISRUC_S3 montage. We evaluate reference-electrode failures (masking A1 or A2) and lobe-specific removal (frontal, central, occipital). Results are shown in Table 21.

Table 21: Region-masking results on the ISRUC_S3 dataset (6 channels).

| Mask Setting | Model | Cohen's Kappa | Weighted F1 | Balanced Accuracy |
|---|---|---|---|---|
| Baseline (no mask) | CBraMod | $0.7407 \pm 0.0251$ | $0.8056 \pm 0.0219$ | $0.7844 \pm 0.0126$ |
| | CodeBrain | $\mathbf{0.7671} \pm 0.0091$ | $\mathbf{0.8202} \pm 0.0071$ | $\mathbf{0.7856} \pm 0.0031$ |
| Right hemisphere | CBraMod | $0.6616 \pm 0.0263$ | $0.7392 \pm 0.0292$ | $0.6706 \pm 0.0212$ |
| | CodeBrain | $\mathbf{0.7430} \pm 0.0131$ | $\mathbf{0.8047} \pm 0.0086$ | $\mathbf{0.7760} \pm 0.0279$ |
| Left hemisphere | CBraMod | $0.6198 \pm 0.1127$ | $0.6964 \pm 0.0958$ | $0.6838 \pm 0.0815$ |
| | CodeBrain | $\mathbf{0.7318} \pm 0.0252$ | $\mathbf{0.7989} \pm 0.0196$ | $\mathbf{0.7819} \pm 0.0133$ |
| Occipital | CBraMod | $0.7089 \pm 0.0412$ | $0.7598 \pm 0.0321$ | $0.7340 \pm 0.0269$ |
| | CodeBrain | $\mathbf{0.7447} \pm 0.0226$ | $\mathbf{0.8065} \pm 0.0187$ | $\mathbf{0.7824} \pm 0.0080$ |
| Central | CBraMod | $0.6725 \pm 0.0106$ | $0.7526 \pm 0.0074$ | $0.6979 \pm 0.0164$ |
| | CodeBrain | $\mathbf{0.7479} \pm 0.0253$ | $\mathbf{0.8111} \pm 0.0197$ | $\mathbf{0.7790} \pm 0.0155$ |
| Frontal | CBraMod | $0.7134 \pm 0.0245$ | $0.7796 \pm 0.0184$ | $0.7250 \pm 0.0327$ |
| | CodeBrain | $\mathbf{0.7295} \pm 0.0531$ | $\mathbf{0.7936} \pm 0.0408$ | $\mathbf{0.7825} \pm 0.0282$ |

Even under this sparse spatial setup, *CodeBrain* consistently maintains strong performance compared with strong baselines across all mask types. This indicates that the learned representations are robust to structured regional dropout and remain stable even when half or more channels are removed.

Across both high-density and minimal-montage datasets, region-based ablation demonstrates that *CodeBrain* preserves strong performance under structured channel dropout, highlighting its spatial robustness and reliable modeling of cross-regional EEG dependencies.

### M.3 NON-STATIONARY ROBUSTNESS

EEG signals are inherently non-stationary, with gradual fluctuations caused by electrode impedance changes, autonomic modulation, motion artifacts, and slow drift in sensor baselines. To examine how well the learned representations tolerate such structured temporal drift, we introduce a simple but effective perturbation: a linear baseline shift. This perturbation exaggerates slow-varying non-stationarity beyond what naturally appears in the data, providing a controlled stress test of robustness. We evaluate both SEED-V (emotion recognition) and ISRUC_S3 (sleep staging), comparing *CodeBrain* with the strongest baseline, CBraMod. Results are summarized in Table 22.

Table 22: Non-stationary robustness under linear baseline shift on SEED-V and ISRUC_S3.

| **SEED-V** | | | | |
|---|---|---|---|---|
| **Setting** | **Model** | **Cohen's Kappa** | **Weighted F1** | **Balanced Accuracy** |
| Reference (no shift) | **CodeBrain** | $\mathbf{0.2735} \pm 0.0032$ | $\mathbf{0.4235} \pm 0.0022$ | $\mathbf{0.4137} \pm 0.0023$ |
| | CBraMod | $0.2569 \pm 0.0143$ | $0.4101 \pm 0.0108$ | $0.4091 \pm 0.0097$ |
| Linear baseline shift | **CodeBrain** | $\mathbf{0.2170} \pm 0.0345$ | $\mathbf{0.3799} \pm 0.0276$ | $\mathbf{0.3706} \pm 0.0259$ |
| | CBraMod | $0.2027 \pm 0.0147$ | $0.3658 \pm 0.0136$ | $0.3628 \pm 0.0111$ |
| **ISRUC_S3** | | | | |
| **Setting** | **Model** | **Cohen's Kappa** | **Weighted F1** | **Balanced Accuracy** |
| Reference (no shift) | **CodeBrain** | $\mathbf{0.7671} \pm 0.0091$ | $\mathbf{0.8202} \pm 0.0071$ | $\mathbf{0.7856} \pm 0.0031$ |
| | CBraMod | $0.7407 \pm 0.0251$ | $0.8056 \pm 0.0219$ | $0.7844 \pm 0.0126$ |
| Linear baseline shift | **CodeBrain** | $\mathbf{0.4762} \pm 0.0835$ | $\mathbf{0.5914} \pm 0.0668$ | $\mathbf{0.5617} \pm 0.0630$ |
| | CBraMod | $0.4242 \pm 0.0068$ | $0.5478 \pm 0.0087$ | $0.5268 \pm 0.0085$ |

This controlled non-stationarity stress test reveals several consistent patterns. Moderate linear drift introduces clear performance degradation across both datasets, as expected for models trained under largely stationary conditions. The effect is especially pronounced on ISRUC_S3, where the limited channel count amplifies the impact of baseline shifts. Despite this, *CodeBrain* retains a larger proportion of its original performance than CBraMod, suggesting that it yields more stable representations under slow global waveform drift. We also observe task-dependent differences in sensitivity: emotion recognition is disproportionately affected in low-frequency components, whereas sleep staging exhibits a more uniform degradation across metrics, reflecting the different spectral structures these tasks rely on. Overall, the results indicate that *CodeBrain* maintains strong robustness to non-stationary perturbations, making it suitable for real-world settings where gradual baseline drift and electrode instability are unavoidable.

## N DETAILED RESULTS ON SCALING LAWS

We provide the detailed scaling law results for both data and model size across three representative EEG datasets (FACED, SEED-V, and ISRUC_S3), covering three evaluation metrics. For brevity, only Cohen's kappa scores are included in the main text, while full results are provided in this section. Prior work (Wang et al., 2025; Jiang et al., 2024) has explored the effect of scaling EEG foundation models using 1 to 1000 hours of pretraining data. We extend this analysis in two key dimensions:

1. Scaling the pretraining data volume from 1k up to 9k hours.

2. Investigating model scaling by varying the depth of the EEGSSM encoder from 3 layers (3.86M parameters) to 24 layers (146.75M parameters) and the hidden size from 128 to 384.

## N.1 Scaling Laws with Respect to Training Data Volume

We examine how the volume of pretraining data influences the downstream performance of *CodeBrain*. Specifically, we scale the pretraining duration from 1k to 9k hours and evaluate the resulting models on three downstream datasets: FACED, SEED-V, and ISRUC_S3. As shown in Figure 22, increasing the amount of pretraining data generally leads to consistent improvements across all datasets and metrics. On FACED and ISRUC_S3, performance gains are steady throughout the entire range up to 9k hours, while on SEED-V, the trend is more modest and plateaus after 5k hours. These results highlight the importance of large-scale data for representation learning in EEG and suggest that further scaling may continue to yield performance benefits.

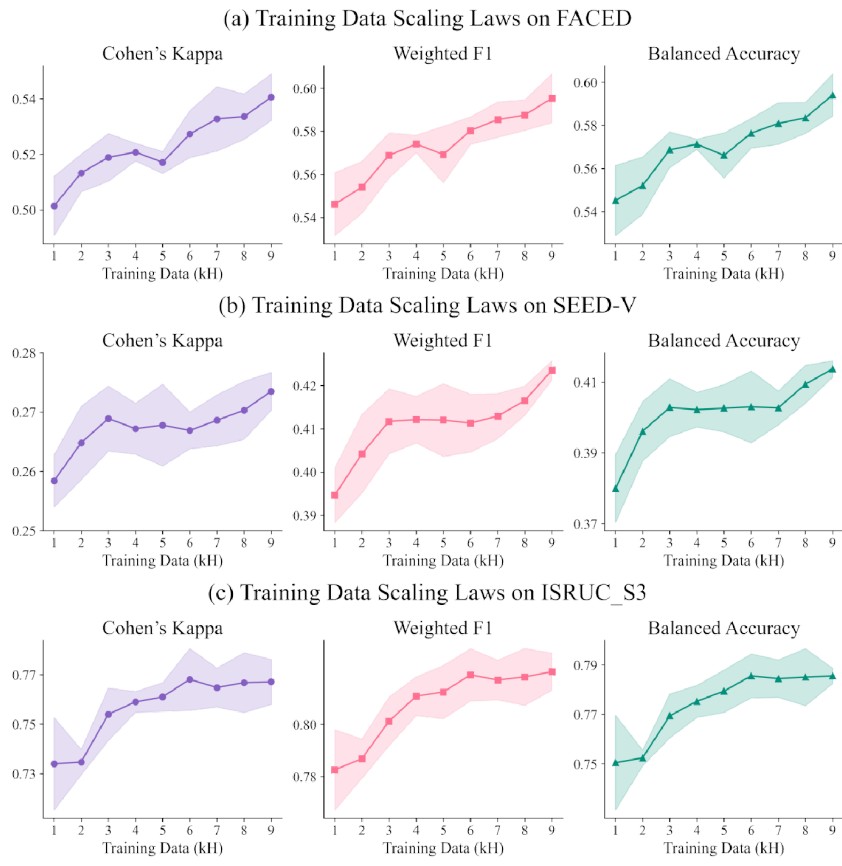

Figure 22: Training data scaling laws on FACED, SEED-V, and ISRUC_S3.

In addition, we visualize the pretraining optimization behavior across different data scales in Figure 23. As expected, larger pretraining data consistently lead to lower training loss, indicating more effective representation learning. Notably, the convergence curves become progressively smoother and more stable as training data volume increases, suggesting improved optimization stability in large-scale regimes. While smaller training data volumes (e.g., 1k-3k hours) show relatively high starting loss and slower convergence, larger training data volumes (6k-9k hours) reach lower final losses and exhibit diminishing returns, aligning with trends observed in downstream performance. These findings provide further empirical support for the scalability of EEG foundation models and reinforce the role of large data in enhancing both optimization and generalization.

## N.2 Scaling Laws with Respect to Model Size

We further investigate how model parameters affect downstream performance by scaling the number of layers in the EEGSSM encoder from 3 to 8, resulting in parameter counts ranging from 6.82M to 15.17M. Figure 24 presents the performance curves as model size increases. Across all three datasets

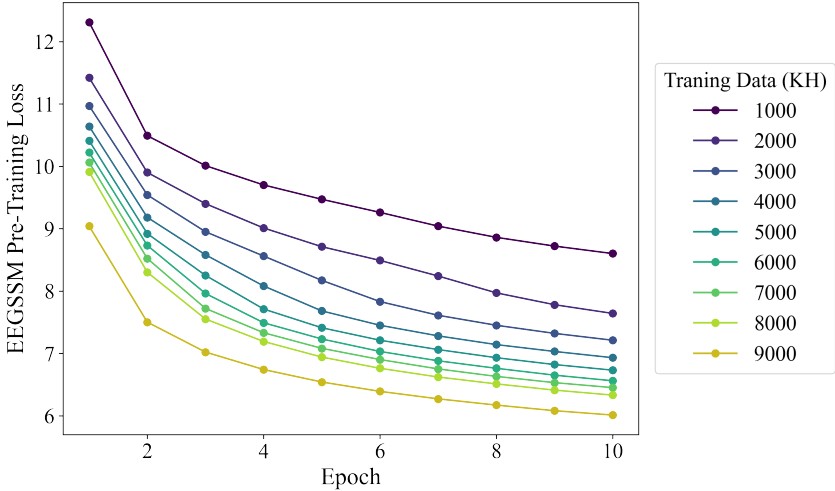

Figure 23: EEGSSM pre-training loss curve for different training data volume.

and evaluation metrics, we observe a consistent performance gain as the model size increases. The improvements are particularly pronounced on the FACED and ISRUC_S3 datasets, where all three metrics show steady growth up to the largest model. In contrast, performance on SEED-V improves more modestly and begins to plateau beyond 13.5M parameters. These results suggest that increasing model capacity can enhance generalization ability, especially for datasets with richer structure or more complex temporal dynamics, while also indicating that optimal scaling may be task-dependent.

To better understand the optimization behavior during pretraining, we plot the training loss curves for different model sizes in Figure 25. As expected, larger models consistently achieve lower final training loss, indicating stronger capacity to fit the pretraining objective. The loss reduction is particularly evident when increasing from 3 to 6 layers, while the gain starts to saturate beyond 7 layers. Even when increased to 24 layers, the reduction in pre-training loss brought by more than 100M parameters is not significant. This trend mirrors the downstream performance in Figure 24, suggesting that both optimization efficiency and generalization benefit from increased model size—though with diminishing returns as parameter count grows. These results reinforce the scalability of EEGSSM and underscore the importance of balancing capacity with task-specific requirements.

Our results demonstrate consistent improvements in downstream performance as both data volume and model capacity increase, suggesting that EEG foundation models may continue to benefit from further scaling, similar to trends observed in vision and language domains.

### N.3 COMPUTATIONAL ANALYSIS ACROSS SCALES.

Table 23: Computational analysis across different model scales.

| Layer | Hidden Size | Params | FLOPs | Throughput | GPU Memory |
|-------|-------------|----------|---------|------------|------------|
| 3 | 128 | 3.96M | 1.7G | 4.90 | 4.87 |
| 3 | 200 | 6.82M | 3.35G | 4.62 | 5.63 |
| 4 | 200 | 8.49M | 4.43G | 3.77 | 6.48 |
| 5 | 200 | 10.16M | 5.51G | 3.67 | 7.33 |
| 6 | 200 | 11.83M | 6.58G | 3.52 | 8.10 |
| 7 | 200 | 13.50M | 7.66G | 2.94 | 8.95 |
| 8 | 200 | 15.17M | 8.74G | 2.78 | 9.79 |
| 12 | 256 | 34.38M | 19.04G | 1.84 | 15.41 |
| 24 | 384 | 146.75M | 72.99G | 1.47 | 38.43 |

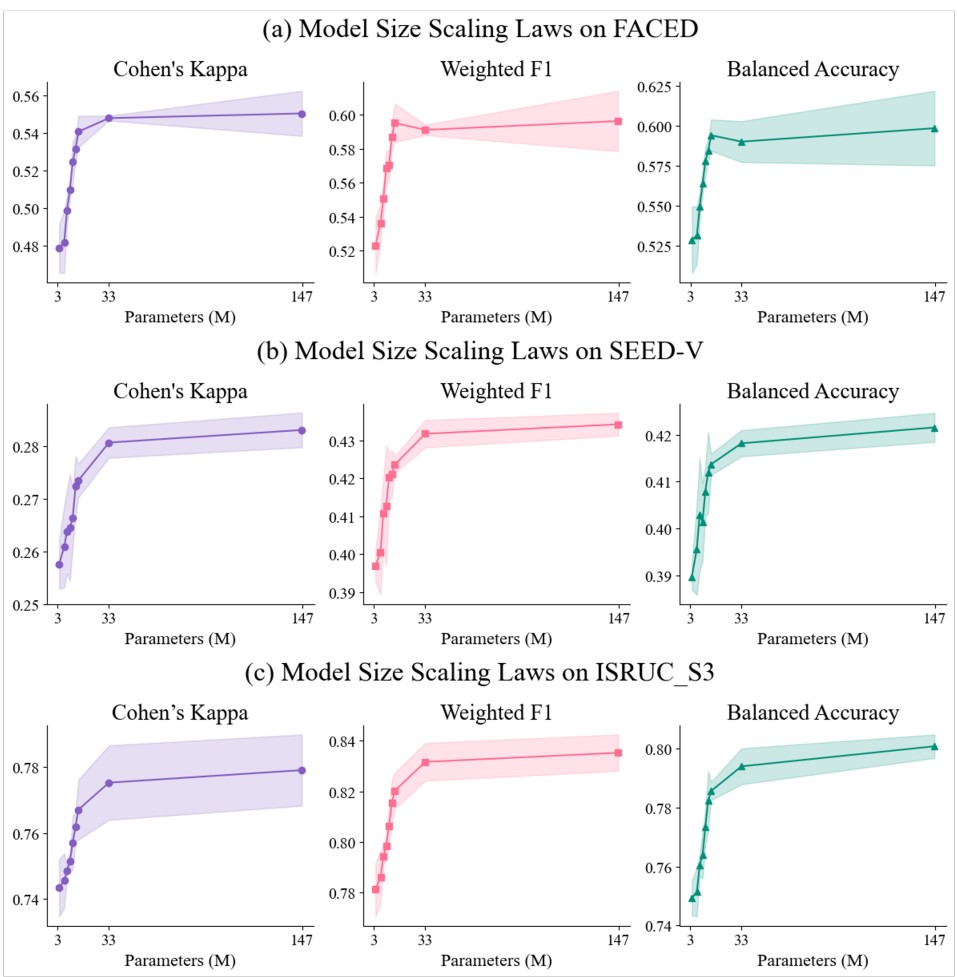

Figure 24: Model size scaling laws on FACED, SEED-V, and ISRUC_S3.

Complementing the scaling analyses on model size and data size presented in the previous subsections, we further examine how computational cost scales with architectural capacity. This provides a detailed breakdown of parameters, FLOPs, throughput, and GPU memory usage across all *CodeBrain* model configurations, allowing a holistic view of the efficiency capacity.

Table 23 shows a smooth increase in parameters, FLOPs, and memory consumption as model depth and hidden size grow. Notably, throughput decreases sub-linearly with scale, demonstrating that the architecture maintains high computational efficiency even at larger capacities.

Lightweight variants (3-6 layers) offer high throughput (3.5-5 samples/s) and low memory usage ($< 8$ GB), suitable for real-time or resource-constrained EEG applications. Mid-sized models (8-12 layers) provide an excellent balance between performance and efficiency, which aligns with where the accuracy scaling curve begins to saturate. The largest configuration (24 layers, 146M parameters) substantially increases capacity and FLOPs, corresponding to the upper regime of diminishing returns observed in the model-size scaling law. We therefore select the 8-layer configuration as the main model used in our experiments after balancing performance and efficiency.

Overall, these results show that *CodeBrain* follows predictable computational scaling behavior and offers flexible operating points for various deployment budgets.

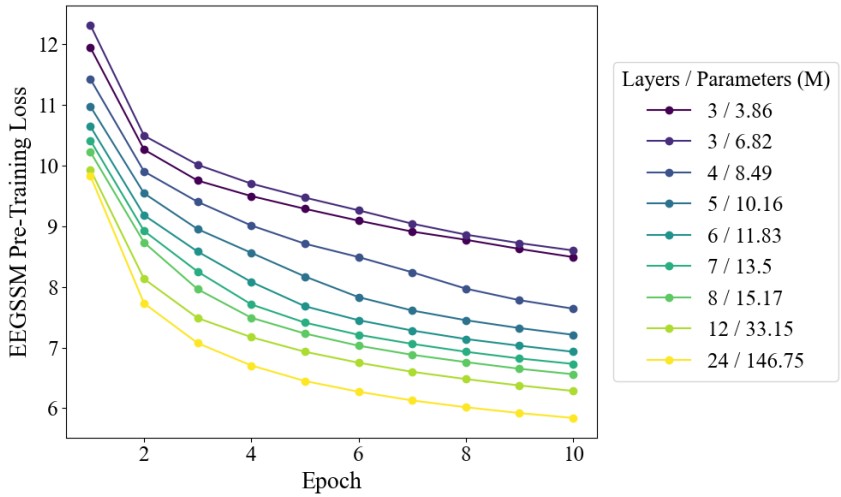

Figure 25: EEGSSM pre-training loss curve for different model size.

Table 24: Comparison under data-limited settings on the FACED dataset (9-class).

| Methods | Cohen's Kappa | Weighted F1 | Balanced Accuracy |
|---|---|---|---|
| *Linear Probing* | | | |
| LaBraM | $0.3026 \pm 0.0121$ | $0.3789 \pm 0.0154$ | $0.3812 \pm 0.0148$ |
| CBraMod | $0.3378 \pm 0.0139$ | $0.4123 \pm 0.0117$ | $0.4146 \pm 0.0123$ |
| CodeBrain | $\mathbf{0.3587} \pm 0.0136$ | $\mathbf{0.4311} \pm 0.0109$ | $\mathbf{0.4327} \pm 0.0127$ |
| *10% Few-Shot* | | | |
| LaBraM | $0.1358 \pm 0.0163$ | $0.2247 \pm 0.0196$ | $0.2265 \pm 0.0174$ |
| CBraMod | $0.1632 \pm 0.0156$ | $0.2595 \pm 0.0138$ | $0.2604 \pm 0.0148$ |
| CodeBrain | $\mathbf{0.1716} \pm 0.0101$ | $\mathbf{0.2599} \pm 0.0104$ | $\mathbf{0.2654} \pm 0.0093$ |
| *30% Few-Shot* | | | |
| BIOT | $0.2573 \pm 0.0346$ | $0.3501 \pm 0.0341$ | $0.3428 \pm 0.0329$ |
| LaBraM | $0.2672 \pm 0.0371$ | $0.3548 \pm 0.0325$ | $0.3513 \pm 0.0315$ |
| CBraMod | $0.3239 \pm 0.0265$ | $0.4056 \pm 0.0256$ | $0.4035 \pm 0.0233$ |
| CodeBrain | $\mathbf{0.3356} \pm 0.0253$ | $\mathbf{0.4114} \pm 0.0225$ | $\mathbf{0.4104} \pm 0.0281$ |
| *Full Finetuning* | | | |
| BIOT | $0.4476 \pm 0.0254$ | $0.5136 \pm 0.0112$ | $0.5118 \pm 0.0118$ |
| LaBraM | $0.4698 \pm 0.0102$ | $0.5288 \pm 0.0188$ | $0.5273 \pm 0.0107$ |
| CBraMod | $0.5041 \pm 0.0122$ | $0.5618 \pm 0.0093$ | $0.5509 \pm 0.0089$ |
| CodeBrain | $\mathbf{0.5406} \pm 0.0084$ | $\mathbf{0.5953} \pm 0.0113$ | $\mathbf{0.5941} \pm 0.0098$ |

## O  LOW-RESOURCE COMPARISON WITH EXISTING METHODS

To evaluate model performance under low-resource conditions, we conduct experiments under three settings: 30% few-shot, 10% few-shot, and linear probing. The 30% and 10% settings reflect data-limited conditions, where only a small portion of labeled training data is available. In contrast, linear probing represents a compute-limited condition where the backbone is frozen, and only a single linear layer is trained. Results for the FACED (9-class) and SEED-V (5-class) datasets are reported in Tables 24 and 25.

Table 25: Comparison under data-limited settings on the SEED-V dataset (5-class).

| Methods | Cohen's Kappa | Weighted F1 | Balanced Accuracy |
|---|---|---|---|
| *Linear Probing* | | | |
| LaBraM | $0.1941 \pm 0.0184$ | $0.3457 \pm 0.0135$ | $0.3413 \pm 0.0144$ |
| CBraMod | $0.2239 \pm 0.0053$ | $0.3823 \pm 0.0041$ | $0.3791 \pm 0.0050$ |
| CodeBrain | $\mathbf{0.2302} \pm 0.0166$ | $\mathbf{0.3889} \pm 0.0154$ | $\mathbf{0.3829} \pm 0.0136$ |
| *10% Few-Shot* | | | |
| LaBraM | $0.0302 \pm 0.0065$ | $0.2194 \pm 0.0079$ | $0.2228 \pm 0.0091$ |
| CBraMod | $0.0174 \pm 0.0029$ | $0.2071 \pm 0.0125$ | $0.2127 \pm 0.0023$ |
| CodeBrain | $\mathbf{0.1690} \pm 0.0170$ | $\mathbf{0.3410} \pm 0.0133$ | $\mathbf{0.3331} \pm 0.0138$ |
| *30% Few-Shot* | | | |
| BIOT | $0.1775 \pm 0.0425$ | $0.3492 \pm 0.0416$ | $0.3505 \pm 0.0375$ |
| LaBraM | $0.2044 \pm 0.0384$ | $0.3700 \pm 0.0321$ | $0.3686 \pm 0.0305$ |
| CBraMod | $0.2291 \pm 0.0246$ | $0.3886 \pm 0.0255$ | $0.3877 \pm 0.0236$ |
| CodeBrain | $\mathbf{0.2376} \pm 0.0284$ | $\mathbf{0.3943} \pm 0.0259$ | $\mathbf{0.3902} \pm 0.0271$ |
| *Full Finetuning* | | | |
| BIOT | $0.2261 \pm 0.0262$ | $0.3856 \pm 0.0203$ | $0.3837 \pm 0.0187$ |
| LaBraM | $0.2386 \pm 0.0209$ | $0.3974 \pm 0.0111$ | $0.3976 \pm 0.0138$ |
| CBraMod | $0.2569 \pm 0.0143$ | $0.4101 \pm 0.0108$ | $0.4091 \pm 0.0097$ |
| CodeBrain | $\mathbf{0.2735} \pm 0.0032$ | $\mathbf{0.4235} \pm 0.0022$ | $\mathbf{0.4137} \pm 0.0023$ |

Across all three settings and both datasets, *CodeBrain* consistently achieves strong performance among the evaluated methods, demonstrating strong data efficiency and reliable transfer behavior under limited conditions. We also observe that few-shot performance is generally lower than linear probing, which is expected because full fine-tuning with very limited labeled data tends to be more sensitive to overfitting and distribution shift, whereas linear probing offers a more stable evaluation by freezing the backbone and relying only on the pretrained representations. This also confirms that the pretrained representations learned by *CodeBrain* already contain meaningful structure. Full-finetuning results under full supervision are also included for reference.

While *CodeBrain* remains comparable to prior EFMs in limited-data and compute-limited conditions, all models exhibit notable performance drops compared to full fine-tuning. EEG signals inherently exhibit extremely low signal-to-noise ratios and substantial domain shift across subjects, devices, montages, and recording setups. Under such conditions, a small number of labeled samples is often insufficient to fully adapt pretrained representations to the target distribution, making low-resource EEG transfer a challenging but important research problem.

## P  LIMITATIONS AND FUTURE WORK

Our work presents promising results but also highlights several limitations that offer directions for future exploration.

First, the interpretability analyses in this paper focus on representation-level structure learned during pretraining. As this paper centers on developing a foundation model, our focus is mainly placed on understanding the representation space learned during pretraining. Future work may incorporate decision-level interpretability during finetuning, for example by exploring sparse codebook selection, task-guided gating mechanisms, or disentangling how temporal and frequency codes contribute to class-specific predictions.

Second, our model and experiments focus on only scalp EEG data. Extending the proposed framework to modalities with richer frequency content, such as intracranial EEG (iEEG), electrocorticography

(ECoG), or stereo-EEG (sEEG) (Chen et al., 2022), offers an opportunity to study how the model behaves under substantially different signal characteristics (Lin et al., 2025b).

These offer promising directions toward building more general and more interpretable brain foundation models that operate across signal modalities and provide both meaningful insights.

## Q  USE OF LLM

In preparing this manuscript, we made use of large language models (LLMs) such as ChatGPT, Gemini, and Deepseek for auxiliary writing support. Specifically, LLMs were employed in three ways:

(i) Polishing the English writing style, including grammar correction and improving fluency;

(ii) Assisting with LaTeX formatting and typesetting to ensure consistent presentation of mathematical equations, tables, and figures.

All core technical content, theoretical results, and experimental findings were designed, implemented, and validated by the authors. LLM usage was restricted to language refinement and formatting assistance, without influencing the originality or validity of the scientific contributions.

