# OpenReview forum: "CodeBrain: Bridging Decoupled Tokenizer and Multi-Scale Architecture for EEG Foundation Model"
_ICLR.cc/2026/Conference — ICLR 2026 Poster_

### Official Review · Reviewer_ATKr · 2025-10-29

**Soundness:** 2
**Presentation:** 3
**Contribution:** 2
**Rating:** 2
**Confidence:** 5

**Summary:**

This paper presents CodeBrain, a two-stage EEG foundation model that aims to achieve both domain-specific interpretability and strong cross-task generalization. The first stage introduces a tokenizer which decouples temporal and frequency EEG features into discrete tokens, effectively enlarging the representation space and improving discriminability while maintaining interpretability linked to neural and spectral events. The second stage proposes a multi-scale SSM combining global convolutional layers and sliding-window attention, designed to capture both long-range dependencies and local neural patterns. CodeBrain demonstrates strong generalization across 8 downstream tasks and 10 datasets under distribution shifts.

**Strengths:**

(1) The authors presented a time-frequency decoupled EEG tokenizer, combined with generative learning with multiscale SSM, with increased computational efficiency, which is somewhat novel especially w.r.t. the decoupled representation.

(2) Strong cross-domain robustness and interpretability visualizations.

(3) High clarity and well-organized figures.

**Weaknesses:**

(1) The overall work, both tokenizer and the generative pretraining, still seem incremental to me. It would be better to discuss and if possible, analyze quantitatively how it substantially improves EEG representation in actual subbands as well as varying, often non-stationary time-frequency landscape. It also fails to convince me how the current work architecturally and mechanistically innovates beyond LaBraM, Beatrix, EEGMamba, etc...  Also, time-frequency representation is not new at all. It seems that the authors still concentrate on Fourier spectrum for the frequency part.

(2) It would be better to consider adding at least one or two supervised baselines as well as contrastive ones for a more comprehensive comparison.

(3) I don't understand why the author completely disregarded gamma oscillations and higher-frequency activities from their analysis. Epileptic seizures, for example, often involve high-frequency outbursts during preictal and interictal stages. Such high-frequency activities are ubiquitous inother physiological activities as well. It is suggested that the authors should perform some ablative experiments to explore how their preprocessing pipeline is justified.

Still, should the authors provide satisfactory clarification or partial resolution of these concerns, I would be open to increasing my score.

**Questions:**

(1) Were the AUROC and AUPRC calculated in a balanced manner?

(2) About channel robustness, have authors ever consider masking specific brain regions to provide insights into the robustness in neuroscientific or clinical settings? Random masking is unconvincing.

(3) The authors used a tripartite splitting in their experimental setup. Why not use subject-independent cross-validation (see experimental setup in FAPEX, BrainWave, Brant, Scatterformer, etc.), which offers more robust statistical estimates while prevents any potential information leakage?

(4) The authors should at least analyze MACs, Throughput/Latency, FLOPs of some of the best-performing baselines along with CodeBrain.

---

> ### Author Response · Authors · 2025-11-21
> **Response to Reviewer ATKr [1/n]**
>
> We sincerely thank you for recognizing our study's **interpretability**, **generalization** and **visualization**.
>
> We address your concerns as follow:
>
> # [W1.1] Clarification on Novelty
>
> We appreciate the chance to further clarify our novelty. Our goal is *not to introduce a new type of frequency representation*. Instead, the key innovations of CodeBrain lie in:
>
> 1. how heterogeneous EEG signals are **stably decoupled** into parallel tokens with domain-specific, representation-level **interpretability**, and
> 2. how the pretraining architecture reflects the brain’s small-world topology by modeling **sparse** long-range dependencies together with **intra-patch patterns**.
>
> Next, we provide a comparison with LaBraM[1], Beatrix[2], and EEGMamba[3]
>
> ## 1) From Frequency-Dominant Solution to Temporal Waveform Interpretability: CodeBrain vs. LaBraM / Beatrix Tokenizers
>
> Since EEGMamba does not include tokenization module, it is not part of this comparison dimension.
>
> Both LaBraM and Beatrix primarily rely on frequency tokenization: LaBraM trains its single mixed codebook only on frequency reconstruction and **reports temporal reconstruction non-convergence**, while Beatrix also trains its tokenizer on frequency features but adds temporal information during fine-tuning. This actually reflects a field-wide gap: **temporal structure is largely absent**.
>
> However, clinicians interpret EEG primarily through temporal waveforms, and domain-aligned interpretability therefore requires robust temporal representations. Our contribution directly targets this gap. By stabilizing the temporal branch via contrastive regularization, CodeBrain introduces the ***first* stable decoupled tokenizer**, enabling domain-specific representation-level interpretability that prior EFMs cannot provide.
>
> In practice, we also observed that frequency-domain codebooks tend to converge more easily; therefore, **we intentionally keep the frequency pathway simple** and focus our design on ensuring a stable temporal branch, preventing the model to collapse into a frequency-dominant solution.
>
> ## 2) Multi-Scale Pretraining architecture: CodeBrain vs. LaBraM / Beatrix / EEGMamba
>
> CodeBrain’s pretraining backbone is different from these three works. Our design is inspired from the small-world topology of brain networks, requiring both ***sparse* long-range inter-patch dependencies** and **fine-grained intra-patch neural patterns** overlooked by prior EFMs.
>
> 1. **Inter-patch modeling (global):**
>
>    LaBraM and Beatrix use dense attention in Transformers, while EEGMamba employs a bidirectional Mamba. In contrast, CodeBrain uses a structured SSM with global convolution to directly capture **sparse** long-range connectivity.
>
> 2. **Intra-patch modeling (local):**
>
>    **LaBraM, Beatrix, or EEGMamba did not model intra-patch patterns.** CodeBrain introduces sliding-window attention to learn intra-patch patterns.
>
> In summary, our architecture substantially differs from existing EEG foundation models.
>
> We also appreciate that you have highlighted a broad set of related works throughout your review. We had already discussed several of them in the previous submission, including LaBraM, EEGMamba, BrainWave, and Brant. To further ensure completeness, we have now incorporated the other works you mentioned, **FAPEX**, **Beatrix**, and **Scatterformer**, into the revised *Related Work* section.
>
> [1] Jiang. et al. Large brain model for learning generic representations with tremendous EEG data in BCI. ICLR, 2024
>
> [2] Zheng. et al. Beatrix: Out-of-Distribution Generalization of Large EEG Model via Invariant Contrastive Fine-Tuning. OpenReview, 2025
>
> [3] Gui. et al. EEGMamba: Bidirectional State Space Model with Mixture of Experts for EEG Multi-task Classification. ArXiv, 2024

---

> ### Author Response · Authors · 2025-11-21
> **Response to Reviewer ATKr [2/n]**
>
> # [W1.2] Empirical Evidence for Subband and Non-stationary Modeling
>
> We provide the following quantitative analyses to evaluate whether CodeBrain effectively models subband structure and non-stationary time-frequency dynamics. In the original paper, Appendix B.1 offers a qualitative study of the learned frequency token representation; here we complement it with quantitative results.
>
> ## 1) Subband masking ablation experiments
>
> We mask each of the five EEG frequency bands:  $\delta$ (0.5-4 Hz), $\theta$ (4–8 Hz), $\alpha$ (8–13 Hz), $\beta$ (13–30 Hz), and $\gamma$ (>30 Hz) and conduct experiments on both the emotion recognition dataset (SEED-V) and the sleep staging dataset (ISRUC_S3). The results are presented below.
>
> |                  |       SEED-V        |                     |                       | ISRUC_S3            |                     |                       |
> | ---------------- | :-----------------: | :-----------------: | --------------------- | ------------------- | ------------------- | --------------------- |
> | **Removed Band** |  **Cohen’s Kappa**  |   **Weighted F1**   | **Balanced Accuracy** | **Cohen’s Kappa**   | **Weighted F1**     | **Balanced Accuracy** |
> | **None**         | **0.2735** ± 0.0032 | **0.4235** ± 0.0022 | **0.4137** ± 0.0023   | **0.7671** ± 0.0091 | **0.8202** ± 0.0071 | **0.7856** ± 0.0031   |
> | **$\delta$**     |   0.0297 ± 0.0027   |   0.2075 ± 0.0105   | 0.2201 ± 0.0004       | 0.0225 ± 0.0391     | 0.1751 ± 0.0843     | 0.2043 ± 0.0237       |
> | **$\theta$**     |   0.0142 ± 0.0050   |   0.1780 ± 0.0176   | 0.2098 ± 0.0039       | 0.0728 ± 0.0557     | 0.1386 ± 0.0341     | 0.2812 ± 0.0749       |
> | **$\alpha$**     |   0.0082 ± 0.0070   |   0.1306 ± 0.0536   | 0.2058 ± 0.0049       | 0.1048 ± 0.0221     | 0.1762 ± 0.0186     | 0.3226 ± 0.0293       |
> | **$\beta$**      |   0.0442 ± 0.0120   |   0.2377 ± 0.0068   | 0.2360 ± 0.0095       | 0.2131 ± 0.0660     | 0.2867 ± 0.1185     | 0.3835 ± 0.0446       |
> | **$\gamma$**     |   0.1372 ± 0.0081   |   0.3141 ± 0.0086   | 0.3092 ± 0.0044       | 0.3618 ± 0.0012     | 0.4831 ± 0.0291     | 0.4764 ± 0.0082       |
>
> As shown in the table, removing any single band leads to substantial drop, indicating that the model relies on information distributed across all frequency ranges. Moreover, the pattern of sensitivity is also **consistent with known knowledge**.
>
> - **Emotion Recognition (SEED-V):** Performance drops most when $\delta, \theta, \alpha$ are removed, aligning with prior affective findings where low-frequency dynamics (0–13 Hz) carry key information.
> - **Sleep Staging (ISRUC-S3):** The degradation is most pronounced when removing $\delta$ and $\theta$, which are important for differentiating NREM stages and slow-wave activity.
>
> We have included these analyses in Appendix K.6 of the revised paper.
>
> ## 2) Non-stationary robustness experiments
>
> To further stress-test robustness to non-stationary dynamics, we introduce a linear baseline shift. Real-world EEG is usually non-stationary, and this perturbation adds extra noise beyond naturally occurring non-stationarity. We apply this perturbation to both emotion recognition dataset (SEED-V) and the sleep staging dataset (ISRUC_S3),  and compare our CodeBrain with the strongest baseline (CBraMod[1]). The results are presented below.
>
> |                           |                  |       SEED-V        |                     |                       | ISRUC_S3            |                     |                       |
> | ------------------------- | ---------------- | :-----------------: | :-----------------: | :-------------------: | ------------------- | ------------------- | --------------------- |
> | **Setting**               | **Model**        |  **Cohen’s Kappa**  |   **Weighted F1**   | **Balanced Accuracy** | **Cohen’s Kappa**   | **Weighted F1**     | **Balanced Accuracy** |
> | **Reference (no shift)**  | CodeBrain (Ours) | **0.2735** ± 0.0032 | **0.4235** ± 0.0022 |  **0.4137** ± 0.0023  | **0.7671** ± 0.0091 | **0.8202** ± 0.0071 | **0.7856** ± 0.0031   |
> |                           | CBraMod          |   0.2569 ± 0.0143   |   0.4101 ± 0.0108   |    0.4091 ± 0.0097    | 0.7407 ± 0.0251     | 0.8056 ± 0.0219     | 0.7844 ± 0.0126       |
> | **Linear baseline shift** | CodeBrain (Ours) | **0.2170** ± 0.0345 | **0.3799** ± 0.0276 |  **0.3706** ± 0.0259  | **0.4762** ± 0.0835 | **0.5914** ± 0.0668 | **0.5617** ± 0.0630   |
> |                           | CBraMod          |   0.2027 ± 0.0147   |   0.3658 ± 0.0136   |    0.3628 ± 0.0111    | 0.4242 ± 0.0068     | 0.5478 ± 0.0087     | 0.5268 ± 0.0085       |
>
> These results indicate that CodeBrain learns more robust representations and maintains stronger resilience under non-stationary setting. We have included these analyses in Appendix M.3 of the revised paper.
>
> [1] Wang. et al. CBraMod: A Criss-Cross Brain Foundation Model for EEG Decoding. ICLR, 2025

---

> ### Author Response · Authors · 2025-11-21
> **Response to Reviewer ATKr [3/n]**
>
> # [W2] Baseline Comparison
>
> Thank you for this valuable suggestion. Following your recommendation, we have added 4 widely used non-foundation baseline: **EEGNet**[1], **EEGConformer**[2], **ContraWR**[3] and **ST-Transformer**[4]. These models represent compact CNN, Transformer, and contrastive-learning based architectures that are commonly adopted in EEG decoding tasks.
>
> We also note that the contrastive-based model you highlighted is represented in our original comparisons through **BENDR**[5], which is a contrastive-pretrained *foundation model*. The newly added ContraWR thus complements this by providing a *non-foundation* contrastive baseline.
>
> Below, we provide the performance comparison on all 10 downstream datasets used in our paper. These results are also reported in Table 2 and Appendix J of the revised paper. Across all ten downstream datasets, CodeBrain consistently achieves higher performance than these non-foundation baselines.
>
> |||FACED|||SEED-V||
> | -------------------- | :-------------------: | :-----------------: | :-------------------: | :-----------------: | :-----------------: | :-------------------: |
> |**Model**|**Cohen’s Kappa**|**Weighted F1**|**Balanced Accuracy**|**Cohen’s Kappa**|**Weighted F1**| **Balanced Accuracy**|
> |EEGNet|0.3342 ± 0.0251|0.4124 ± 0.0141|0.4090 ± 0.0122|0.1006 ± 0.0143|0.2749 ± 0.0098|0.2961 ± 0.0102|
> |EEGConformer|0.3858 ± 0.0186|0.4514 ± 0.0107|0.4559 ± 0.0125|0.1772 ± 0.0174|0.3487 ± 0.0136|0.3537 ± 0.0112|
> |ContraWR|0.4231 ± 0.0151|0.4887 ± 0.0078|0.4887 ± 0.0078|0.1905 ± 0.0188|0.3544 ± 0.0121| 0.3546 ± 0.0105|
> | ST-Transformer|0.4137 ± 0.0133|0.4795 ± 0.0096|0.4810 ± 0.0079|0.1083 ± 0.0121|0.2833 ± 0.0105|0.3052 ± 0.0072|
> | **CodeBrain (Ours)** |  **0.5406** ± 0.0084  | **0.5953** ± 0.0113 |  **0.5941** ± 0.0098  | **0.2735** ± 0.0032 | **0.4235** ± 0.0022 |  **0.4137** ± 0.0023  |
> |||**ISRUC_S3**|||**BCIC 2020-T3**||
> |**Model**|**Cohen’s Kappa**|**Weighted F1**|**Balanced Accuracy**|**Cohen’s Kappa**|**Weighted F1**| **Balanced Accuracy**|
> | EEGNet|0.7396 ± 0.0155|0.7407 ± 0.0184|0.7121 ± 0.0134|0.4413 ± 0.0102|0.3016 ± 0.0123|    0.4413 ± 0.009|
> | EEGConformer|0.7482 ± 0.0164|0.7501 ± 0.0211|0.7212 ± 0.0181|0.4488 ± 0.0154|0.3133 ± 0.0183|0.4506 ± 0.0133|
> | ContraWR|0.7493 ± 0.0150|0.7513 ± 0.0185|0.7226 ± 0.0164|   0.4407 ± 0.0182|0.3078 ± 0.0218|0.4257 ± 0.0162|
> | ST-Transformer|0.7388 ± 0.0195|0.7399 ± 0.0223|0.7116 ± 0.0197|0.4247 ± 0.0138|0.2941 ± 0.0159|    0.4126 ± 0.0122|
> | **CodeBrain (Ours)** |  **0.7671** ± 0.0091  | **0.8202** ± 0.0071 |  **0.7856** ± 0.0031  | **0.5127** ± 0.0065 | **0.6101** ± 0.0053 |  **0.6101** ± 0.0052  |
> |||**Mental Arithmetic**|||**CHB-MIT**||
> |**Model**|**AUROC**|**AUC-PR**| **Balanced Accuracy** |**AUROC**|**AUC-PR**| **Balanced Accuracy**|
> |EEGNet| 0.7321 ± 0.0108|0.5763 ± 0.0102|0.6770 ± 0.0116|0.8048 ± 0.0136|0.1914 ± 0.0182|0.5658 ± 0.0106|
> |EEGConformer|0.7424 ± 0.0128|   0.5829 ± 0.0134|0.6805 ± 0.0123|0.8226 ± 0.0170|0.2209 ± 0.0215|0.5976 ± 0.0141|
> |ContraWR|0.7332 ± 0.0082|0.5787 ± 0.0164| 0.6631 ± 0.0097|   0.8103 ± 0.0144|0.2279 ± 0.0183|0.6351 ± 0.0122    |
> |ST-Transformer|0.7132 ± 0.0174|0.5672 ± 0.0259|0.6631 ± 0.0173|0.8237 ± 0.0491   |0.1422 ± 0.0094|0.5915 ± 0.0195|
> |**CodeBrain (Ours)**|**0.8707** ± 0.0209|**0.7177** ± 0.0421|**0.7514** ± 0.0203|**0.8961** ± 0.0174|**0.4377** ± 0.0288|**0.7273** ± 0.0240|
> |||**ISRUC_S1**|||**SHU-MI**||
> | **Model**|**Cohen’s Kappa**|**Weighted F1**|**Balanced Accuracy** |**AUROC**|**AUC-PR**| **Balanced Accuracy**|
> | EEGNet|0.7040 ± 0.0173|0.7513 ± 0.0124|0.7154 ± 0.0121|0.6283 ± 0.0152|0.6311 ± 0.0142|0.5889 ± 0.0177|
> | EEGConformer|0.7143 ± 0.0162|0.7634 ± 0.0151|0.7400 ± 0.0133|0.6351 ± 0.0101|0.6370 ± 0.0093|0.5900 ± 0.0107|
> | ContraWR|0.7178 ± 0.0156|0.7610 ± 0.0137|0.7402 ± 0.0126|0.6273 ± 0.0113|0.6315 ± 0.0105|0.5873 ± 0.0128|
> | ST-Transformer|0.7013 ± 0.0352|0.7681 ± 0.0175|0.7381 ± 0.0205|0.6431 ± 0.0111|0.6394 ± 0.0122|0.5992 ± 0.0206|
> | **CodeBrain (Ours)** |**0.7476** ± 0.0040|**0.8020** ± 0.0018|**0.7835** ± 0.0033|**0.7124** ± 0.0050|**0.7166** ± 0.0106|**0.6431** ± 0.0066|
> |||**TUEV**|||**TUAB**||
> |**Model**|**Cohen’s Kappa**|**Weighted F1**|**Balanced Accuracy**|**AUROC**|**AUC-PR**|**Balanced Accuracy**|
> |EEGNet|0.3577 ± 0.0155|0.6539 ± 0.0120|0.3876 ± 0.0143|0.7642 ± 0.0036|0.8299 ± 0.0043|0.8412 ± 0.0031|
> |EEGConformer|0.3967 ± 0.0195|0.6983 ± 0.0152|0.4074 ± 0.0164|0.7758 ± 0.0049|0.8427 ± 0.0054|0.8445 ± 0.0038|
> |ContraWR|0.3912 ± 0.0237|0.6893 ± 0.0136|0.4384 ± 0.0349|0.7746 ± 0.0041|0.8421 ± 0.0104|0.8456 ± 0.0074|
> |ST-Transformer|0.3765 ± 0.0306|0.6823 ± 0.0190|0.3984 ± 0.0228|0.7966 ± 0.0023|0.8521 ± 0.0026|0.8707 ± 0.0019|
> |**CodeBrain (Ours)**|**0.6912** ± 0.0101|**0.8362** ± 0.0048|**0.6428** ± 0.0062| **0.9030** ± 0.0009|**0.9100** ± 0.0006|**0.8294** ± 0.0013|

---

> ### Author Response · Authors · 2025-11-21
> **Response to Reviewer ATKr [4/n]**
>
> # [W3] Justification of Preprocessing Pipeline
>
> We would like to clarify that our preprocessing **does not discard gamma-band activity**. The commonly used definition of scalp-EEG $\gamma$ oscillations spans 30–80 Hz, and our 0.3–75 Hz filter fully preserves this frequency range.
>
> ### 1) Using a 75 Hz upper cutoff is established community practice in scalp EEG foundation models.
>
> Most existing scalp EEG foundation models apply bandpass filters with an upper cutoff of 75 Hz [1-4] during preprocessing for the pretraining dataset, so we strictly follows this practice. We additionally conducted an ablation study in which the high-frequency filter was removed. Across both the SEED-V emotion recognition dataset and the ISRUC_S3 sleep staging dataset, we consistently observed that **retaining high-frequency components degraded performance**. The results are presented below.
>
> |                                |     **SEED-V**      |                     |                       | **ISRUC_S3**        |                     |                       |
> | ------------------------------ | :-----------------: | :-----------------: | --------------------- | ------------------- | ------------------- | --------------------- |
> | **preprocessing  setting**     |  **Cohen’s Kappa**  |   **Weighted F1**   | **Balanced Accuracy** | **Cohen’s Kappa**   | **Weighted F1**     | **Balanced Accuracy** |
> | **75 Hz upper-band filtering** | **0.2735** ± 0.0032 | **0.4235** ± 0.0022 | **0.4137** ± 0.0023   | **0.7671** ± 0.0091 | **0.8202** ± 0.0071 | **0.7856** ± 0.0031   |
> | **no upper-band filtering**    |   0.2651 ± 0.0041   |   0.4168 ± 0.0057   | 0.4096 ± 0.0065       | 0.7542 ± 0.0124     | 0.8089 ± 0.0112     | 0.7730 ± 0.0106       |
>
> There are two main reasons why the 75 Hz cutoff is appropriate for a scalp-EEG foundation model:
>
> - **Most downstream scalp-EEG tasks depend primarily on low–mid frequency activity (0-40 Hz).**
>   Tasks such as emotion recognition, sleep staging, and a broad range of BCI tasks (such as Motor Imagery) mainly rely on $\delta-\beta$ bands. These constitute the dominant downstream applications for scalp-EEG foundation models.
> - **High-frequency (>80 Hz) activity is poorly represented in scalp EEG.**
>   Due to volume conduction and the low signal-to-noise ratio of scalp recordings at higher frequencies, retaining the >75 Hz band tends to introduce more noise than meaningful information.
>
> ### 2) Scalp EEG inherently has limited sensitivity to high-frequency oscillations, which is an limitation of the recording modality rather than a modeling choice.
>
> This limitation is particularly evident in scalp-EEG epilepsy datasets, where the sampling rate inherently restricts the recoverable frequency range (*According to the Nyquist theorem, the highest frequency that can be faithfully reconstructed is half of the sampling rate*). For example, the most widely used scalp-EEG epilepsy dataset,  CHB-MIT, which is also one of our downstream tasks, has a sampling rate of 256 Hz, implying a Nyquist limit of 128 Hz. High-frequency oscillations beyond this range are **inherently unrecoverable in scalp EEG**. Our goal is to build a scalp EEG foundation model, and within the frequency range that scalp recordings can reliably support, our model already achieves strong performance across tasks, **including CHB-MIT epilepsy detection**.
>
> ### 3) We have added limitation to the paper that our model focuses on scalp EEG.
>
> To prevent potential misunderstanding, we have added a statement in the Appendix P *Limitations and Future Work* section of the revised paper clarifying that our work focuses on scalp EEG. Modeling high-frequency activity such as HFOs lies outside this scope and typically requires different data modalities, including intracranial EEG (iEEG), electrocorticography (ECoG), or stereo-EEG (sEEG), which offer substantially higher resolution.
>
> [1] Jiang. et al. Large brain model for learning generic representations with tremendous EEG data in BCI. ICLR, 2024
>
> [2] Wang. et al. CBraMod: A Criss-Cross Brain Foundation Model for EEG Decoding. ICLR, 2025
>
> [3] Jiang. et al. NeuroLM: A universal multi-task foundation model for bridging the gap between language and EEG signals. ICLR, 2025
>
> [4] Zhou. et al. CSBrain: A Cross-scale Spatiotemporal Brain Foundation Model for EEG Decoding. NeurIPS, 2025

---

> ### Author Response · Authors · 2025-11-21
> **Response to Reviewer ATKr [5/n]**
>
> # [Q1] Clarification on Metric
>
> In our paper, AUROC and AUPRC are only used for binary classification tasks. To the best of our knowledge, there is no standard *balanced* variant of these metrics in the binary setting. AUROC is typically used when classes are relatively balanced, while AUPRC is more sensitive to class imbalance. To explicitly account for imbalance, we additionally report Balanced Accuracy.
>
> These evaluation metrics for binary task follow the standard practice in existing scalp-EEG foundation models such as LaBraM[1], EEGPT[2], and CBraMod[3].
>
> # [Q2] Region-Based Channel Robustness Experiments
>
> While random channel masking simulates *incidental electrode dropout* commonly seen in hospitals, it ignores that EEG channels are spatially correlated. To provide a neuroscientifically and clinically meaningful robustness analysis, we conducted region masking on the dataset with the most channels (SEED-V, 62 channels) and the dataset with the fewest channels (ISRUC-S3, 6 channels), covering reference-electrode failures, lateralized dropout, and lobe-specific artefacts. We also compared our model with the strongest baseline, CBraMod[3].
>
> ## 1) Region-based ablations on the high-density SEED-V dataset (62 channels)
>
> We designed 9 meaningful masks, including:
>
> - **Reference/hemisphere failures** (midline-only; left-hemisphere masked; right-hemisphere masked), simulating partial cap detachment or unstable reference electrodes
> - **Lobe-level dropout**: occipital (poor visual-area contact), frontal (muscle/EMG noise), temporal (chewing/face motion contamination), central (movement/EMG), frontocentral block and a parietal (attention modulation)
>
> The results are presented below.
>
> | Mask Setting (Region)     | Model                | Cohen’s Kappa       | Weighted F1         | Balanced Accuracy   |
> | ------------------------- | -------------------- | ------------------- | ------------------- | ------------------- |
> | **Baseline (no mask)**    | CBraMod              | 0.2569 ± 0.0143     | 0.4101 ± 0.0108     | 0.4091 ± 0.0097     |
> |                           | **CodeBrain (Ours)** | **0.2735** ± 0.0032 | **0.4235** ± 0.0022 | **0.4137** ± 0.0023 |
> | **Only keep midline**     | CBraMod              | 0.0114 ± 0.0098     | **0.1740** ± 0.0121 | 0.2077 ± 0.0018     |
> |                           | **CodeBrain (Ours)** | **0.0201** ± 0.0098 | 0.1610 ± 0.0087     | **0.2169** ± 0.0067 |
> | **Left hemisphere mask**  | CBraMod              | 0.0672 ± 0.0260     | 0.2505 ± 0.0227     | 0.2541 ± 0.0226     |
> |                           | **CodeBrain (Ours)** | **0.0956** ± 0.0030 | **0.2779** ± 0.0051 | **0.2750** ± 0.0030 |
> | **Right hemisphere mask** | CBraMod              | 0.0586 ± 0.0171     | 0.2500 ± 0.0214     | 0.2466 ± 0.0149     |
> |                           | **CodeBrain (Ours)** | **0.0678** ± 0.0195 | **0.2540** ± 0.0066 | **0.2569** ± 0.0166 |
> | **Occipital mask**        | CBraMod              | 0.1316 ± 0.0217     | 0.3066 ± 0.0167     | 0.3068 ± 0.0174     |
> |                           | **CodeBrain (Ours)** | **0.2258** ± 0.0067 | **0.3851** ± 0.0040 | **0.3818** ± 0.0090 |
> | **Frontal mask**          | CBraMod              | 0.0632 ± 0.0222     | 0.2486 ± 0.0177     | 0.2491 ± 0.0189     |
> |                           | **CodeBrain (Ours)** | **0.1255** ± 0.0090 | **0.3025** ± 0.0080 | **0.2976** ± 0.0046 |
> | **Temporal mask**         | CBraMod              | 0.1571 ± 0.0284     | 0.3309 ± 0.0222     | 0.3265 ± 0.0227     |
> |                           | **CodeBrain (Ours)** | **0.2311** ± 0.0008 | **0.3922** ± 0.0010 | **0.3824** ± 0.0004 |
> | **Central mask**          | CBraMod              | 0.1562 ± 0.0419     | 0.3305 ± 0.0336     | 0.3247 ± 0.0321     |
> |                           | **CodeBrain (Ours)** | **0.2410** ± 0.0078 | **0.3997** ± 0.0067 | **0.3894** ± 0.0055 |
> | **Frontocentral mask**    | CBraMod              | **0.1304** ± 0.0304 | **0.3077** ± 0.0263 | **0.3039** ± 0.0244 |
> |                           | **CodeBrain (Ours)** | 0.1254 ± 0.0090     | 0.3025 ± 0.0080     | 0.2976 ± 0.0047     |
> | **Parietal mask**         | CBraMod              | 0.1571 ± 0.0284     | 0.3309 ± 0.0222     | 0.3265 ± 0.0227     |
> |                           | **CodeBrain (Ours)** | **0.2311** ± 0.0008 | **0.3922** ± 0.0010 | **0.3824** ± 0.0004 |
>
> Across most settings, CodeBrain consistently outperforms CBraMod.

---

> ### Author Response · Authors · 2025-11-21
> **Response to Reviewer ATKr [6/n]**
>
> ## 2) Region-based ablations on the minimal-montage ISRUC-S3 dataset (6 channels)
>
> Given the A1/A2-referenced 6-channel montage, we designed 5 meaningful masks, including:
>
> - **Reference-electrode failures** (masking A1 or A2), corresponding to left/right hemisphere loss
> - **Lobe-level dropout**: given the symmetric 6-channel montage (occipital, central, frontal on each hemisphere), we exhaustively masked each region
>
> The results are presented below.
>
> | Mask Setting (Region)     | Model                | Cohen’s Kappa       | Weighted F1         | Balanced Accuracy   |
> | ------------------------- | -------------------- | ------------------- | ------------------- | ------------------- |
> | **Baseline (no mask)**    | CBraMod              | 0.7407 ± 0.0251     | 0.8056 ± 0.0219     | 0.7844 ± 0.0126     |
> |                           | **CodeBrain (Ours)** | **0.7671** ± 0.0091 | **0.8202** ± 0.0071 | **0.7856** ± 0.0031 |
> | **Keep right hemisphere** | CBraMod              | 0.6616 ± 0.0263     | 0.7392 ± 0.0292     | 0.6706 ± 0.0212     |
> |                           | **CodeBrain (Ours)** | **0.7430** ± 0.0131 | **0.8047** ± 0.0086 | **0.7760** ± 0.0279 |
> | **Keep left hemisphere**  | CBraMod              | 0.6198 ± 0.1127     | 0.6964 ± 0.0958     | 0.6838 ± 0.0815     |
> |                           | **CodeBrain (Ours)** | **0.7318** ± 0.0252 | **0.7989** ± 0.0196 | **0.7819** ± 0.0133 |
> | **Occipital mask**        | CBraMod              | 0.7089 ± 0.0412     | 0.7598 ± 0.0321     | 0.7340 ± 0.0269     |
> |                           | **CodeBrain (Ours)** | **0.7447** ± 0.0226 | **0.8065** ± 0.0187 | **0.7824** ± 0.008  |
> | **Central mask**          | CBraMod              | 0.6725 ± 0.0106     | 0.7526 ± 0.0074     | 0.6979 ± 0.0164     |
> |                           | **CodeBrain (Ours)** | **0.7479** ± 0.0253 | **0.8111** ± 0.0197 | **0.7790** ± 0.0155 |
> | **Frontal mask**          | CBraMod              | 0.7134 ± 0.0245     | 0.7796 ± 0.0184     | 0.7250 ± 0.0327     |
> |                           | **CodeBrain (Ours)** | **0.7295** ± 0.0531 | **0.7936** ± 0.0408 | **0.7825** ± 0.0282 |
>
>
> Under this highly constrained setup, CodeBrain consistently remains stronger performance.
>
> These region masking experiments demonstrate that CodeBrain maintains strong performance under structured channel dropout. We have added the full results and corresponding analyses to Appendix M.2 of the revised paper.
>
> [1] Jiang. et al. Large brain model for learning generic representations with tremendous EEG data in BCI. ICLR, 2024
>
> [2] Wang. et al. Eegpt: Pretrained transformer for universal and reliable representation of eeg signals. NeurIPS, 2024
>
> [3] Wang. et al. CBraMod: A Criss-Cross Brain Foundation Model for EEG Decoding. ICLR, 2025

---

> ### Author Response · Authors · 2025-11-21
> **Response to Reviewer ATKr [7/n]**
>
> # [Q3] Subject-Independent Cross-Validation
>
> We first confirm that except for the small SEED-V dataset, all experiments in our study use **subject-independent** splits, ensuring a fair comparison across models.
>
> ### 1) Our evaluation contains no information leakage and follows the community-standard and official subject-independent splits
>
> Our evaluation strictly follows the **publicly released data-splitting scripts from CBraMod**[1], the first comprehensive and reproducible benchmark for EEG foundation models. These splits have also been adopted by many subsequent scalp-EEG foundation models, like CSBrain[2], and using them provides a stable, widely accepted evaluation protocol.
>
> For datasets that come with **official subject-independent splits**, such as BCIC2020-T3, TUEV, and TUAB, we directly follow their released protocols.
>
> Overall, 9 out of 10 downstream datasets in our evaluation follow subject-independent splits, ensuring no information leakage.
>
> ### 2) Additional 10-fold subject-independent cross-validation on ISRUC_S3
>
> To further assess statistical robustness, we additionally conducted subject-independent 10-fold cross-validation on ISRUC_S3 following your suggestion, which contains only 10 subjects and is therefore the dataset where k-fold CV typically exhibits the largest variance. Even under this most variance-sensitive setting, CodeBrain could achieve strong performance.
>
> | Model                |    Cohen’s Kappa    |     Weighted F1     |  Balanced Accuracy  |
> | -------------------- | :-----------------: | :-----------------: | :-----------------: |
> | EEGNet               |   0.7207 ± 0.0831   |   0.7462 ± 0.0727   |   0.7270 ± 0.0919   |
> | EEGConformer         |   0.7325 ± 0.0950   |   0.7576 ± 0.0734   |   0.7381 ± 0.0946   |
> | ContraWR             |   0.7324 ± 0.0816   |   0.7598 ± 0.0718   |   0.7355 ± 0.0939   |
> | ST-Transformer       |   0.7199 ± 0.0891   |   0.7444 ± 0.0756   |   0.7255 ± 0.0952   |
> | BENDR                |   0.5846 ± 0.0837   |   0.6854 ± 0.0695   |   0.6531 ± 0.0860   |
> | BIOT                 |   0.6989 ± 0.0825   |   0.7929 ± 0.0639   |   0.7727 ± 0.0894   |
> | LaBraM               |   0.7025 ± 0.0838   |   0.7948 ± 0.0752   |   0.7756 ± 0.0897   |
> | EEGPT                |   0.6021 ± 0.1552   |   0.6450 ± 0.1165   |   0.6839 ± 0.1106   |
> | CBraMod              |   0.7218 ± 0.0937   |   0.8111 ± 0.0772   |   0.7933 ± 0.0891   |
> | **CodeBrain (Ours)** | **0.7452** ± 0.0847 | **0.8237** ± 0.0684 | **0.7945** ± 0.0876 |
>
> In summary, our evaluation protocol strictly prevents information leakage, follows established community standards and official dataset splits, employs subject-independent evaluation for 9 of 10 datasets, and is further validated by cross-subject 10-fold cross-validation on ISRUC_S3 compared with baselines.
>
>
> # [Q4] Efficiency Analysis
>
> Following your suggestion, we report stable and hardware-agnostic metrics, including model parameters, MACs, and FLOPs, for CodeBrain and all EEG foundation-model baselines. We would like to note that inference latency and throughput can vary substantially with instantaneous hardware load, making direct cross-model comparisons less reproducible.
>
>
> | Model     |  MACs  |  Params  |  FLOPs  |
> | :-------- | :----: | :------: | :-----: |
> | BENDR     | 12.51G | 959.84 M | 25.02 G |
> | BIOT      | 255 M  |  3.2 M   |  510 M  |
> | LaBarM    | 0.67 G |  6.02 M  | 1.34 G  |
> | CBraMod   | 0.64 G |  4.03 M  | 1.29 G  |
> | EEGPT     | 4.89 G | 25.24 M  | 9.79 G  |
> | CodeBrain | 4.37 G | 15.17 M  | 8.74 G  |
>
> These metrics provide a fair evaluation of computational efficiency across models. Overall, CodeBrain achieves a balanced compute-performance trade-off. We have added these results in Appendix L of the revised paper.
>
> [1] Wang. et al. CBraMod: A Criss-Cross Brain Foundation Model for EEG Decoding. ICLR, 2025
>
> [2] Zhou. et al. CSBrain: A Cross-scale Spatiotemporal Brain Foundation Model for EEG Decoding. NeurIPS, 2025
>
> *Thank you once again for your thorough and thoughtful review. We hope our clarifications and additional analyses address your concerns. We look forward to your feedback, and are happy to provide further information if needed.*
>
> Best regards,
>
> The Authors

---

> > ### Comment · Reviewer_ATKr · 2025-11-21
> >
> > The authors largely addressed my concerns. The score will be raised to 4.

---

> > > ### Author Response · Authors · 2025-11-21
> > > **Appreciation for Reviewer ATKr’s Follow-up**
> > >
> > > Dear Reviewer ATKr,
> > >
> > > **Thank you very much for your prompt follow-up and for your willingness to raise the score.**
> > >
> > > We sincerely appreciate the time you invested in our work and are glad that our rebuttal has resolved some of your concerns. *If there are any remaining issues, we would be very happy to elaborate further or provide additional analyses to improve the work.*
> > >
> > > Thank you again for your constructive engagement during the discussion.
> > >
> > > Best regards,
> > >
> > > The Authors

---

> > > ### Author Response · Authors · 2025-11-25
> > > **Follow-up to Reviewer ATKr**
> > >
> > > Dear Reviewer ATKr,
> > >
> > > Thank you again for your earlier positive follow-up.
> > >
> > > **We are eager to discuss any remaining concerns you may have, and would be grateful if you could let us know.** We have had constructive exchanges with other reviewers, which has been very helpful in strengthening our paper. We would likewise appreciate any further comments you may wish to share.
> > >
> > > **We also truly appreciate your willingness to update your assessment.** We noticed that the other reviewers have completed their updates with positive evaluations, and we would be happy to provide any additional information that might help you finalize yours as well.
> > >
> > > Thank you again for your thoughtful engagement. We look forward to your further response.
> > >
> > > Best regards,
> > >
> > > The Authors

---

### Official Review · Reviewer_a5Jx · 2025-10-31

**Soundness:** 3
**Presentation:** 4
**Contribution:** 3
**Rating:** 4
**Confidence:** 4

**Summary:**

This study presents an interpretable EEG foundation model, CodeBrain, which integrates both local and global information. Selected codes deriving from TFDual-Tokenizer contain strong clinical meaning and could serve as an effective “EEG vocabulary”, which seems to improve the interpretability. Further, the authors conduct extensive experiments to validate the effectiveness of nearly every component of the proposed framework.

**Strengths:**

1. The authors propose a domain-specific and interpretable TFDual-Tokenizer, which decomposes raw EEG signals into frequency and temporal domains to generate dual-domain tokens, thereby enhancing the representation capability of the model.
2. The proposed EEGSSM architecture integrates multi-scale information and enhances the representation of intra-patch features through a sliding-window attention mechanism.
3. In addition to extensive empirical experiments, the study provides strong theoretical support for the proposed methods.

**Weaknesses:**

1. Although the selected codes in Appendix B are associated with clinically meaningful patterns, the criteria for their selection and their specific contribution to model performance remain unclear. While these codes are intuitively appealing, the paper does not provide concrete evidence demonstrating their impact on improving downstream task performance. Without such evidence, the explanation may not be important for model decision in the down-stream tasks and could potentially mislead researchers, as model reasoning does not necessarily align with human intuition. Moreover, the analysis of class-specific token ratios does not convincingly support a relationship between the proposed interpretable codes and performance gains. In a word, no evidence directly shows that those interpretable codes play a primary role in the model’s decision-making process.

2. If my understanding is correct, in Equation (17) defining the dominance ratio, $max_y N_c^{(y)}$ should always be smaller than $\sum_{y} N_c^{(y)}$ since the latter item $\sum_{y} N_c^{(y)} = max_y N_c^{(y)} + other  N_c^{(y)}$. Therefore, the dominance ratio would be less than 1. Could the authors clarify why they define Dominance(c) > 1 as the criterion for identifying class-specific tokens?
3. Duplication in Figure 3, the left and middle images are identical.

**Questions:**

1. According to the paper, the authors pretrain several versions of the EFM. I recommend including some main versions and their corresponding model sizes in the main text rather than only in the Appendix. Initially, it was unclear which model version the reported results referred to in the main content, forcing readers to cross-check details in the Appendix. Presenting this information earlier would improve clarity and readability.
2. Regarding the TFDual-Tokenizer, have the authors considered a sparse code strategy based on a gate mechanism, where only the frequency, temporal, or occasionally dual code is selected for a given patch? $N^2$ representation space may introduce some redundancy, as suggested by the unused token experiments.

I will remain active during the rebuttal and may consider adjusting my score based on the authors’ responses.

---

> ### Author Response · Authors · 2025-11-21
> **Response to Reviewer a5Jx [1/n]**
>
> We sincerely thank you for recognizing our study's **novelty**, **theoretical support** and **comprehensive validation**.
>
> We address your concerns as follow:
>
> # [W1] Clarification on the Interpretable Codes
>
> Thank you for raising this important concern. Because our paper focuses on **building a foundation model**, our interpretability analyses are intentionally positioned at the level of *representation structure* learned during pretraining, rather than task-specific *decision-level* reasoning. To avoid any potential misunderstanding, we have systematically revised the paper so that:
>
> 1. all mentions of interpretability are explicitly claimed as **representation-level**
> 2. all links to neurophysiological patterns are presented as observations about the structures captured by the learned codebooks, without implying any causal relevance or downstream discriminative influence.
>
> Below we break down your issue into four parts for detailed clarification.
>
> ## 1) Selection Criteria for Interpretable Codes
>
> The interpretable codes are selected using **quantitative token-activation statistics** on the ISRUC\_S3 test set. Using the frozen *TFDual-Tokenizer*, we identify class-specific tokens, focusing on N2 and N3 where neural events are most distinct, and choose the most frequently activated ones for analysis. These codes were independently reviewed by sleep physicians, and we report the proportion judged clinically meaningful. Representative examples are shown in Figure 5–9. We have explicitly included this criterion in AppendixB.1 and B.2 of the revised paper.
>
> ## 2) Additional Evidence on Codebook Contribution
>
> To provide additional evidence on the contribution of the codebook, we conducted two pretraining ablations on the ISRUC_S3 dataset, beyond the comparisons presented in Table 3, where the decoupled design outperforms temporal-only, frequency-only, and mixed codebooks. These additional ablations aim to examine whether removing tokenization entirely or masking high-activation tokens affects the quality of the learned representation. We compare the following settings:
>
> - **Raw-Signal Reconstruction**
>   We disable the tokenizer and train EEGSSM directly to reconstruct raw signals, thereby removing the discretization step entirely.
> - **Pretraining with masked codebook**
>   For ISRUC_S3, we identify the top 50% most frequently activated tokens in both codebooks. During pretraining, whenever a segment is assigned one of these tokens, we replace it with a placeholder. This masks a portion of the vocabulary and prevents the model from using these codes during representation learning.
>
> The downstream results on ISRUC_S3 dataset are shown below:
>
> |Setting|    Cohen’s Kappa    |     Weighted F1     |  Balanced Accuracy  |
> | ------------------------- | :-----------------: | :-----------------: | :-----------------: |
> |TFDual_tokenizer| **0.7671** ± 0.0091 | **0.8202** ± 0.0071 | **0.7856** ± 0.0031 |
> |Raw-Signal Reconstruction|   0.7503 ± 0.0087   |   0.8014 ± 0.0079   |   0.7763 ± 0.0048   |
> |Masked Codebook|   0.7426 ± 0.0102   |   0.7931 ± 0.0084   |   0.7690 ± 0.0063   |
>
> Although these experiments do not suggest that individual interpretable codes directly drive downstream decisions, they demonstrate that the decoupled tokenizer encodes useful structure that supports effective representation learning during pretraining. We include these additional experiments in Appendix B.4.
>
> ## 3) Our intent for interpretability
>
> We appreciate your concern about possible misinterpretation and take it seriously. We clarify that our interpretability aims at the **representation level**: all analyses examine how the tokenizer structures the latent space learned during pretraining, not the reasoning chain of the downstream classifier. We have therefore revised the paper to align our positioning with prior work on foundation models, where interpretability typically focuses on understanding the general representation structure learned during pretraining[1,2]. To prevent over-interpretation, we have stated this discussion in Appendix P *Limitation and Future Work*.
>
> ## 4) Intent of class-specific token-ratio analysis
>
> In line with our overall positioning of interpretability at the *representation level*, the class-specific token-ratio analysis is intended to examine how the **decoupled tokenizer** induces better structured token usage patterns to support our motivation for decoupling. Importantly, this analysis is performed on **unseen downstream datasets**, illustrating that the tokenizer trained on large-scale pretraining data learns generalizable representational structures. We have also revised AppendixB.3 to clarify this intent and kept the description focused on structured representation patterns rather than class discrimination.
>
> [1] Tenney. et al. BERT Rediscovers the Classical NLP Pipeline. ACL, 2019
>
> [2] Jin. et al. Reading Your Heart: Learning ECG Words and Sentences via Pre-training ECG Language Model. ICLR, 2025

---

> ### Author Response · Authors · 2025-11-21
> **Response to Reviewer a5Jx [2/n]**
>
> # [W2] Correction of Equation Typo
>
> Thank you for your careful reading! Your understanding is correct. Our intention was to use $Dominance(c) \geq \tau$ as the selection criterion, and we set $\tau = 1$ to obtain the most strict definition of class-specific tokens. **The paper accidentally wrote $>\tau$ instead of $\geq \tau$,** which was a typographical error in our notation, and we have corrected it in the revised version. We sincerely appreciate your attention to this detail.
>
> # [W3] Correction of Figure 3
>
> Thank you for pointing out the duplication issue in Figure 3. The middle subfigure should have shown the results from the SEED-V dataset. While the correct numerical values for SEED-V were provided in Appendix N, the figure in the main text did not reflect them. We have corrected Figure 3 in the revised version. We greatly appreciate your careful examination.
>
> # [Q1] Clarification of Model Version
>
> Thank you for this suggestion. In the revised paper, we now explicitly specify in the main text that the primary model used in our experiments is the **8 layers (15.17M)** version (line 352-353). We also state that our scaling-law analysis spans models from **3 layers (3.86M)** to **24 layers (146.75M)** (line 464-466). This clarification removes the need for readers to cross-check the Appendix. We appreciate your helpful feedback.
>
> # [Q2] Discussion on Sparse Code and Gating Mechanism
>
> Thank you for this interesting question. Below we clarify why the 4096×4096 decoupled codebooks are appropriate during pretraining, and why sparse gating is more suitable for finetuning than for pretraining.
>
> ## 1) Parameter Efficiency and Required Capacity of Our Tokenizer
>
> 1. $N^2$ Space Is Achieved by a Parallel Codebook Structure, Not by Increasing the Code Count
>
>    Although the representational space spans $N^2$ possible combinations, the tokenizer itself remains *parameter-efficient*. Instead of enlarging the vocabulary as in mixed designs (e.g., the 8192-code LaBraM tokenizer[1]), we decouple the space into two parallel 4096-entry codebooks. This design preserves the code count (i.e., the parameter size remains unchanged) while enabling a substantially richer representational space.
>
> 2. A Foundation-Model Tokenizer Requires Sufficient Capacity for Unseen Tasks
>
>    Table 15 shows that smaller codebook configurations saturate the frequency space (no unused codes), indicating limited capacity. For a tokenizer expected to generalize across diverse downstream datasets, slight over-completeness is preferable to exact saturation. The 4096×4096 configuration provides this necessary capacity margin while remaining compact and parameter-efficient.
>
> ## 2) Feasibility of Sparse Gating in Pretraining
>
> Sparse code gating is indeed a promising idea, and we agree that **a task-guided gating mechanism during finetuning could be highly meaningful**, as task supervision provides strong semantic signals for learning stable and sparse code routing. However, implementing such routing during *self-supervised pretraining* could be challenging.
>
> 1. **Lack of supervision for routing**
>
>    Deciding whether an EEG patch should rely more on temporal, frequency or joint structure is task-dependent. Different tasks emphasize different EEG characteristics:
>
>    - sleep staging often depends on **frequency** features
>    - event detection tends to rely on **temporal** transients
>    - cognitive or affective tasks commonly require both **temporal and frequency** information.
>
>    Since such preferences acannot be reliably inferred from unlabeled pretraining data alone, a gating module would lack supervision to learn stable routing.
>
> 2. **Premature routing could harm representation learning**
>
>    Pretraining aims to learn broad temporal and frequency structure across heterogeneous EEG. Early routing would force the model to “choose a domain” before learning is complete, preventing the tokenizer from fully capturing both structures.
>
> For these reasons, pretraining may not be the appropriate stage for sparse gating. We have added this discussion in Appendix P under *Limitation and Future Work*.
>
> [1] Jiang. et al. Large brain model for learning generic representations with tremendous EEG data in BCI. ICLR, 2024
>
> *Thank you once again for your thorough and thoughtful review. We hope our clarifications and additional analyses address your concerns. We look forward to your feedback, and are happy to provide further information if needed.*
>
> Best regards,
>
> The Authors

---

> > ### Comment · Reviewer_a5Jx · 2025-11-21
> >
> > Thank you for the authors’ clarification. After the explanation of the representation-level interpretability of CodeBrain, together with the additional experimental evidence supporting the effectiveness of the pretraining, my concerns have been addressed. I have raised my score accordingly.

---

> > > ### Author Response · Authors · 2025-11-21
> > > **Appreciation for Reviewer a5Jx’s Follow-up**
> > >
> > > Dear Reviewer a5Jx,
> > >
> > > **Thank you very much for your prompt follow-up and for updating the score.**
> > >
> > > We sincerely appreciate your careful reading and constructive suggestions, especially the insightful reframing of the representation-level interpretability discussion. We truly learned a lot from your perspective. We are glad that our rebuttal has addressed your concerns.
> > >
> > > Thank you again for your time and for helping us improve the paper.
> > >
> > > Best regards,
> > >
> > > The Authors

---

### Official Review · Reviewer_mz4F · 2025-11-01

**Soundness:** 3
**Presentation:** 4
**Contribution:** 3
**Rating:** 8
**Confidence:** 5

**Summary:**

The paper introduces CodeBrain, an EEG foundation model (EFM) designed to address key challenges in EEG analysis. CodeBrain uses a two-stage framework to improve both interpretability and efficiency in processing EEG data. In the first stage, the TFDual-Tokenizer decouples the temporal and frequency components of EEG signals into discrete tokens, enhancing the model’s ability to interpret domain-specific features. In the second stage, the EEGSSM architecture combines structured global convolutions with sliding window attention to capture both long-range dependencies and local neural events. The model is pretrained on the largest publicly available EEG corpus and tested across multiple downstream tasks and datasets, demonstrating strong generalization and performance improvements over existing baselines.

**Strengths:**

CodeBrain introduces the decoupling of temporal and frequency EEG components via the TFDual-Tokenizer, which provides domain-specific interpretability. This is a novel approach not seen in previous EEG foundation models. The manuscript is comprehensive, well-structured, and tackles important issues in EEG signal representation learning. It presents a clear solution to existing challenges in terms of both efficiency and interpretability. The paper includes extensive experiments across 8 downstream tasks and 10 datasets, demonstrating the model’s generalizability under various distribution shifts. Detailed ablation studies, scaling-law analyses, and interpretability evaluations further confirm the robustness and effectiveness of CodeBrain.

**Weaknesses:**

The paper lacks sufficient evidence to support the claim that EFMs outperform traditional models like EEGNet and STTransformer, as it does not include these models for comparison. Additionally, the justification for choosing the Swin transformer decoder over other transformer-based decoders is not well-explained, and there is a lack of detail regarding pretraining time and other crucial parameters, which hinders the assessment of model efficiency and scalability.

**Questions:**

1. Contrastive Learning: Why does splitting the data into two parts generate positive and negative samples? Are we conducting the contrastive analysis over each temporal code individually, or are we considering the entire patch (n code) for the contrastive task? Furthermore, how does the use of SimCLR contribute to solving convergence issues during training?
2. State Space Model Architecture: Can you elaborate on the architectural choices in the state space model?
3. Baseline: The paper claims that EFMs significantly outperform smaller supervised models like EEGNet and STTransformer.(“We do not include traditional supervised baselines such as EEGNet (Lawhern et al., 2018) or STTransformer (Song et al., 2021) in this comparison, as previous studies have consistently shown that EFMs significantly outperform these smaller supervised models. Our focus is therefore on evaluating the effectiveness of different foundation model designs and pretraining paradigms under comparable settings.” ) Can you provide more concrete evidence or experimental results to justify excluding these baselines from comparison? Also, how about other supervised models specifically  for each testing dataset?
4. Swin vs. Other Transformer-Based Decoders: Why is the Swin transformer decoder preferred over other transformer-based decoders? Are there specific advantages that make Swin more suitable for EEG analysis, and if so, can you provide additional experimental results or explanations?
5. Pretraining Time and Details: The paper lacks specific details about the pretraining time and other parameters. Could you clarify the pretraining setup, including the duration and computational resources involved, as well as any other important details that would help evaluate the practicality and scalability of the model?

---

> ### Author Response · Authors · 2025-11-21
> **Response to Reviewer mz4F [1/n]**
>
> We sincerely thank you for your positive feedback and for recognizing our study's **novelty**, **interpretability**, **presentation** and **comprehensive validation**.
>
> We address your concerns as follow:
>
> # [Q1] Justification for Contrastive Learning
>
> Thank you for this important question. We clarify the design of our contrastive objective below.
>
> **1) Construction of Positive and Negative Samples**
>
> We split each EEG segment into two halves because temporal segments originating from the **same** sample naturally share underlying temporal dependencies (e.g., continuity of rhythms and morphology); therefore, the two halves form a **positive pair**. For example, we divide an original 30s signal into two parts: the first 15s and the last 15s. The representations of these two parts are positive samples for each other. Negative samples come from **other samples in the same batch**, as EEG segments from different subjects or sessions exhibit distinct temporal characteristics. This setup follows standard contrastive learning practices for physiological signals.
>
> **2) The contrastive loss is applied at the *segment level***
>
> Similar to answer1, we additionally clarify that we perform contrastive learning over the aggregated representation of each half (i.e., the whole segment), not over individual temporal codes.
>
> **3) SimCLR Enhances Convergence by Providing an Easier Optimization Objective**
>
> SimCLR improves convergence because temporal VQ codebooks trained with direct raw-signal reconstruction often face instability due to noise and non-stationarity. The contrastive objective replaces the difficult reconstruction target with a smoother and more learnable alignment objective, stabilizing encoder gradients and providing a more favorable optimization landscape. Empirically, adding SimCLR eliminates the non-convergence observed in pure reconstruction-based temporal tokenizer training in Appendix F.
>
> # [Q2] Rationale for Architectural Choice
>
> We use SGConv as the main SSM model for the following two reasons.
>
> 1. Compared to other sequence models, such as LSTM or Transformer, SGConv has an efficiency advantage in computing long sequences, with a computational complexity of O(nlogn). This is more advantageous for data like EEG with high sampling rates and long sampling periods.
> 2. Compared to other SSM models like Mamba, SGConv is essentially a full-size convolutional neural network accelerated by FFT, which in our experience offers advantages in common EEG classification tasks. Mamba's selective compression mechanism makes it easy to overlook some sequence information, while a global convolution can retain all the information. At the same time, we also designed SWA to enhance the model's local perception ability.

---

> ### Author Response · Authors · 2025-11-21
> **Response to Reviewer mz4F [2/n]**
>
> # [Q3] Baseline Comparison
>
> Thank you for this valuable suggestion. Following your recommendation, we have added 4 widely used non-foundation baseline: **EEGNet**[1], **EEGConformer**[2], **ContraWR**[3] and **ST-Transformer**[4] for evidence support. These models represent compact CNN, Transformer, and contrastive-learning based architectures that are commonly adopted in EEG decoding tasks.
>
> Below, we provide the performance comparison on all 10 downstream datasets used in our paper. These results are also reported in Table 2 and Appendix J of the revised paper. Across all ten downstream datasets, CodeBrain consistently achieves higher performance than these non-foundation baselines.
>
> |||FACED|||SEED-V||
> | -------------------- | :-------------------: | :-----------------: | :-------------------: | :-----------------: | :-----------------: | :-------------------: |
> |**Model**|**Cohen’s Kappa**|**Weighted F1**|**Balanced Accuracy**|**Cohen’s Kappa**|**Weighted F1**| **Balanced Accuracy**|
> |EEGNet|0.3342 ± 0.0251|0.4124 ± 0.0141|0.4090 ± 0.0122|0.1006 ± 0.0143|0.2749 ± 0.0098|0.2961 ± 0.0102|
> |EEGConformer|0.3858 ± 0.0186|0.4514 ± 0.0107|0.4559 ± 0.0125|0.1772 ± 0.0174|0.3487 ± 0.0136|0.3537 ± 0.0112|
> |ContraWR|0.4231 ± 0.0151|0.4887 ± 0.0078|0.4887 ± 0.0078|0.1905 ± 0.0188|0.3544 ± 0.0121| 0.3546 ± 0.0105|
> | ST-Transformer|0.4137 ± 0.0133|0.4795 ± 0.0096|0.4810 ± 0.0079|0.1083 ± 0.0121|0.2833 ± 0.0105|0.3052 ± 0.0072|
> | **CodeBrain (Ours)** |**0.5406** ± 0.0084|**0.5953** ± 0.0113|**0.5941** ± 0.0098|**0.2735** ± 0.0032| **0.4235** ± 0.0022 |  **0.4137** ± 0.0023  |
> |||**ISRUC_S3**|||**BCIC 2020-T3**||
> |**Model**|**Cohen’s Kappa**|**Weighted F1**|**Balanced Accuracy**|**Cohen’s Kappa**|**Weighted F1**| **Balanced Accuracy**|
> | EEGNet|0.7396 ± 0.0155|0.7407 ± 0.0184|0.7121 ± 0.0134|0.4413 ± 0.0102|0.3016 ± 0.0123|    0.4413 ± 0.009|
> | EEGConformer|0.7482 ± 0.0164|0.7501 ± 0.0211|0.7212 ± 0.0181|0.4488 ± 0.0154|0.3133 ± 0.0183|0.4506 ± 0.0133|
> | ContraWR|0.7493 ± 0.0150|0.7513 ± 0.0185|0.7226 ± 0.0164|   0.4407 ± 0.0182|0.3078 ± 0.0218|0.4257 ± 0.0162|
> | ST-Transformer|0.7388 ± 0.0195|0.7399 ± 0.0223|0.7116 ± 0.0197|0.4247 ± 0.0138|0.2941 ± 0.0159|    0.4126 ± 0.0122|
> | **CodeBrain (Ours)** |  **0.7671** ± 0.0091  | **0.8202** ± 0.0071|**0.7856** ± 0.0031|**0.5127** ± 0.0065 | **0.6101** ± 0.0053 |  **0.6101** ± 0.0052  |
> |||**Mental Arithmetic**|||**CHB-MIT**||
> |**Model**|**AUROC**|**AUC-PR**| **Balanced Accuracy** |**AUROC**|**AUC-PR**| **Balanced Accuracy**|
> |EEGNet| 0.7321 ± 0.0108|0.5763 ± 0.0102|0.6770 ± 0.0116|0.8048 ± 0.0136|0.1914 ± 0.0182|0.5658 ± 0.0106|
> |EEGConformer|0.7424 ± 0.0128|   0.5829 ± 0.0134|0.6805 ± 0.0123|0.8226 ± 0.0170|0.2209 ± 0.0215|0.5976 ± 0.0141|
> |ContraWR|0.7332 ± 0.0082|0.5787 ± 0.0164| 0.6631 ± 0.0097|   0.8103 ± 0.0144|0.2279 ± 0.0183|0.6351 ± 0.0122    |
> |ST-Transformer|0.7132 ± 0.0174|0.5672 ± 0.0259|0.6631 ± 0.0173|0.8237 ± 0.0491   |0.1422 ± 0.0094|0.5915 ± 0.0195|
> |**CodeBrain (Ours)**|**0.8707** ± 0.0209|**0.7177** ± 0.0421|**0.7514** ± 0.0203|**0.8961** ± 0.0174|**0.4377** ± 0.0288|**0.7273** ± 0.0240|
> |||**ISRUC_S1**|||**SHU-MI**||
> | **Model**|**Cohen’s Kappa**|**Weighted F1**|**Balanced Accuracy** |**AUROC**|**AUC-PR**| **Balanced Accuracy**|
> | EEGNet|0.7040 ± 0.0173|0.7513 ± 0.0124|0.7154 ± 0.0121|0.6283 ± 0.0152|0.6311 ± 0.0142|0.5889 ± 0.0177|
> | EEGConformer|0.7143 ± 0.0162|0.7634 ± 0.0151|0.7400 ± 0.0133|0.6351 ± 0.0101|0.6370 ± 0.0093|0.5900 ± 0.0107|
> | ContraWR|0.7178 ± 0.0156|0.7610 ± 0.0137|0.7402 ± 0.0126|0.6273 ± 0.0113|0.6315 ± 0.0105|0.5873 ± 0.0128|
> | ST-Transformer|0.7013 ± 0.0352|0.7681 ± 0.0175|0.7381 ± 0.0205|0.6431 ± 0.0111|0.6394 ± 0.0122|0.5992 ± 0.0206|
> | **CodeBrain (Ours)** |**0.7476** ± 0.0040|**0.8020** ± 0.0018|**0.7835** ± 0.0033|**0.7124** ± 0.0050|**0.7166** ± 0.0106|**0.6431** ± 0.0066|
> |||**TUEV**|||**TUAB**||
> |**Model**|**Cohen’s Kappa**|**Weighted F1**|**Balanced Accuracy**|**AUROC**|**AUC-PR**|**Balanced Accuracy**|
> |EEGNet|0.3577 ± 0.0155|0.6539 ± 0.0120|0.3876 ± 0.0143|0.7642 ± 0.0036|0.8299 ± 0.0043|0.8412 ± 0.0031|
> |EEGConformer|0.3967 ± 0.0195|0.6983 ± 0.0152|0.4074 ± 0.0164|0.7758 ± 0.0049|0.8427 ± 0.0054|0.8445 ± 0.0038|
> |ContraWR|0.3912 ± 0.0237|0.6893 ± 0.0136|0.4384 ± 0.0349|0.7746 ± 0.0041|0.8421 ± 0.0104|0.8456 ± 0.0074|
> |ST-Transformer|0.3765 ± 0.0306|0.6823 ± 0.0190|0.3984 ± 0.0228|0.7966 ± 0.0023|0.8521 ± 0.0026|0.8707 ± 0.0019|
> |**CodeBrain (Ours)**|**0.6912** ± 0.0101|**0.8362** ± 0.0048|**0.6428** ± 0.0062| **0.9030** ± 0.0009|**0.9100** ± 0.0006|**0.8294** ± 0.0013|
>
> [1] Lawhern. et al. EGNet: a compact convolutional neural network for EEG-based brain-computer interfaces. Journal of neural engineering, 2018
>
> [2] Song. et al. EEG conformer: Convolutional transformer for EEG decoding and visualization. TNSRE, 2022
>
> [3] Yang. et al. Self-supervised EEG representation learning for automatic sleep staging. JMIR AI, 2021
>
> [4] Song. et al. Transformer-based spatial-temporal feature learning for EEG decoding. ArXiv, 2021

---

> ### Author Response · Authors · 2025-11-21
> **Response to Reviewer mz4F [3/n]**
>
> # [Q4] Analysis on Sliding Window Attention
>
> We would like to clarify that we do not use the full Swin Transformer as a decoder; instead, we only adopt its sliding-window attention to model local inter-patch dependencies. We evaluated this choice against standard attention on three representative downstream datasets (SEED-V, FACED, ISRUC_S3), and sliding-window attention consistently outperformed global attention.
>
> The results on the SEED-V dataset are shown in the table below.
>
> | Model Variant                         |    Cohen’s Kappa    |     Weighted F1     |  Balanced Accuracy  | Δ (vs. w/o Attn) |
> | :------------------------------------ | :-----------------: | :-----------------: | :-----------------: | :--------------: |
> | CodeBrain w/o Attention               |   0.2561 ± 0.0051   |   0.4106 ± 0.0042   |   0.4019 ± 0.0034   |        –         |
> | CodeBrain w/ Full Attention           |   0.2580 ± 0.0098   |   0.4116 ± 0.0072   |   0.4025 ± 0.0066   |   +0.0019    |
> | **CodeBrain w/ SWA Attention (ours)** | **0.2735** ± 0.0032 | **0.4235** ± 0.0022 | **0.4137** ± 0.0023 |   **+0.0174**    |
>
>
> The results on the FACED dataset are shown in the table below.
>
> | Model Variant                         |    Cohen’s Kappa    |     Weighted F1     |  Balanced Accuracy  | Δ (vs. w/o Attn) |
> | :------------------------------------ | :-----------------: | :-----------------: | :-----------------: | :--------------: |
> | CodeBrain w/o Attention               |   0.5192 ± 0.0092   |   0.5792 ± 0.0093   |   0.5736 ± 0.0075   |        –         |
> | CodeBrain w/ Full Attention           |   0.5267 ± 0.0095   |   0.5826 ± 0.0091   |   0.5814 ± 0.0078   |     +0.0075      |
> | **CodeBrain w/ SWA Attention (ours)** | **0.5406** ± 0.0084 | **0.5953** ± 0.0113 | **0.5941** ± 0.0098 |   **+0.0214**    |
>
> The results on the ISRUC-S3 dataset are shown in the table below.
>
> | Model Variant                         |    Cohen’s Kappa    |     Weighted F1     |  Balanced Accuracy  | Δ (vs. w/o Attn) |
> | :------------------------------------ | :-----------------: | :-----------------: | :-----------------: | :--------------: |
> | CodeBrain w/o Attention               |   0.7359 ± 0.0324   |   0.7950 ± 0.0259   |   0.7621 ± 0.0311   |        –         |
> | CodeBrain w/ Full Attention           |   0.7587 ± 0.0151   |   0.8191 ± 0.0101   |   0.7831 ± 0.0074   |   +0.0228    |
> | **CodeBrain w/ SWA Attention (ours)** | **0.7671** ± 0.0091 | **0.8202** ± 0.0071 | **0.7856** ± 0.0031 |   **+0.0312**    |
>
>
> The improvement is also intuitive since the global dependencies are already modeled by the global convolution in our SSM backbone, and adding another global attention introduces redundancy. In contrast, sliding-window attention provides complementary local context, which aligns with our multi-scale design and yields better performance.
>
> # [Q5] Pretraining Time and Details
>
> Thank you for raising this important point. In the revised paper, we have added a full description of the pretraining setup (line 352-354). In brief, all pretraining was performed on NVIDIA A100 GPUs. The tokenizer was trained for around **10 hours** on six A100-40GB GPUs, and the 8-layer backbone was pretrained for roughly **one day** on two A100-40GB GPUs.
>
> *Thank you once again for your thorough and thoughtful review, and encouraging feedback. We hope our clarifications and additional analyses address your concerns. We look forward to your feedback, and are happy to provide further information if needed.*
>
> Best regards,
>
> The Authors

---

> > ### Comment · Reviewer_mz4F · 2025-11-28
> > **The authors have addressed my questions, and congratulations**
> >
> > Thank you to the authors for their thoughtful and comprehensive responses to my comments. I appreciate the time and effort invested in addressing each concern in detail, as well as the additional analyses and clarifications provided. The revisions and explanations offered by the authors significantly improve the clarity, rigor, and overall quality of the work. I acknowledge the substantial work involved in preparing these responses, value the authors’ commitment to strengthening the manuscript, and congratulate them on the substantial improvements achieved in this revised version.

---

> > > ### Author Response · Authors · 2025-11-28
> > > **Appreciation for Reviewer mz4F's Follow-up**
> > >
> > > Dear Reviewer mz4F,
> > >
> > > **Thank you very much for your follow-up and for maintaining the score.**
> > >
> > > We sincerely appreciate your careful examination of the methodological rationale and technical details, as well as your constructive suggestions. We are glad that our rebuttal has addressed your concerns.
> > >
> > > Thank you again for your time and for helping us improve the paper.
> > >
> > > Best regards,
> > >
> > > The Authors

---

### Official Review · Reviewer_uKaR · 2025-11-01

**Soundness:** 3
**Presentation:** 3
**Contribution:** 3
**Rating:** 6
**Confidence:** 5

**Summary:**

This paper proposes a novel EEG foundation model named CodeBrain, with its core contributions as follows:  1) Designing the **TFDual-Tokenizer**, which generates discrete tokens by decoupling the heterogeneous temporal and frequency signals of EEG. This not only expands the representation space but also achieves domain-specific interpretability;  2) Proposing the **EEGSSM architecture**, which combines structured global convolution with sliding window attention. By simulating the brain's small-world topology, it efficiently captures both sparse long-range dependencies and local dependencies;  3) After pretraining on the large-scale public dataset, CodeBrain achieves superior generalization performance compared to 5 existing EEG foundation models across 8 downstream tasks (covering emotion recognition, sleep staging, etc.) and 10 datasets. Additionally, the robustness and scalability of the model are validated through ablation experiments and scaling law analyses.

**Strengths:**

1. **Novel Architectural Design**: The decoupling of time-frequency tokenization addresses a critical limitation in existing EEG models, where joint encoding often conflates neural dynamics. This aligns with neurophysiological understanding that brain activity operates across dissociable temporal and spectral dimensions.
2. **Comprehensive Benchmarking**: Testing on 8 diverse tasks (spanning cognitive, motor, and clinical domains) provides strong evidence for generalizability, a key requirement for foundation models.
3. **Preliminary Interpretability**: The authors make meaningful preliminary efforts to link model components (i.e., temporal/frequency tokens) to known neural correlates (e.g., sleep spindles, slow waves, and spectral rhythms like delta activity).

**Weaknesses:**

This work demonstrates promising prospects for advancing EEG foundation models; however, addressing the following weaknesses is critical to validating its contributions to artificial intelligence and neuroscience.

## Model-Related Weaknesses

1. **Unsupported "Approximate Independence" Assumption in Proposition 2.1 (Section 2.2):**

   A key concern is that EEG temporal representations $X_t$ and frequency representations $X_f$ are not independent—they exhibit inherent correlations. Yet, Proposition 2.1 assumes they are "approximately independent" to derive the distortion bound of the decoupled codebook, with no empirical analysis or theoretical justification provided. This casts doubt on the generalizability of Proposition 2.1.

2. **Missing Codebook Initialization Strategy for TFDual-Tokenizer (Section 2.2):**

   The TFDual-Tokenizer uses temporal and frequency codebooks, but the **codebook initialization method** is unspecified. For vector quantization based models, initialization directly affects codebook utilization and training stability. The absence of this detail hinders understanding of how codebook initialization influences model performance.

3. **Unclear Structure and Mechanism of the "Lightweight Convolutional Module" (Section 2.3):**

   The "lightweight convolutional module" for dynamic positional embeddings is poorly elaborated. Its unspecified architecture and the unexplained mechanism for capturing inter-channel correlations and adapting to unseen channels weaken the claims about the model's adaptability.

4. **Unvalidated Rationale for SGConv Kernel Parameters (Section 2.3):**

   The SGConv layer optimizes the convolution kernel $\bar{K}$ via "sparse parameterization" and "kernel decay," with the decay coefficient fixed at $\alpha=0.5$. However, no comparisons with other $\alpha$ values are conducted to assess impacts on model performance. The paper provides no basis for selecting $\alpha=0.5$ (e.g., cross-validation results, neurophysiological plausibility), relying solely on an empirical value.

## Experiment-Related Weaknesses

1. **Incomplete Baseline Comparisons (Section 3):**

   While the paper compares CodeBrain with 5 existing EEG foundation models, it omits traditional non-foundation models (e.g., EEGNet[1], EEG-Conformer[2]) as baselines. This makes it impossible to intuitively quantify the advantage of CodeBrain as a foundation model.

2. **Unexplained Importance of the Gating Mechanism (Section 3.4):**

   Ablation experiments show that removing the gating module causes a substantial drop in Cohen’s Kappa: 52.3% for FACED, 19.1% for SEED-V, and 28.6% for ISRUC S3. Despite this critical impact, the gating module is not a core innovation of the work. The paper fails to explain **why this specific gating design was adopted** (e.g., compatibility with EEGSSM) or **why it drives such significant performance gains**, weakening the justification for its inclusion.

3. **Insufficient Details for Reproducibility (Appendices I):**

   Appendix I lacks details on learning rate decay (e.g., warm-up steps for the Cosine scheduler), which is critical for stable training. The temperature parameter $τ$ for the contrastive loss (SimCLR) is not specified, and the rationale for its selection is not provided.

## Analysis-Related Weaknesses

1. **Inadequate Quantitative Support for Interpretability Claims (Figure 4):**

   Figure 4 illustrates associations between temporal tokens and slow waves, and frequency tokens and delta rhythms, but only reports percentage values. No **statistical significance tests** (e.g., comparing with the matching rate of random tokens to rule out chance correlations) are performed, leaving the interpretability conclusions without statistical validation.

2. **Missing Trade-off Analysis for Scaling Laws (Appendix N):**

   Appendix N shows diminishing performance returns as model parameters increase from 3M to 150M, but no analysis of **computational efficiency trade-offs** is provided (e.g., FLOPs, training time, memory usage for different model sizes). Without quantifying whether marginal performance gains justify increased computational costs, the work fails to guide model size selection for practical applications.

[1] Lawhern V J, Solon A J, Waytowich N R, et al. EEGNet: a compact convolutional neural network for EEG-based brain–computer interfaces[J]. Journal of neural engineering, 2018, 15(5): 056013.

[2] Song Y, Zheng Q, Liu B, et al. EEG conformer: Convolutional transformer for EEG decoding and visualization[J]. IEEE Transactions on Neural Systems and Rehabilitation Engineering, 2022, 31: 710-719.

**Questions:**

1. What is the codebook initialization strategy for the TFDual-Tokenizer? Do different strategies have significant impacts?
2. What is the structure of the lightweight convolutional module? How does it capture inter-channel correlations?
3. What is the rationale for the SGConv kernel parameters?
4. What is the value of the temperature parameter τ in the SimCLR contrastive loss, and what is its selection basis?
5. Why is the gating mechanism so crucial for model performance?

---

> ### Author Response · Authors · 2025-11-21
> **Response to Reviewer uKaR [1/n]**
>
> We sincerely thank you for your positive feedback and for recognizing our study's **novelty**, **comprehensive validation** and **interpretability**.
>
> We address your concerns as follow:
>
> # [W1.1] Theory Assumption
>
> Thank you for this insightful comment. We agree that, from a strict signal-processing perspective, temporal and frequency representations of EEG are mathematically coupled. We would like to clarify that our intention in Proposition 2.1 was not to assume strict statistical independence, but to describe a *representation-level* abstraction: after TFConv encoding, the temporal and frequency embeddings behave as heterogeneous feature sources with different optimization objectives.
>
> Empirically, using a shared codebook causes conflicts during learning, which consistent with LaBraM’s reported non-convergence when reconstructing the temporal waveforms, and ablation results in our paper(Table 3). These findings support our use of two decoupled codebooks rather than implying true signal-level independence.
>
> To avoid misunderstanding, we have added a remark after proposition 2.1 (line 1236-1241) to clarify that *approximate independence assumption* refers to the learned representations, not to the raw EEG signal.
>
> # [W1.2+Q1] Codebook Initialization Strategy
>
> The codebooks in the TFDual-Tokenizer are **randomly initialized and then $L^2$normalized**, following common practice in VQ-based models to ensure stable early-stage updates and avoid dominance by high-norm embeddings. In our experiments, training remained stable and codebook utilization was balanced as shown in Appendix E and F. We have added this detail in the revised paper (line 1582).
>
> # [W1.3+Q2] Lightweight Convolutional Module
>
> We view the ability to handle diverse EEG channel configurations as a baseline requirement for EEG foundation models. In our work, we follow the well-established ACPE module from CBraMod[1], which uses a **single depthwise 2D convolution** with an asymmetric kernel to generate dynamic positional embeddings. This allows the model to learn relative positions, achieving strong adaption to unseen channel configurations. We have added these details in the *Method* section of the revised paper (line 251-254).
>
> # [W1.4+Q3] Rationale for SGConv Kernel Parameters
>
> Our choice of $\alpha=0.5$ follows the natural “half-decay per step” commonly used in spatial kernel design of the the decay coefficient. To empirically validate this choice, we have conducted a sensitivity analysis across three representative downstream datasets (FACED, ISRUC_S3, SEED-V).
>
> We provide the sensitivity analysis on FACED dataset below:
>
> | $\alpha$ Value  |    Cohen’s Kappa    |     Weighted F1     |  Balanced Accuracy  | Δ Kappa (vs. $\alpha=0.5$) |
> | :-------------: | :-----------------: | :-----------------: | :-----------------: | :----------------: |
> | **0.5 (default)** | **0.5406** ± 0.0084 | **0.5953** ± 0.0113 | **0.5941** ± 0.0090 |         –          |
> |       0.1        |   0.5295 ± 0.0080   |   0.5853 ± 0.0074   |   0.5839 ± 0.0069   |      −0.0111       |
> |       0.9       |   0.4681 ± 0.0069   |   0.5270 ± 0.0065   |   0.5314 ± 0.0050   |      −0.0725       |
>
> We provide the sensitivity analysis on SEED-V dataset below:
>
> |   $\alpha$ Value    |    Cohen’s Kappa    |     Weighted F1     |  Balanced Accuracy  | Δ Kappa (vs. $\alpha=0.5$) |
> | :-----------------: | :-----------------: | :-----------------: | :-----------------: | :----------------: |
> | **α = 2 (default)** | **0.2735** ± 0.0032 | **0.4235** ± 0.0022 | **0.4137** ± 0.0023 |         –          |
> |         0.1          |   0.2629 ± 0.0017   |   0.4159 ± 0.0006   |   0.4044 ± 0.0017   |      −0.0106       |
> |         0.9         |   0.2318 ± 0.0074   |   0.3851 ± 0.0141   |   0.3819 ± 0.0046   |      −0.0417       |
>
> We provide the sensitivity analysis on ISRUC_S3 dataset below:
>
> | $\alpha$ Value  |    Cohen’s Kappa    |     Weighted F1     |  Balanced Accuracy  | Δ Kappa (vs. $\alpha=0.5$) |
> | :-------------: | :-----------------: | :-----------------: | :-----------------: | :----------------: |
> | **0.5 (default)** | **0.7671** ± 0.0091 | **0.8202** ± 0.0071 | **0.7856** ± 0.0031 |         –          |
> |       0.1        |   0.7132 ± 0.0646   |   0.7834 ± 0.0523   |   0.7575 ± 0.0621   |      −0.0539       |
> |       0.9       |   0.6073 ± 0.0230   |   0.6966 ± 0.0176   |   0.6564 ± 0.0058   |      −0.1598       |
>
> Across all three datasets, $\alpha = 0.5$ consistently provides the best performance, while a larger value such as $\alpha = 0.9$ leads to mild degradation and a very small value such as $\alpha = 0.1$ results in clear drops across all metrics. These results validate the default choice of $\alpha = 0.5$, which achieves a stable balance between locality preservation and kernel sparsification.
>
> We have included these results in Appendix K.5 of the revised paper.
>
> [1] Wang. et al. CBraMod: A Criss-Cross Brain Foundation Model for EEG Decoding. ICLR, 2025

---

> ### Author Response · Authors · 2025-11-21
> **Response to Reviewer uKaR [2/n]**
>
> # [W2.1] Baseline Comparison
>
> Thank you for this valuable suggestion. Following your recommendation, we have added 4 widely used non-foundation baseline: **EEGNet**[1], **EEGConformer**[2], **ContraWR**[3] and **ST-Transformer**[4]. These models represent compact CNN, Transformer, and contrastive-learning based architectures that are commonly adopted in EEG decoding tasks.
>
> Below, we provide the performance comparison on all 10 downstream datasets used in our paper. These results are also reported in Table 2 and Appendix J of the revised paper. Across all ten downstream datasets, CodeBrain consistently achieves higher performance than these non-foundation baselines.
>
> |||FACED|||SEED-V||
> | -------------------- | :-------------------: | :-----------------: | :-------------------: | :-----------------: | :-----------------: | :-------------------: |
> |**Model**|**Cohen’s Kappa**|**Weighted F1**|**Balanced Accuracy**|**Cohen’s Kappa**|**Weighted F1**| **Balanced Accuracy**|
> |EEGNet|0.3342 ± 0.0251|0.4124 ± 0.0141|0.4090 ± 0.0122|0.1006 ± 0.0143|0.2749 ± 0.0098|0.2961 ± 0.0102|
> |EEGConformer|0.3858 ± 0.0186|0.4514 ± 0.0107|0.4559 ± 0.0125|0.1772 ± 0.0174|0.3487 ± 0.0136|0.3537 ± 0.0112|
> |ContraWR|0.4231 ± 0.0151|0.4887 ± 0.0078|0.4887 ± 0.0078|0.1905 ± 0.0188|0.3544 ± 0.0121| 0.3546 ± 0.0105|
> | ST-Transformer|0.4137 ± 0.0133|0.4795 ± 0.0096|0.4810 ± 0.0079|0.1083 ± 0.0121|0.2833 ± 0.0105|0.3052 ± 0.0072|
> | **CodeBrain (Ours)** |  **0.5406** ± 0.0084  | **0.5953** ± 0.0113 |  **0.5941** ± 0.0098  | **0.2735** ± 0.0032 | **0.4235** ± 0.0022 |  **0.4137** ± 0.0023  |
> |||**ISRUC_S3**|||**BCIC 2020-T3**||
> |**Model**|**Cohen’s Kappa**|**Weighted F1**|**Balanced Accuracy**|**Cohen’s Kappa**|**Weighted F1**| **Balanced Accuracy**|
> | EEGNet|0.7396 ± 0.0155|0.7407 ± 0.0184|0.7121 ± 0.0134|0.4413 ± 0.0102|0.3016 ± 0.0123|    0.4413 ± 0.009|
> | EEGConformer|0.7482 ± 0.0164|0.7501 ± 0.0211|0.7212 ± 0.0181|0.4488 ± 0.0154|0.3133 ± 0.0183|0.4506 ± 0.0133|
> | ContraWR|0.7493 ± 0.0150|0.7513 ± 0.0185|0.7226 ± 0.0164|   0.4407 ± 0.0182|0.3078 ± 0.0218|0.4257 ± 0.0162|
> | ST-Transformer|0.7388 ± 0.0195|0.7399 ± 0.0223|0.7116 ± 0.0197|0.4247 ± 0.0138|0.2941 ± 0.0159|    0.4126 ± 0.0122|
> | **CodeBrain (Ours)** |  **0.7671** ± 0.0091  | **0.8202** ± 0.0071 |  **0.7856** ± 0.0031  | **0.5127** ± 0.0065 | **0.6101** ± 0.0053 |  **0.6101** ± 0.0052  |
> |||**Mental Arithmetic**|||**CHB-MIT**||
> |**Model**|**AUROC**|**AUC-PR**| **Balanced Accuracy** |**AUROC**|**AUC-PR**| **Balanced Accuracy**|
> |EEGNet| 0.7321 ± 0.0108|0.5763 ± 0.0102|0.6770 ± 0.0116|0.8048 ± 0.0136|0.1914 ± 0.0182|0.5658 ± 0.0106|
> |EEGConformer|0.7424 ± 0.0128|   0.5829 ± 0.0134|0.6805 ± 0.0123|0.8226 ± 0.0170|0.2209 ± 0.0215|0.5976 ± 0.0141|
> |ContraWR|0.7332 ± 0.0082|0.5787 ± 0.0164| 0.6631 ± 0.0097|   0.8103 ± 0.0144|0.2279 ± 0.0183|0.6351 ± 0.0122    |
> |ST-Transformer|0.7132 ± 0.0174|0.5672 ± 0.0259|0.6631 ± 0.0173|0.8237 ± 0.0491   |0.1422 ± 0.0094|0.5915 ± 0.0195|
> |**CodeBrain (Ours)**|**0.8707** ± 0.0209|**0.7177** ± 0.0421|**0.7514** ± 0.0203|**0.8961** ± 0.0174|**0.4377** ± 0.0288|**0.7273** ± 0.0240|
> |||**ISRUC_S1**|||**SHU-MI**||
> | **Model**|**Cohen’s Kappa**|**Weighted F1**|**Balanced Accuracy** |**AUROC**|**AUC-PR**| **Balanced Accuracy**|
> | EEGNet|0.7040 ± 0.0173|0.7513 ± 0.0124|0.7154 ± 0.0121|0.6283 ± 0.0152|0.6311 ± 0.0142|0.5889 ± 0.0177|
> | EEGConformer|0.7143 ± 0.0162|0.7634 ± 0.0151|0.7400 ± 0.0133|0.6351 ± 0.0101|0.6370 ± 0.0093|0.5900 ± 0.0107|
> | ContraWR|0.7178 ± 0.0156|0.7610 ± 0.0137|0.7402 ± 0.0126|0.6273 ± 0.0113|0.6315 ± 0.0105|0.5873 ± 0.0128|
> | ST-Transformer|0.7013 ± 0.0352|0.7681 ± 0.0175|0.7381 ± 0.0205|0.6431 ± 0.0111|0.6394 ± 0.0122|0.5992 ± 0.0206|
> | **CodeBrain (Ours)** |**0.7476** ± 0.0040|**0.8020** ± 0.0018|**0.7835** ± 0.0033|**0.7124** ± 0.0050|**0.7166** ± 0.0106|**0.6431** ± 0.0066|
> |||**TUEV**|||**TUAB**||
> |**Model**|**Cohen’s Kappa**|**Weighted F1**|**Balanced Accuracy**|**AUROC**|**AUC-PR**|**Balanced Accuracy**|
> |EEGNet|0.3577 ± 0.0155|0.6539 ± 0.0120|0.3876 ± 0.0143|0.7642 ± 0.0036|0.8299 ± 0.0043|0.8412 ± 0.0031|
> |EEGConformer|0.3967 ± 0.0195|0.6983 ± 0.0152|0.4074 ± 0.0164|0.7758 ± 0.0049|0.8427 ± 0.0054|0.8445 ± 0.0038|
> |ContraWR|0.3912 ± 0.0237|0.6893 ± 0.0136|0.4384 ± 0.0349|0.7746 ± 0.0041|0.8421 ± 0.0104|0.8456 ± 0.0074|
> |ST-Transformer|0.3765 ± 0.0306|0.6823 ± 0.0190|0.3984 ± 0.0228|0.7966 ± 0.0023|0.8521 ± 0.0026|0.8707 ± 0.0019|
> |**CodeBrain (Ours)**|**0.6912** ± 0.0101|**0.8362** ± 0.0048|**0.6428** ± 0.0062| **0.9030** ± 0.0009|**0.9100** ± 0.0006|**0.8294** ± 0.0013|
>
> [1] Lawhern. et al. EGNet: a compact convolutional neural network for EEG-based brain-computer interfaces. Journal of neural engineering, 2018
>
> [2] Song. et al. EEG conformer: Convolutional transformer for EEG decoding and visualization. TNSRE, 2022
>
> [3] Yang. et al. Self-supervised EEG representation learning for automatic sleep staging. JMIR AI, 2021
>
> [4] Song. et al. Transformer-based spatial-temporal feature learning for EEG decoding. ArXiv, 2021

---

> ### Author Response · Authors · 2025-11-21
> **Response to Reviewer uKaR [3/n]**
>
> # [W2.2+Q5] Gating Mechanism
>
> Although the gating mechanism is not a core architectural novelty, it plays a crucial functional role during fine-tuning. In deep, pre-trained models, overfitting can occur within only a few epochs. The sigmoid-based gate helps **selectively suppress unstable or task-irrelevant activations**, effectively reducing overfitting by modulating feature flow.
>
> Empirically, this effect is very clear: on the FACED dataset, the variant without the gate reaches a training loss of 0.5 within about only 10 epochs, while the with-gate version typically requires about 20 epochs to reach a similar level, which is consistent with our observation that such fast loss collapse reflects overfitting rather than genuine convergence. Similar trends appear in SEED-V and ISRUC_S3 dataset. This directly explains why removing the gate leads to the substantial performance drops reported in our ablation study.
>
> To improve clarity, we have added this rationale in the ablation study part in the revised paper (line 460-462).
>
> # [W2.3+Q4] Detailed Parameter
>
> We have added the mentioned hyperparameters to Appendix I of the revised paper for reproducibility. Specifically:
>
> - **Learning rate schedule.** We use AdamW with an initial learning rate of $5\times10^{-4}$. A short warm-up of 5 steps is applied before the cosine decay scheduler.
> - **Contrastive temperature.** The temperature parameter for the SimCLR-style contrastive loss is 0.5. We choose a relatively higher temperature because our goal is to encourage similarity across *neighboring temporal segments* of the same sample. EEG recordings from different subjects exhibit substantial inter-sample variability; a smaller temperature would overly amplify these between-sample differences, contradicting the purpose of applying contrastive learning in our setting.
>
> # [W3.1] Quantitative Support for Interpretability Claims
>
> Following your suggestion, we conducted a random-token baseline analysis by sampling 5 random codes from the same codebook and evaluating each on 500 assigned patches. Across the three representative tokens examined in Appendix B.2, the learned tokens show statistically higher matching rates than the random baseline. Specifically, the learned tokens exhibit matching rates of about 5.3%, 7.0%, and 40.4%, whereas the random-token baselines are only about 0.8%, 1.4%, and 2.0%. A one-sided proportion test confirms that all differences are statistically significant (p < 0.05), indicating that the observed associations are highly unlikely due to chance.
>
> # [W3.2] Trade-off Analysis for Scaling Laws
>
> we have added a detailed computational trade-off analysis, including FLOPs, throughput, and GPU memory usage for all model scales. To ensure reproducibility of the throughput measurements, all reported efficiency experiments were conducted on the same hardware setup (one NVIDIA H200 GPU).
>
> | Layer | Hidden Size | Params  | FLOPs  | Throughput (iters/s) | GPU Memory |
> | :---: | :---------: | :-----: | :----: | :------------------: | :--------: |
> |   3   |     128     |  3.96M  |  1.7G  |         4.9          |   4.87G    |
> |   3   |     200     |  6.82M  | 3.35G  |         4.62         |   5.63G    |
> |   4   |     200     |  8.49M  | 4.43G  |         3.77         |   6.48G    |
> |   5   |     200     | 10.16M  | 5.51G  |         3.67         |   7.33G    |
> |   6   |     200     | 11.83M  | 6.58G  |         3.52         |   8.10G    |
> |   7   |     200     | 13.50M  | 7.66G  |         2.94         |   8.95G    |
> |   8   |     200     | 15.17M  | 8.74G  |         2.78         |   9.79G    |
> |  12   |     256     | 34.38M  | 19.04G |         1.84         |   15.41G   |
> |  24   |     384     | 146.75M | 72.99G |         1.47         |   38.43G   |
>
> The results show that although performance increases as model size increases from 3M to 15M parameters, the improvements plateau quickly beyond this point. Larger models bring only minimal accuracy gains while incurring substantially higher computational cost. Therefore, the 8-layer model offers better balance between performance and efficiency, and we use it as the primary version in the main paper. We have added these trade-off analysis in Appendix N.3 of our revised paper.
>
> *Thank you once again for your thorough and thoughtful review, and encouraging feedback. We hope our clarifications and additional analyses address your concerns. We look forward to your feedback, and are happy to provide further information if needed.*
>
> Best regards,
>
> The Authors

---

> > ### Comment · Reviewer_uKaR · 2025-11-22
> >
> > Thank you for the authors’ detailed response. The additional experimental results and mechanistic explanations have addressed most of my concerns. However, there are still two points that need further clarification:
> >
> > 1. In Line 275, you mention *“where $α$ denotes the decay coefficient, usually chosen to be 0.5”*, but in the final setting you adopted $α = 2.0$. Is this a typo error? If not, it would be helpful to include an ablation experiment with $α = 2.0$, as this is a commonly used value.
> > 2. I also have a question regarding the pretraining setup, where the number of channels was fixed at 19. Since the ACPE positional encoding in CBraMod is learned through a 2D CNN that relies on local spatiotemporal structures, its spatial embedding may be strongly coupled with the specific electrode layout. When applied to downstream datasets with different channel configurations (e.g., 64‑channel or 128‑channel), the model would need to relearn the channel embedding during fine‑tuning. Wouldn’t this limit the transferability of the pretrained representations?

---

> ### Author Response · Authors · 2025-11-23
> **Response to Reviewer uKaR [4/n]**
>
> Dear Reviewer uKaR,
>
> Thank you very much for your prompt follow-up. We are glad that our previous responses have resolved most of your concerns, and we appreciate the opportunity to provide further clarification.
>
> # [Q1] Clarification on the Decay Coefficient
>
> Thank you for your careful reading. The value reported in the main text ($\alpha$ = 0.5) is the correct one. In our previous rebuttal, *we mistakenly wrote its reciprocal*. We sincerely apologize for this typo. Consequently, the reported ablations were run with $\alpha$ = **0.1, 0.5, 0.9**. We have additionally evaluated the setting $\alpha=2$ across three representative downstream datasets. The complete and corrected ablation results are provided below.
>
> |||FACED|||
> |-|-|-|-|-|
> |**α Value**|**Cohen’s Kappa**|**Weighted F1**|**Balanced Accuracy**|**Δ Kappa (vs. $\alpha$=0.5)**|
> |**0.5 (default)**|**0.5406** ± 0.0084|**0.5953** ± 0.0113|**0.5941** ± 0.0090|–|
> |0.1|0.5295 ± 0.0080|0.5853 ± 0.0074|0.5839 ± 0.0069|−0.0111|
> |0.9|0.4681 ± 0.0069|0.5270 ± 0.0065|0.5314 ± 0.0050|−0.0725|
> |2.0|0.4782 ± 0.0037|0.5282 ± 0.0068|0.5360 ± 0.0035|−0.0624|
> |||**SEED-V**|||
> |**0.5 (default)**|**0.2735** ± 0.0032|**0.4235** ± 0.0022|**0.4137** ± 0.0023|–|
> |0.1|0.2629 ± 0.0017|0.4159 ± 0.0006|0.4044 ± 0.0017|−0.0106|
> |0.9|0.2318 ± 0.0074|0.3851 ± 0.0141|0.3819 ± 0.0046|−0.0417|
> |2.0|0.2332 ± 0.0077|0.3851 ± 0.0090|0.3809 ± 0.0069|−0.0403|
> |||**ISRUC_S3**|||
> |**0.5 (default)**|**0.7671** ± 0.0091|**0.8202** ± 0.0071|**0.7856** ± 0.0031|–|
> | 0.1|0.7132 ± 0.0646|0.7834 ± 0.0523|0.7575 ± 0.0621| −0.0539|
> | 0.9|0.6073 ± 0.0230|0.6966 ± 0.0176|0.6564 ± 0.0058| −0.1598|
> | 2.0|0.5600 ± 0.0492|0.6572 ± 0.0430|0.6227 ± 0.0427| −0.2071|
>
> We have also corrected this error in Appendix K.5 of the revised paper and updated the corresponding part in the previous rebuttal.
>
> # [Q2] Clarification on Pretraining Setup
>
> Thank you for this thoughtful question. We clarify that **because ACPE encodes *neighbourhood structures* rather than electrode labels, it learns layout-agnostic positional priors, allowing us to adopt a compact and fully reproducible pretraining setup without compromising transferability.** We elaborate on this rationale below.
>
> ## 1) Empirically, CodeBrain transfers well across 10 downstream datasets with 6-62 channels
>
> These strong empirical results confirm the robustness and transferability of the learned representations.
>
> ## 2) ACPE encodes layout-agnostic *local spatiotemporal structures*
>
> ACPE's CNN operates on the **patch grid**, not on absolute electrode indices. Because convolution learns correlations within its local receptive field, **it captures the relationships among neighbouring patches instead of the identities of the channels they originate from**. Therefore, it learns local *relative* spatial neighbourhoods and cross-channel interactions that are invariant to channel layout.
>
> Importantly, the ablation study in CBraMod has demonstrated that ACPE outperforms both absolute positional embeddings and no positional encoding, indicating that **the pretrained model has learned transferable positional priors rather than relying on downstream finetuning to adapt to new electrode layouts.**
>
> ## 3) ACPE’s strong transferability enables a compact and fully reproducible pretraining setup.
>
> Because ACPE provides a more transferable positional prior than earlier EFMs that rely on absolute channel encodings, it **greatly simplifies the pretraining pipeline**. Hard-coded approaches such as LaBraM[2] use extensive multi-dataset pretraining, including private datasets, to cover diverse channel layouts. While such comprehensive efforts make valuable contributions, they also introduce substantial data-processing complexity and reduce reproducibility.
>
> In contrast, ACPE’s layout-agnostic design allows us to pretrain on a **single, public, large-scale dataset (TUEG)** using the unified pipeline proposed by CBraMod. This configuration is increasingly recognized as a reproducible benchmark in the community. For example, CSBrain [3] follows the same setting.
>
> We directly adopt this established configuration and inherit its advantages. By leveraging ACPE and the TUEG-based pretraining pipeline, our setup **improves reproducibility and comparison fairness, enabling evaluations to focus on architectural contributions**.
>
> *We sincerely appreciate your careful verification of nearly every technical detail and the constructive discussion. Your feedback has substantially improved the rigor of our work. We are very happy to continue the discussion if further clarification would be helpful.*
>
> Best regards,
>
> The Authors
>
> **References**
>
> [1] Wang. et al. CBraMod: A Criss-Cross Brain Foundation Model for EEG Decoding. ICLR, 2025
>
> [2] Jiang. et al. Large brain model for learning generic representations with tremendous EEG data in BCI. ICLR, 2024
>
> [3] Zhou. et al. CSBrain: A Cross-scale Spatiotemporal Brain Foundation Model for EEG Decoding. NeurIPS, 2025

---

> > ### Comment · Reviewer_uKaR · 2025-11-24
> >
> > Thank you for the authors' detailed response. I have one remaining concern regarding the ACPE positional encoding. The reported downstream evaluations appear to have been conducted under a full fine‑tuning paradigm, which allows the model to adapt the ACPE module to the specific spatial structure of each dataset when sufficient data are available. I am curious whether the transferability would remain as strong in low‑data scenarios. To more comprehensively assess the generalization ability of the model, I suggest that the authors include additional experiments, such as linear probing or few‑shot evaluations, to examine its performance under data‑limited conditions.

---

> > > ### Author Response · Authors · 2025-11-25
> > > **Response to Reviewer uKaR [5/n]**
> > >
> > > Dear Reviewer uKaR,
> > >
> > > Thank you very much for your prompt follow-up and for raising this remaining insightful concern. We will provide a detailed analysis to address this point.
> > >
> > > # 1) Current EEG Foundation Models (EFMs) primarily adopt full fine-tuning
> > >
> > > Our full fine-tuning evaluation setting is currently the default paradigm used across EFMs [1-4], partly because:
> > >
> > > 1. EEG exhibits **extremely low signal-to-noise ratio and substantial domain shift** across montages, devices, subjects, and recording paradigms.
> > > 2. Current EFMs remain **moderate in scale** (typically tens of millions of parameters), making full fine-tuning computationally feasible
> > >
> > > # 2) Experiments Under 30% Few-Shot Settings in Appendix O of the Original Submission
> > >
> > > We fully agree that evaluating performance under data-limited conditions is highly important,  so **our original submission already included experiments under a 30% few-shot setting in Appendix O** across two representative datasets, providing initial evidence of CodeBrain’s transferability. For completeness, we restate these results here.
> > >
> > > |                 | FACED               |                     |                       |
> > > | --------------- | ------------------- | ------------------- | --------------------- |
> > > | **Methods**     | **Cohen’s Kappa**   | **Weighted F1**     | **Balanced Accuracy** |
> > > | BIOT (30%)      | 0.2573 ± 0.0346     | 0.3501 ± 0.0341     | 0.3428 ± 0.0329       |
> > > | LaBraM (30%)    | 0.2672 ± 0.0371     | 0.3548 ± 0.0325     | 0.3513 ± 0.0315       |
> > > | CBraMod (30%)   | 0.3239 ± 0.0265     | 0.4056 ± 0.0256     | 0.4035 ± 0.0233       |
> > > | CodeBrain (30%) | **0.3356** ± 0.0253 | **0.4114** ± 0.0225 | **0.4104** ± 0.0281   |
> > > |                 | **SEED-V**          |                     |                       |
> > > | **Methods**     | **Cohen’s Kappa**   | **Weighted F1**     | **Balanced Accuracy** |
> > > | BIOT (30%)      | 0.1775 ± 0.0425     | 0.3492 ± 0.0416     | 0.3505 ± 0.0375       |
> > > | LaBraM (30%)    | 0.2044 ± 0.0384     | 0.3700 ± 0.0321     | 0.3686 ± 0.0305       |
> > > | CBraMod (30%)   | 0.2291 ± 0.0264     | 0.3886 ± 0.0255     | 0.3877 ± 0.0236       |
> > > | CodeBrain (30%) | **0.2376** ± 0.0284 | **0.3943** ± 0.0252 | **0.3902** ± 0.0271   |
> > >
> > > These results show that CodeBrain outperforms representative baselines under 30% few-shot setting.

---

> > > ### Author Response · Authors · 2025-11-25
> > > **Response to Reviewer uKaR [6/n]**
> > >
> > > # 3) Additional Experiments Under 10% Few-Shot Settings
> > >
> > > To further assess the generalization ability of CodeBrain under more challenging low-resource conditions, we additionally conducted experiments in **10% few-shot setting**. We compare CodeBrain against two representative baselines:
> > >
> > > - **CBraMod**[3], which also uses ACPE and allows us to evaluate the **architectural contribution** of CodeBrain
> > > - **LaBraM**[1], which relies on hard-coded positional embeddings, allowing us to examine the transferability advantage provided by learnable positional embedding like ACPE.
> > >
> > > The 10% few-shot results shown below demonstrate that CodeBrain outperforms representative baselines.
> > >
> > >
> > > |                 | FACED               |                     |                       |
> > > | --------------- | ------------------- | ------------------- | --------------------- |
> > > | **Methods**     | **Cohen’s Kappa**   | **Weighted F1**     | **Balanced Accuracy** |
> > > | LaBraM (10%)    | 0.1358 ± 0.0163     | 0.2247 ± 0.0196     | 0.2265 ± 0.0174       |
> > > | CBraMod (10%)   | 0.1632 ± 0.0156     | 0.2595 ± 0.0138     | 0.2604 ± 0.0148       |
> > > | CodeBrain (10%) | **0.1716** ± 0.0101 | **0.2599** ± 0.0104 | **0.2654** ± 0.0093   |
> > > |                 | **SEED-V**          |                     |                       |
> > > | **Methods**     | **Cohen’s Kappa**   | **Weighted F1**     | **Balanced Accuracy** |
> > > | LaBraM (10%)    | 0.0302 ± 0.0065     | 0.2194 ± 0.0079     | 0.2228 ± 0.0091       |
> > > | CBraMod (10%)   | 0.0174 ± 0.0029     | 0.2071 ± 0.0125     | 0.2127 ± 0.0023       |
> > > | CodeBrain (10%) | **0.1690** ± 0.0170 | **0.3410** ± 0.0133 | **0.3331** ± 0.0138   |
> > >
> > > # 4) Additional Experiments Under Linear Probing Settings
> > >
> > > To directly evaluate the quality of the pretrained representations and assess performance under compute-limited conditions, we additionally conduct experiments under a **linear probing setting**, where *all backbone parameters are frozen and only a one-layer linear classifier is trained*. We also compared our model with CBraMod and LaBraM.
> > >
> > > As shown in the linear probing results below, CodeBrain provides superior representations compared to representative baselines.
> > >
> > >
> > > |             | FACED               |                     |                       |
> > > | ----------- | ------------------- | ------------------- | --------------------- |
> > > | **Methods** | **Cohen’s Kappa**   | **Weighted F1**     | **Balanced Accuracy** |
> > > | LaBraM      | 0.3026 ± 0.0121     | 0.3789 ± 0.0154     | 0.3812 ± 0.0148       |
> > > | CBraMod     | 0.3378 ± 0.0139     | 0.4123 ± 0.0117     | 0.4146 ± 0.0123       |
> > > | CodeBrain   | **0.3587** ± 0.0136 | **0.4311** ± 0.0109 | **0.4327** ± 0.0127   |
> > > |             | **SEED-V**          |                     |                       |
> > > | **Methods** | **Cohen’s Kappa**   | **Weighted F1**     | **Balanced Accuracy** |
> > > | LaBraM      | 0.1941 ± 0.0184     | 0.3457 ± 0.0135     | 0.3413 ± 0.0144       |
> > > | CBraMod     | 0.2239 ± 0.0053     | 0.3823 ± 0.0041     | 0.3791 ± 0.0050       |
> > > | CodeBrain   | **0.2302** ± 0.0166 | **0.3889** ± 0.0154 | **0.3829** ± 0.0136   |
> > >
> > > # 5) Conclusion
> > >
> > > These results consistently show that CodeBrain outperforms representative strong baselines across multiple data‑limited conditions.
> > >
> > > - CodeBrain surpasses CBraMod despite both using ACPE, highlighting **the architectural advantage of our framework**.
> > > - CodeBrain outperforms LaBraM, indicating that incorporating **learnable positional embeddings such as ACPE does not harm transferability** and may offer advantages over hard-coded embeddings.
> > > - We also observed that few-shot performance is lower than linear probing. This is likely because full fine-tuning with very limited data is highly sensitive to overfitting and distribution shift. In contrast, linear probing provides a more stable evaluation of the pretrained features and shows that the representations already contain meaningful structure. It further indicates that the **ACPE module exhibits a degree of transferability, rather than relying solely on re-learning during fine-tuning**.
> > >
> > > We have added these new experiments and analysis to Appendix O in the revised version of the paper.
> > >
> > > *Thank you once again for your careful consideration and for the time you have dedicated to discussing our work. If any further clarification would be helpful, we remain fully available and are more than happy to continue the conversation.*
> > >
> > > Best regards,
> > >
> > > The Authors
> > >
> > > **References**
> > >
> > > [1] Jiang. et al. Large brain model for learning generic representations with tremendous EEG data in BCI. ICLR, 2024
> > >
> > > [2] Wang. et al. Eegpt: Pretrained transformer for universal and reliable representation of eeg signals. NeurIPS, 2024
> > >
> > > [3] Wang. et al. CBraMod: A Criss-Cross Brain Foundation Model for EEG Decoding. ICLR, 2025
> > >
> > > [4] Zhou. et al. CSBrain: A Cross-scale Spatiotemporal Brain Foundation Model for EEG Decoding. NeurIPS, 2025

---

> > > > ### Comment · Reviewer_uKaR · 2025-11-25
> > > >
> > > > Thank you for the authors' detailed response and the additional experiments. The updated results and clarifications have resolved my previous concern, and I will raise my score accordingly.

---

> > > > > ### Author Response · Authors · 2025-11-25
> > > > > **Appreciation for Reviewer uKaR's Follow-up**
> > > > >
> > > > > Dear Reviewer uKaR,
> > > > >
> > > > > **Thank you very much for your prompt follow-ups throughout the discussion and for updating the score.**
> > > > >
> > > > > We sincerely appreciate your continued interaction during the discussion period, as well as your careful examination of the details and constructive suggestions. We are glad that our rebuttal has addressed your concerns.
> > > > >
> > > > > Thank you again for your time, thoughtful engagement, and for helping us improve the paper.
> > > > >
> > > > > Best regards,
> > > > >
> > > > > The Authors

---

### Author Response · Authors · 2025-11-28
**Global Response and Revisions Summary**

Dear Reviewers, AC, SAC, and PC,

We sincerely thank all reviewers for the time, constructive feedback, and highly engaging discussion throughout the rebuttal period. We are grateful for the detailed examination of our work and for the many positive assessments we received regarding its **novelty** (uKaR, mz4F, a5Jx), **comprehensive evaluation** (uKaR, mz4F, a5Jx, ATKr), **strong generalizability** (uKaR, mz4F), **robustness** (mz4F, ATKr), **representation-level interpretability enabled by the proposed tokenization** (uKaR, mz4F, a5Jx, ATKr), **presentation** (mz4F, a5Jx, ATKr), and the **theoretical soundness** (a5Jx).

We particularly appreciate the reviewers’ in-depth follow-ups during the discussion phase, which greatly helped us strengthen the manuscript. We have incorporated all feasible suggestions and clarified every point raised. **All revisions have been marked in blue in the revised paper.** Below we summarize the key revisions made in response to reviewers’ comments:

- Following the suggestions of uKaR, mz4F, and ATKr, we added **four non-foundation model baselines** (EEGNet, EEGConformer, ContraWR, ST-Transformer) (Table 2; Appendix J).
- We added additional **reproducibility details and methodological rationales**:
  - Following uKaR, we clarified that the **assumption** in Proposition 2.1 is at the representation level rather than the raw-signal level (lines 1236–1241, p. 23).
  - Following uKaR, we added the codebook initialization strategy (line 1582, p. 30), the learning rate schedule (line 1586, p. 30), and the contrastive learning temperature parameter (line 1593, p. 30).
  - Following uKaR, we added a description of the lightweight convolutional module in ACPE (lines 251–254, p. 5).
  - Following uKaR, we clarified why the gating mechanism contributes substantially during fine-tuning (lines 460–462, p. 9).
  - Following mz4F, we added details on pretraining time and computational setup (lines 351–353, p. 7).
  - Following a5Jx, we clarified that our interpretability claims refer to representation-level interpretability and revised wording throughout the paper to avoid potential misinterpretation.
  - Following a5Jx, we corrected the mathematical notation (line 1068, p. 20).
  - Following a5Jx, we corrected Figure 3 (p. 9).
  - Following ATKr, we added citations to FAPEX, Beatrix, and Scatterformer in the *Related Work* sections.
- We added additional **ablation studies**:
  - Following uKaR, we added an ablation on the SGConv decay coefficient (Appendix K.5).
  - Following uKaR and a5Jx, we added analyses of the specific model variant used in the main paper, including parameter–performance trade-offs (lines 463–469, p. 9; Appendix N.3).
  - Following a5Jx, we included more detailed ablations on codebook contributions (Appendix B.4).
  - Following ATKr, we added sub-band ablation experiments (Appendix K.6).
- We conducted additional **robustness evaluations**:
  - Following uKaR, we added low-resource experiments, including 10% few-shot settings and linear probing under compute-limited conditions (Appendix O).
  - Following ATKr, we added robustness analyses under non-stationary conditions (Appendix M.3).
  - Following ATKr, we added region-based channel-drop robustness experiments (Appendix M.2).
- Following ATKr, we further added **efficiency analyses** comparing CodeBrain with baseline models (Appendix L).

We thank all reviewers once again for the constructive discussions, careful reading, and valuable follow-up questions. Your feedback has substantially strengthened our work. *We hope the revised manuscript and rebuttal address all remaining concerns, and we remain fully open to providing any additional information if needed.*

Best Regards,

The Authors

---

### Meta-Review · Area_Chair_HYiJ · 2026-01-17

**Summary:**

The reviews are heterogeneous, and the case is on the boundary.

Overall, there are many positive qualities of the manuscript. It seems quite performant, and the experiments are extensive. On the otherhand, I agree with multiple of the weaknesses brought up by reviewers, that the interpretability analysis is weak at best (`uKaR`), and the removal of gamma band (`ATKr`).

It further is certainly incremental (`ATKr`), though I feel that such a quality isn't fully disqualifying. The method has some novelty in its frequency/time token split, but indeed, it does produce one more foundation model into a crowded domain with modest improvement.

On the merits of its experiments and the novel architecture, I recommend acceptance, but I admit that this work is still borderline with respect to the overall pile.

**Reviewer Concerns:**

Interpretation is insufficiently addressed. This quality is ambiguous by nature, but including the word in the title of the paper would promise more. Instead we find a very good foundation model with relatively similar interpretable pieces.

**Reviewer Scores:**

None,

---

### Decision · Program_Chairs · 2026-01-26

Accept (Poster)